# Doubly-Robust Estimation of Counterfactual Policy Mean Embeddings

**Houssam Zenati**
Gatsby Computational Neuroscience Unit
University College London
h.zenati@ucl.ac.uk

**Bariscan Bozkurt**
Gatsby Computational Neuroscience Unit
University College London
bariscan.bozkurt.23@ucl.ac.uk

**Arthur Gretton**
Gatsby Computational Neuroscience Unit
University College London
Google Deepmind
arthur.gretton@gmail.com

## Abstract

Estimating the distribution of outcomes under counterfactual policies is critical for decision-making in domains such as recommendation, advertising, and healthcare. We propose and analyze a novel framework—Counterfactual Policy Mean Embedding (CPME)—that represents the entire counterfactual outcome distribution in a reproducing kernel Hilbert space (RKHS), enabling flexible and nonparametric distributional off-policy evaluation. We introduce both a plug-in estimator and a doubly robust estimator; the latter enjoys improved convergence rates by correcting for bias in both the outcome embedding and propensity models. Building on this, we develop a doubly robust kernel test statistic for hypothesis testing, which achieves asymptotic normality and thus enables computationally efficient testing and straightforward construction of confidence intervals. Our framework also supports sampling from the counterfactual distribution. Numerical simulations illustrate the practical benefits of CPME over existing methods.

## 1 Introduction

Effective decision-making requires anticipating the *outcomes* of *actions* driven by given *policies* [1]. This is especially critical when decisions rely on historical data—whether experimentation is limited or infeasible [2], or even under sequential designs [3]. For instance, doctors weigh drug effects before prescribing [4], and businesses predict revenue impact from ads [5]. Off-Policy Evaluation (OPE) addresses this challenge by estimating the effect of a *target policy* using data sampled under a different *logging policy*. Each logged record includes covariates (e.g., user or patient data), an action (e.g., recommendation or treatment), and the resulting outcome (e.g., engagement or health status). The goal is to evaluate the expected outcome under the *target policy*, which involves inferring *counterfactual* outcomes—what would have happened under the alternative target policy.

Although many works have focused on estimating the mean of outcome distributions, for example, with the policy expected risk (payoff) [6] or the average treatment effects [7]- and their variants thereof - seminal works have considered inference on counterfactual distributions of outcomes [8] instead. The developing field of distributional reinforcement learning (RL) [9] and distributional OPE [10, 11] provides insights on distribution-driven decision making, which goes beyond expected policy risks. Indeed, reasoning on such distributions allows using alternative risk measure such as conditional value-at-risk (CVaR) [12], higher moments or quantiles of the distribution [13, 14].

39th Conference on Neural Information Processing Systems (NeurIPS 2025).

However, most existing approaches leverage cumulative distribution functions (CDF) [15, 16] which are not suited for inference on more complex and structured outcomes.

Conversely, counterfactual mean embeddings (CME) [17] represent the outcome distributions as elements in a reproducing kernel Hilbert space (RKHS) [18, 19] and allow inference for distributions over complex outcomes such as images, sequences, and graphs [20]. Such embeddings leverage kernel mean embeddings [21], a framework for representing a distribution maintaining all of its information for sufficiently rich kernels [22, 23]. This framework allows to quantify distributional treatment effects [17], perform hypothesis testing [24] or even sample [25, 26] from the counterfactual outcome distribution. Recent works have employed counterfactual mean embeddings for causal inference in the context of distributional treatment effects [27–29], however these approaches have not been applied to OPE and limited mostly to binary treatments. Developing analogous distributional embeddings for counterfactual outcomes under target policies could enable a range of new applications, including principled evaluation, hypothesis testing, and efficient sampling from complex outcome distributions.

Our estimates will employ *doubly robust* methods, which have become a central tool in causal inference due to the desirable property of consistency if either the outcome model or the propensity model is correctly specified [30, 31]. DR estimators have since been studied under various functional estimation tasks, including treatment effects [32] and policy evaluation [33]. These estimators originally leverage efficient influence functions [34–36] and sample-splitting techniques [37, 38] to achieve bias reduction and enable valid inference in high-dimensional and nonparametric settings [39, 40]. Recently, doubly robust tests have been introduced for kernel treatment effects [28, 29]. Moreover, an extension of semiparametric efficiency theory and efficient influence functions has been proposed for differentiable Hilbert-space-valued parameters [41]. Leveraging these efficient influence functions to build doubly robust estimators of counterfactual mean embeddings would therefore enable more theoretically grounded distributional OPE.

In this work, we propose a novel approach to distributional OPE that embeds the counterfactual outcome distribution, which procedure we term as Counterfactual Policy Mean Embedding (CPME). Our contributions are as follows: i) First, we define and formalize the CPME in the distributional OPE problem. We proposing a plug-in estimator, and analyze its consistency with a convergence rate of up to $\mathcal{O}(n^{-1/4})$ under standard regularity assumptions involving kernels and underlying distributions. ii) We then derive the Hilbert-space–valued efficient influence function of the CPME to propose a doubly robust estimator, and establish its convergence in the RKHS with an improved consistency rate of up to $\mathcal{O}(n^{-1/2})$ under the same assumptions. iii) Consequently, we propose an efficient doubly robust and asymptotically normal statistic which allows a computationally efficient kernel test. iv) We demonstrate that our estimators enable sampling from the outcome distribution. v) Finally, we provide numerical simulations on synthetic and semi-synthetic data, including structured outcomes, to support our claims in a range of scenarios.

The remainder of the paper is organized as follows. Section 2 formalizes the CPME framework. Sections 3 and 4 introduce, respectively, the nonparametric plug-in estimator with consistency guarantees and an efficient-influence-function-based estimator with improved convergence. Section 5 illustrates applications to hypothesis testing and sampling, Section 6 reports numerical results, and Section 7 concludes.

## 2 Counterfactual policy mean embeddings

We begin by formalizing the counterfactual policy mean embedding (CPME) framework, which provides a kernel-based foundation for distributional OPE.

### 2.1 Distributional off-policy evaluation setting

We are given an observational dataset generated from interactions between a decision-making system and units with covariates $x_i$. For each instance $i \in \{1, \ldots, n\}$, a context $x_i$ was drawn i.i.d. from an unknown distribution $P_X$, i.e., $x_i \sim P_X$. Given $x_i$, an action $a_i$ was sampled from a *logging policy* $\pi_0 \in \Pi$, such that $a_i \sim \pi_0(\cdot \mid x_i)$. Following the potential outcomes framework [7], we denote the set of potential outcomes by $\{Y(a)\}_{a \in \mathcal{A}}$, and observe the realized outcome $y_i = Y(a_i) \sim P_{Y|X, A=x_i, a_i}$. The data-generating process is therefore characterized by the joint distribution $P_0 = P_{Y|X,A} \times \pi_0 \times P_X$. The dataset consists of $n$ i.i.d. *logged* observations

$\{(x_i, a_i, y_i)\}_{i=1}^n \sim P_0$. The action space $\mathcal{A}$ may be either finite or continuous. For notational purposes, we will also abbreviate the joint distribution $P_\pi = P_{Y|X,A} \times \pi \times P_X$.

Given only this logged data from $P_0$, the goal of *distributional off-policy evaluation* is to estimate $\nu(\pi)$, the distribution of outcomes induced by a target policy $\pi$ belonging to the policy set $\Pi$:

$$\nu(\pi) = \mathbb{E}_{\pi \times P_X} \left[ P_{Y|X,A}(Y(a)) \right]. \tag{1}$$

$\nu(\pi)$ represents the marginal distribution of outcomes over $\pi \times P_X$, therefore when actions are taken from the target policy $\pi \in \Pi$ in a *counterfactual* manner. Compared to "classical" OPE where only the average of the outcome distribution is considered, distributional RL and OPE [9–11] allows defining further risk measures [16] depending for example on quantiles of the outcome distribution [42]. In this work we focus on distributional OPE leveraging distributional embeddings.

## 2.2 Distributional embeddings

In this work, we employ kernel methods to represent, compare, and estimate probability distributions. For both domains $\mathcal{F} \in \{\mathcal{A} \times \mathcal{X}, \mathcal{Y}\}$, we associate an RKHS $\mathcal{H}_\mathcal{F}$ of real-valued functions $\ell : \mathcal{F} \to \mathbb{R}$, where the point evaluation functional is bounded [43]. Each RKHS is uniquely determined by its continuous, symmetric, and positive semi-definite kernel function $k_\mathcal{F} : \mathcal{F} \times \mathcal{F} \to \mathbb{R}$. We denote the induced RKHS inner product and norm in $\mathcal{H}_\mathcal{F}$ by $\langle \cdot, \cdot \rangle_{\mathcal{H}_\mathcal{F}}$ and $\| \cdot \|_{\mathcal{H}_\mathcal{F}}$, respectively. Throughout the paper, we denote the feature maps $k_{\mathcal{A}\mathcal{X}}(\cdot, (a,x)) = \phi_{\mathcal{A}\mathcal{X}}(a,x)$ and $k_\mathcal{Y}(.,y) = \phi_\mathcal{Y}(y)$ for $\mathcal{H}_{\mathcal{A}\mathcal{X}}$ and $\mathcal{H}_\mathcal{Y}$, the RKHSs over $\mathcal{A} \times \mathcal{X}$ and $\mathcal{Y}$. See Appendix 9.1 for further background.

Building upon the framework of Muandet et al. [17], we define the *counterfactual policy mean embedding* (CPME) [1] as:

$$\chi(\pi) = \mathbb{E}_{P_\pi} \left[ \phi_\mathcal{Y}(Y(a)) \right], \tag{2}$$

which is the kernel mean embedding of the counterfactual distribution $\nu(\pi)$. This *causal* embedding allows to i) perform statistical tests [24], (ii) sample from the counterfactual distribution [25, 26] or even (iii) recover the counterfactual distribution from the mean embedding [22]. While Muandet et al. [17] introduced the *counterfactual mean embedding* (CME) of the distribution of the potential outcome $Y(a)$ under a single, hard intervention for binary treatments, we focus instead on counterfactual embeddings of stochastic interventions for more general policy action and sets $\Pi, \mathcal{A}$ in the OPE problem. Next, we provide further assumptions for the identification of the causal CPME.

## 2.3 Identification

In seminal works, Rosenbaum and Rubin [44] and Robins [45] established sufficient conditions under which causal functions—defined in terms of potential outcomes $Y(a)$—can be identified from observable quantities such as the outcome $Y$, treatment $A$, and covariates $X$. These conditions are commonly referred to as selection on observables.

**Assumption 1.** *(Selection on Observables). Assume i) Consistency: $Y = Y(a)$ when $A = a$, ii) Conditional exchangeability : $Y(a) \perp A \mid X$, iii) Strong positivity: $\inf_{P \in \mathcal{P}} \operatorname{ess\,inf}_{a \in \mathcal{A}, x \in \mathcal{X}} \pi_0(a \mid x) > 0$, where the essential infimum is under $P_X$.*

Assumption 1 (i), combined with the no-interference assumption (which rules out interference between units, ensuring that each individual's outcome depends only on their own treatment assignment) is also known as the stable unit treatment value assumption (SUTVA). Condition (ii) asserts that, conditional on covariates $X$, the treatment assignment is independent of the potential outcomes, implying that treatment is as good as randomized once we condition on $X$—thus ruling out unmeasured confounding. Finally, (iii) guarantees that all treatment levels have a nonzero probability of being assigned for any covariate value with positive density, preventing deterministic treatment allocation and ensuring overlap in the support of treatment assignment. Note that this mild condition on the essential infimum is slightly stronger than the common positivity assumption [46]; as in [17] this will prove useful for the importance weighting in the counterfactual mean embedding. Now define the following conditional mean embedding [47] of the distribution $P_{Y|X,A}$:

$$\mu_{Y|A,X}(a,x) = \mathbb{E}_{P_{Y|X,A}}[\phi_\mathcal{Y}(Y) \mid A = a, X = x]. \tag{3}$$

---

[1]Despite its name, the CPME represents the mean embedding of interventional (do-) distributions—thus corresponding to the second rung of Pearl's ladder. The term "counterfactual" is retained for consistency with prior work [e.g. 17], where potential outcomes $Y(t)$ are colloquially called "counterfactuals".

Under the three conditions stated earlier, the CPME can be identified as follows.

**Proposition 2.** *(Identified Counterfactual Policy Mean Embedding) Let us assume that Assumption 1 holds, then the counterfactual policy mean embedding can be written as:*

$$\chi(\pi) = \mathbb{E}_{\pi \times P_X} \left[ \mu_{Y|A,X}(a, x) \right]. \tag{4}$$

Further details on this Proposition are given in Appendix 9.3. We are now in position to use our RKHS assumption to provide a nonparametric estimator of the CPME in the next section.

## 3 A plug-in estimator

We further require some regularity conditions on the RKHS, which are commonly assumed [48, 49].

**Assumption 3.** *(RKHS regularity conditions). Assume that i)* $k_{\mathcal{AX}}$*, and* $k_{\mathcal{Y}}$ *are continuous and bounded, i.e.,* $\sup_{a,x \in \mathcal{A} \times \mathcal{X}} \|\phi_{\mathcal{AX}}(a, x)\|_{\mathcal{H}_{\mathcal{AX}}} \leqslant \kappa_{a,x}, \quad \sup_{y \in \mathcal{Y}} \|\phi(y)\|_{\mathcal{H}_{\mathcal{Y}}} \leqslant \kappa_y$*; ii)* $\phi_{\mathcal{AX}}(a, x)$*, and* $\phi_{\mathcal{Y}}(y)$ *are measurable; iii)* $k_{\mathcal{Y}}$ *is characteristic.*

Let $\mathcal{C}_{Y|A,X} \in S_2(\mathcal{H}_{\mathcal{AX}}, \mathcal{H}_{\mathcal{Y}})$ be the conditional mean operator, where $S_2(\mathcal{H}_{\mathcal{AX}}, \mathcal{H}_{\mathcal{Y}})$ denotes the Hilbert space of the Hilbert-Schmidt operators [50] from $\mathcal{H}_{\mathcal{AX}}$ to $\mathcal{H}_{\mathcal{Y}}$. Under the regularity condition that $\mathbb{E}[h(Y)|A = \cdot, X = \cdot] \in \mathcal{H}_{\mathcal{AX}}$ for all $h \in \mathcal{H}_{\mathcal{Y}}$, the operator $\mathcal{C}_{Y|A,X}$ exists[2] such that $\mu_{Y|A,X}(a, x) = \mathcal{C}_{Y|A,X}\{\phi_{\mathcal{AX}}(a, x)\}$. Moreover, define $\mu_\pi$, the joint policy-context mean embedding as:

$$\mu_\pi = \mathbb{E}_{\pi \times P_X} \left[ \phi_{\mathcal{AX}}(a, x) \right]. \tag{5}$$

Importantly, note that $\mu_\pi$ denotes the joint embedding of actions under $\pi$ and covariates under $P_X$. We now state the following proposition, with its proof provided in Appendix 10.1.

**Proposition 4.** *(Decoupling via joint policy-context mean embedding) Suppose Assumptions 1 and 3 hold. Then, the CPME can be expressed as:*

$$\chi(\pi) = C_{Y|A,X}\mu_\pi \tag{6}$$

This result suggests that an estimator for the counterfactual policy mean embedding $\chi(\pi)$ can be constructed by pluging-in an estimate $\hat{C}_{Y|A,X}$ of the conditional mean operator and an estimate $\hat{\mu}_\pi$ of the joint policy-context mean embedding. The resulting plug-in estimator $\hat{\chi}_{pi}$ writes:

$$\hat{\chi}_{pi}(\pi) = \hat{C}_{Y|A,X} \, \hat{\mu}_\pi. \tag{7}$$

Thus, we first require the estimation of the conditional mean embedding operator $\hat{C}_{Y|A,X}$. To do so, given the regularization parameter $\lambda > 0$, we consider the following learning objective [53]:

$$\hat{\mathcal{L}}^c(C) = \frac{1}{n} \sum_{i=1}^n \|\phi_{\mathcal{Y}}(y_i) - C\{\phi_{\mathcal{AX}}(a_i, x_i)\}\|_{\mathcal{H}_{\mathcal{Y}}}^2 + \lambda \|C\|_{S_2(\mathcal{H}_{\mathcal{AX}}, \mathcal{H}_{\mathcal{Y}})}^2, \quad C \in S_2(\mathcal{H}_{\mathcal{AX}}, \mathcal{H}_{\mathcal{Y}})$$

whose minimizer is denoted as, $\hat{C}_{Y|A,X} = \arg\min_{C \in S_2(\mathcal{H}_{\mathcal{AX}}, \mathcal{H}_{\mathcal{Y}})} \hat{\mathcal{L}}^c(C)$. Given the observations $\{a_i, x_i, y_i\}_{i=1}^n$, the solution to this problem [53] is given by $\hat{C}_{Y|A,X} = \hat{C}_{Y,(A,X)} \left( \hat{C}_{A,X} + \lambda I \right)^{-1}$, where $\hat{C}_{Y,(A,X)} = \frac{1}{n} \sum_{i=1}^n \phi_{\mathcal{Y}}(y_i) \otimes \phi_{\mathcal{AX}}(a_i, x_i)$ and $\hat{C}_{A,X} = \frac{1}{n} \sum_{i=1}^n \phi_{\mathcal{AX}}(a_i, x_i) \otimes \phi_{\mathcal{AX}}(a_i, x_i)$. Since we work with infinite-dimensional feature mappings, it is convenient to express the solution in terms of feature inner products (i.e., kernels), using the representer theorem [54]:

$$\hat{\mu}_{Y|A,X}(a, x) = \hat{C}_{Y|A,X}\phi_{\mathcal{AX}}(a, x) = \sum_{i=1}^n \phi_{\mathcal{Y}}(y_i)\beta_i(a, x) = \Phi_{\mathcal{Y}}\boldsymbol{\beta}(a, x),$$

---

[2]The conditional mean operator formulation is valid under mild smoothness assumptions ensuring that the conditional mean function $F_\star(x) = \mathbb{E}[\phi(Y) \mid X = x]$ belongs to a Sobolev-type vector-valued RKHS. In particular, Li et al. [51] show that when the Matérn kernel is used on the $X$-space and $F_\star \in H^m(X; \mathcal{H}_Y)$, the induced operator $C_{Y|X}$ exists and acts boundedly from $\mathcal{H}_X$ to $\mathcal{H}_Y$. A regression-based alternative [52] can also be used, but the operator view is often more convenient for theoretical analysis.

where $\Phi_{\mathcal{Y}} = [\phi_{\mathcal{Y}}(y_1) \quad \dots \quad \phi_{\mathcal{Y}}(y_n)]$, $\boldsymbol{\beta}(a,x) = (K_{AX,AX} + n\lambda I)^{-1}(K_{AX,ax})$, $K_{AX,AX}$ is the kernel matrix over the set $\{(a_i, x_i)\}_{i=1}^n$, and $K_{AX,ax}$ is the kernel vector between training points $\{(a_i, x_i)\}_{i=1}^n$ and the target variable $(a, x)$.

We provide a bound on the estimation error $\|\hat{C}_{Y|A,X} - C_{Y|A,X}\|_{S_2}$ in Appendix 10, using the main result from Li et al. [48]. This bound plays a key role in establishing the consistency of both our plug-in and doubly robust estimators. The derivation relies on the widely adopted source condition (SRC) [55, 56] and eigenvalue decay (EVD) assumptions, as formalized in Assumptions 15 and 16.

Second, we estimate the joint policy-context mean embedding $\hat{\mu}_\pi$ which represents the joint embedding of the distribution $\pi \times P_X$. We employ the empirical kernel mean embedding estimator [24], which takes the following explicit form for discrete action spaces:

$$\hat{\mu}_\pi = \frac{1}{n} \sum_{i=1}^n \sum_{a \in \mathcal{A}} \phi_{\mathcal{AX}}(a, x_i)\pi(a|x_i). \tag{8}$$

For continuous action spaces, we propose an empirical kernel mean embedding estimator combined with a resampling strategy over actions (see Appendix 10.3). In OPE, the target policy is specified by the designer, making these estimators directly applicable. A summary of the plug-in estimation procedure is provided in Appendix 10.3 (see pseudo-code in Algorithms 3, 4).

Importantly, our plug-in estimator of the CPME differs substantially from the approach of Muandet et al. [17]. First, they propose and analyze an importance-weighted estimator for kernel treatment effects under the assumption of known propensities. Second, although they discuss an application to OPE, their method lacks a formal analysis and is not evaluated beyond linear kernels.

Next, we arrive at the theoretical guarantee for the plug-in estimator under the conditions we presented and the common Assumptions 15, 16, 17—stated in Appendix 10.2 for space considerations.

**Theorem 5.** *(Consistency of the plug-in estimator). Suppose Assumptions 1, 3, 15, 16 and 17 hold. Set $\lambda = n^{-1/(c+1/b)}$, which is rate optimal regularization. Then, with high probability, $\hat{\chi}_{pi}$ defined in Equation (7) achieves the convergence rate with parameters $b \in (0,1]$ and $c \in (1,3]$*

$$\|\hat{\chi}_{pi}(\pi) - \chi(\pi)\|_{\mathcal{H}_y} = \mathcal{O}\left[r_C(n, b, c)\right] = \mathcal{O}\left[n^{-(c-1)/\{2(c+1/b)\}}\right].$$

Here, $r_C(n, b, c)$ bounds the error in estimating $\hat{C}_{Y|A,X}$, with $c$ and $b$ denoting the source condition and spectral decay parameters (Assumptions 15 and 16). Appendix 10.2 provides a proof with explicit constants hidden in the $\mathcal{O}(\cdot)$ notation. Smaller values of $b$ indicates slower eigenvalue decay of the correlation operator defined in Assumption 15; as $b \to \infty$ the effective dimension is finite. The parameter $c$ controls the smoothness of the conditional mean operator $C_{Y|A,X}$. The optimal rate is $n^{-1/4}$, which can be attained when $c = 3$ [51]. The convergence rate is obtained by combining two minimax-optimal rates: $n^{-(c-1)/\{2(c+1/b)\}}$ for the conditional mean operator $C_{Y|A,X}$[51, Theorem 3], and $n^{-1/2}$ for kernel mean embedding $\mu_\pi$ [57, Theorem 1]. In the next section, we introduce a doubly robust estimator of the CPME that improves upon this rate.

## 4 An efficient influence function-based estimator

To design our estimator, we rely on semiparametric efficiency theory for *Hilbert space–valued parameters* [34, 41]. As in the finite-dimensional setting, *efficient influence functions* (EIFs) [34–36] quantify the local sensitivity of a target parameter to perturbations of the underlying distribution. When they exist, they enable the construction of *one-step estimators* [58, 59], which correct the plug-in bias and often exhibit *doubly robust* properties [32]. Assuming the existence of an EIF $\psi^\pi$ for the CPME $\chi(\pi)$, the one-step estimator takes the form

$$\hat{\chi}_{dr}(\pi) = \hat{\chi}_{pi}(\pi) + \sum_{i=1}^n \hat{\psi}^\pi(a_i, x_i, y_i). \tag{9}$$

One-step estimators rely on *pathwise differentiability*, which describes how the target parameter varies under infinitesimal perturbations of the data distribution [60, 61]. When this condition holds, the EIF coincides with the *Riesz representer* of the pathwise derivative [41]—in our case, the unique RKHS

element whose inner product with any score function recovers the target parameter's directional derivative. This derivative captures the target parameter's *first-order sensitivity* to distributional changes, and projecting it onto the model's tangent space yields the optimal linear correction that removes the plug-in estimator's leading bias (see Appendix 11.1).

To define the corresponding one-step estimator, we assume the spaces $\mathcal{A}, \mathcal{X}, \mathcal{Y}$ are Polish (Assumption 17). Under these conditions, we derive the result stated below and prove it in Appendix 11.2.

**Lemma 4.1.** *(Existence and form of the efficient influence function). Suppose Assumptions 1 and 17 hold. Then, the CPME $\chi(\pi)$ admits an EIF which is P-Bochner square integrable and takes the form*

$$\psi^\pi(y, a, x) = \frac{\pi(a \mid x)}{\pi_0(a \mid x)} \left\{ \phi_{\mathcal{Y}}(y) - \mu_{Y|A,X}(a, x) \right\} + \int \mu_{Y|A,X}(a', x)\pi(da' \mid x) - \chi(\pi). \quad (10)$$

Note that the EIF defined in Equation (10), similar to the EIF of the expected policy risk in OPE [33, 32], depends on both the propensity score $\pi_0(a|x)$ and the conditional mean embedding $\mu_{Y|A,X}$. Note that, since we consider stochastic interventions, our EIF remains valid for continuous treatments—unlike the setting in [28], which would require a kernel localization argument [62–64] to handle continuity. Estimating $\int \mu_{Y|A,X}(a', x)\pi(da' \mid x)$ corresponds to the plug-in estimation procedure described previously, while estimating the importance weighted term $\frac{\pi(a|x)}{\pi_0(a|x)}\phi_{\mathcal{Y}}(y)$ aligns with the CME estimator for kernel treatment effects analyzed by [17], who, however, assume known propensities $\pi_0(a|x)$. By contrast, our framework permits estimation of propensities $\hat{\pi}_0(a|x)$ with machine learning algorithms [40]. Leveraging the EIF, we define the following one-step doubly robust $\hat{\chi}_{dr}$ estimator:

$$\hat{\chi}_{dr}(\pi) = \frac{1}{n} \sum_{i=1}^{n} \left\{ \frac{\pi(a_i \mid x_i) \left( \phi_{\mathcal{Y}}(y_i) - \hat{\mu}_{Y|A,X}(a_i, x_i) \right)}{\hat{\pi}_0(a_i \mid x_i)} + \int \hat{\mu}_{Y|A,X}(a, x_i)\pi(da \mid x_i) \right\}. \quad (11)$$

Like all one-step estimators in OPE, our estimator enjoys a doubly robust property: it remains consistent if either $\hat{\pi}_0$ or $\hat{\mu}_{Y|X,A}$ is correctly specified. We elaborate on this property in Appendix 11.2. Note that originally, Luedtke and Chung [41] proposed a cross-fitted variant of the one-step estimator. In Appendix 11.4, we discuss this variant and show that, under a stochastic equicontinuity condition [65], cross-fitting may be discarded—thus improving statistical power. We now state a consistency result.

**Theorem 6.** *(Consistency of the doubly robust estimator). Suppose Assumptions 1, 3, 15, 16 and 17. Set $\lambda = n^{-1/(c+1/b)}$, which is rate optimal regularization. Then, with high probability,*

$$\|\hat{\chi}_{dr}(\pi) - \chi(\pi)\|_{\mathcal{H}_{\mathcal{Y}}} = \mathcal{O}\left[ n^{-1/2} + r_{\pi_0}(n).r_C(n, b, c) \right].$$

Here, $r_{\pi_0}(n)$ denotes the error in estimating the propensity score $\pi_0(a \mid x)$. The proof of this consistency result, along with explicit constants hidden by the $\mathcal{O}(\cdot)$ notation, is provided in Appendix 11.3. These rates approach $n^{-1/2}$ when the product $r_{\pi_0}(n, \delta) \cdot r_C(n, \delta, b, c)$ scales as $n^{-1/2}$—for instance, when both the conditional mean embedding and propensity score estimators converge at rate $n^{-1/4}$. This result constitutes a clear improvement over Theorem 5.

## 5 Testing and sampling from the counterfactual outcome distribution

In this section, we now discuss important applications of the proposed CPME framework.

### 5.1 Testing

CPME enables to assess differences in counterfactual outcome distributions $\nu(\pi)$ and $\nu(\pi')$. Such a difference in the two distributions can be formulated as a problem of hypothesis testing, or more specifically, two-sample testing [24]. Moreover, we want to perform that test while only being given acess to the logged data. The null hypothesis $H_0$ and the alternative hypothesis $H_1$ are thus defined as

$$H_0 : \nu(\pi) = \nu(\pi'), \quad H_1 : \nu(\pi) \neq \nu(\pi').$$

Specifically, we equivalently test $H_0 : \mathbb{E}_{P_\pi}[k(\cdot, y)] - \mathbb{E}_{P_{\pi'}}[k(\cdot, y)] = 0$ given the characteristic assumption on kernel $k_{\mathcal{Y}}$. Moreover, leveraging the EIF formulated in Section 4, we have:

$$\mathbb{E}_{P_\pi}[\phi_{\mathcal{Y}}(y)] - \mathbb{E}_{P_{\pi'}}[\phi_{\mathcal{Y}}(y)] = \mathbb{E}_{P_0}[\varphi_{\pi,\pi'}(y, a, x)], \quad (12)$$

where we take the difference of EIFs of $\chi(\pi)$ and $\chi(\pi')$:

$$\varphi_{\pi,\pi'}(y,a,x) = \left\{ \frac{\pi(a|x)}{\pi_0(a|x)} - \frac{\pi'(a|x)}{\pi_0(a|x)} \right\} \left\{ \phi_{\mathcal{Y}}(y) - \mu_{Y|A,X}(a,x) \right\} + \beta_\pi(x) - \beta_{\pi'}(x), \qquad (13)$$

and use the shorthand notation $\beta_\pi(x) = \int \mu_{Y|A,X}(a',x)\pi(da' \mid x)$. Thus, we can equivalently test for $H_0 : \mathbb{E}[\varphi_{\pi,\pi'}(y,a,x)] = 0$. With this goal in mind, and recalling that the MMD is a degenerated statistic [24], we define the following statistic using a cross U-statistic as Kim and Ramdas [66]:

$$T_{\pi,\pi'}^\dagger := \frac{\sqrt{m}\, \bar{f}_{\pi,\pi'}^\dagger}{S_{\pi,\pi'}^\dagger}, \text{ where } f_{\pi,\pi'}^\dagger(y_i,a_i,x_i) = \frac{1}{n-m} \sum_{j=m+1}^{n} \langle \hat{\varphi}_{\pi,\pi'}(y_i,a_i,x_i), \tilde{\varphi}_{\pi,\pi'}(y_j,a_j,x_j) \rangle.$$

Importantly, above, $m$ balances the two splits, $\hat{\varphi}_{\pi,\pi'}(y,a,x)$ is an estimate of $\varphi_{\pi,\pi'}(y,a,x)$ (using $\hat{\pi}_0$ and $\hat{\mu}_{Y|A,X}$) on the first $m$ samples while $\tilde{\varphi}$ is an estimate of the same quantity on the remaining $n-m$ samples. Further, $\bar{f}_{\pi,\pi'}^\dagger$ and $S_{\pi,\pi'}^\dagger$ denote the empirical mean and standard error of $f_{\pi,\pi'}^\dagger$ :

$$\bar{f}_{\pi,\pi'}^\dagger = \frac{1}{m} \sum_{i=1}^{m} f_{\pi,\pi'}^\dagger(y_i,a_i,x_i), \; S_{\pi,\pi'}^\dagger = \sqrt{\frac{1}{m} \sum_{i=1}^{m} \left( f_{\pi,\pi'}^\dagger(y_i,a_i,x_i) - \bar{f}_{\pi,\pi'}^\dagger \right)^2}. \qquad (14)$$

Having defined this cross U-statistic, we are now in position to prove the following asymptotic normality result, as in [28] for kernel treatment effects.

**Theorem 7.** *(Asymptotic normality of the test statistic) Suppose that the conditions of Theorem 6 hold, and that $\mathbb{E}_{P_0}\left[\|\varphi_{\pi,\pi'}(Y,A,X)\|^4\right] < \infty$. Assume the non-degeneracy condition $\mathbb{E}[\langle \varphi_{\pi,\pi'}(Z), \varphi_{\pi,\pi'}(Z')\rangle_{\mathcal{H}_Y}] > 0$, and that the product of nuisance convergence rates satisfies $r_{\pi_0}(n)\, r_C(n,b,c) = \mathcal{O}(n^{-1/2})$. Set $\lambda = n^{-1/(c+1/b)}$ and $m = \lfloor n/2 \rfloor$. Then it follows that*

$$T_{\pi,\pi'}^\dagger \xrightarrow{d} \mathcal{N}(0,1).$$

We provide a proof of this result in Appendix 12. Note here that while a Hilbert space CLT would allow to show asymptotic normality of the EIF of the CPME in the RKHS [41], using a cross-U statistic here is necessary due to the degeneracy of the MMD metric. Kim and Ramdas [66] show that $m = \lfloor n/2 \rfloor$ maximizes the power of the test. Moreover, the doubly robust estimator of the CPME allows to obtain a faster convergence rate which is instrumental for the asymptotic normality of the statistic. Based on the normal asymptotic behaviour of $T_{\pi,\pi'}^\dagger$, we propose as in [28] to test the null hypothesis $H_0 : \nu(\pi) = \nu(\pi')$ given the p-value $p = 1 - \Phi(T_{\pi,\pi'}^\dagger)$, where $\Phi$ is the CDF of a standard normal. For an $\alpha$-level test, the test rejects the null if $p \leq \alpha$. Algorithm 1 below illustrates the full procedure of the test, which we call DR-KPT (Doubly Robust Kernel Policy Test).

---

**Algorithm 1** DR-KPT

---

**Require:** Data $\mathcal{D} = (x_i,a_i,y_i)_{i=1}^{n}$, kernels $k_{\mathcal{Y}}, k_{\mathcal{A},\mathcal{X}}$
**Ensure:** The p-value of the test
  1: Set $m = \lfloor n/2 \rfloor$ and estimate $\hat{\mu}_{Y|A,X}, \hat{\pi}_0$ on first $m$ samples, $\tilde{\mu}_{Y|A,X}, \tilde{\pi}_0$ on remaining $n-m$.
  2: Define $\hat{\varphi}(y,a,x) = \left\{ \frac{\pi(a|x)}{\hat{\pi}_0(a|x)} - \frac{\pi'(a|x)}{\hat{\pi}_0(a|x)} \right\} \left\{ \phi_{\mathcal{Y}}(y) - \hat{\mu}_{Y|A,X}(a,x) \right\} + \hat{\beta}_\pi(x) - \hat{\beta}_{\pi'}(x)$ and $\tilde{\varphi}$.
  3: Define $f_{\pi,\pi'}^\dagger(y_i,a_i,x_i) = \frac{1}{n-m} \sum_{j=m+1}^{n} \langle \hat{\varphi}(y_i,a_i,x_i), \tilde{\varphi}(y_j,a_j,x_j)\rangle$ for $i = 1,\dots,m$
  4: Calculate $\bar{f}_{\pi,\pi'}^\dagger$ and $S_{\pi,\pi'}^\dagger$ using Equation (14), then $T_{\pi,\pi'}^\dagger = \frac{\sqrt{m}\bar{f}_{\pi,\pi'}^\dagger}{S_{\pi,\pi'}^\dagger}$
  5: **return** p-value $p = 1 - \Phi(T_{\pi,\pi'}^\dagger)$

---

Note that as Martinez Taboada et al. [28], our test is computationally efficient compared to the permutation tests required in CME Muandet et al. [17], which would require the dramatic fitting of the plug-in estimator for each iterations.

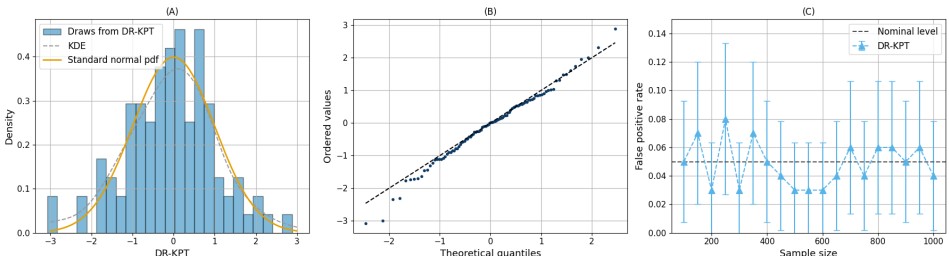

Figure 1: Illustration of 100 simulations of DR-KPT under the null: (A) Histogram with standard normal pdf for $n = 400$, (B) Normal Q–Q plot for $n = 400$, (C) False positive rate across sample sizes. The results confirm the Gaussian behavior and good calibration of the test under the null.

## 5.2 Sampling

We now present a deterministic procedure that uses the estimated distribution embeddings $\hat{\chi}(\pi)$ to provide samples $(\tilde{y}_j)$ from the counterfactual outcome distribution. The procedure is a variant of kernel herding [25, 17] and is given in Algorithm 2.

---

**Algorithm 2** Sampling from the counterfactual distribution

---

**Require:** Estimated CPME $\hat{\chi}(\pi) : \mathcal{Y} \to \mathbb{R}$, kernel $k_{\mathcal{Y}} : \mathcal{Y} \times \mathcal{Y} \to \mathbb{R}$, and number of samples $m \in \mathbb{N}$
 1: $\tilde{y}_1 := \arg\max_{y \in \mathcal{Y}} \hat{\chi}(\pi)(y)$
 2: **for** $t = 2$ to $m$ **do**
 3: $\quad \tilde{y}_t := \arg\max_{y \in \mathcal{Y}} \left[ \hat{\chi}(\pi)(y) - \frac{1}{t} \sum_{\ell=1}^{t-1} k_{\mathcal{Y}}(\tilde{y}_\ell, y) \right]$
 4: **end for**
 5: **Output:** $\tilde{y}_1, \ldots, \tilde{y}_m$

---

Below, we prove that these samples converge in distribution to the counterfactual distribution. We state an additional regularity condition under which we can prove that the empirical distribution $\tilde{P}_Y^m$ of the herded samples $(\tilde{y}_j)_{j=1}^m$, calculated from the distribution embeddings, weakly converges to the desired distribution.

**Assumption 8.** *(Additional regularity). Assume i) $\mathcal{Y}$ is locally compact. ii) $\mathcal{H}_y \subset \mathcal{C}_0$, where $\mathcal{C}_0$ is the space of bounded, continuous, real valued functions that vanish at infinity.*

As discussed by Simon-Gabriel et al. [67], the combined assumptions that $\mathcal{Y}$ is Polish and locally compact impose weak restrictions. In particular, if $\mathcal{Y}$ is a Banach space, then to satisfy both conditions it must be finite dimensional. Trivially, $\mathcal{Y} = \mathbb{R}^{\dim(Y)}$ satisfies both conditions.

**Proposition 9.** *(Convergence of MMD of herded samples, weak convergence to the counterfactual outcome distribution) Suppose the conditions of Lemma 4.1 and Assumption 8 hold. Let $(\tilde{y}_{dr,j})$ and $\tilde{P}_{Y,dr}^m$ (resp. $(\tilde{y}_{pi,j})$, $\tilde{P}_{Y,pi}^m$) be generated from $\hat{\chi}_{dr}(\pi)$ (resp. $\hat{\chi}_{pi}(\pi)$) via Algorithm 2. Then, with high probability, $\mathrm{MMD}(\tilde{P}_{Y,pi}^m, \nu(\pi)) = O_p(r_C(n, b, c) + m^{-1/2})$ and $\mathrm{MMD}(\tilde{P}_{Y,dr}^m, \nu(\pi)) = O_p(n^{-1/2} + r_{\pi_0}(n) r_C(n, b, c) + m^{-1/2})$. Moreover, $(\tilde{y}_{dr,j}) \rightsquigarrow \nu(\pi)$ and $(\tilde{y}_{\pi,j}) \rightsquigarrow \nu(\pi)$.*

The proof is provided in Appendix 13. This proposition shows that the DR estimator of CPME yields an empirical outcome distribution with improved MMD convergence, with weak convergence toward the counterfactual outcome distribution [67].

## 6 Numerical experiments

In this section, we present numerical simulations for testing and sampling from the counterfactual distributions. Full experimental details, including additional simulations, are provided in Appendix 14. All code and simulation materials used in this study are publicly available at `https://github.com/houssamzenati/counterfactual-policy-mean-embedding`.

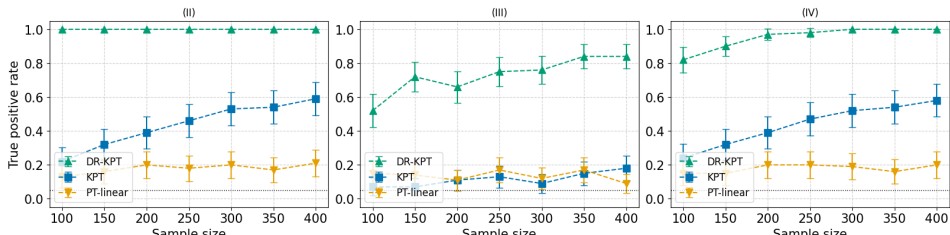

Figure 2: True positive rates of 100 simulations of the tests in Scenarios II, III, and IV. DR-KPT shows notable true positive rates in every scenario, unlike competitors.

## 6.1 Testing

We assess the empirical calibration and power of the proposed DR-KPT test in the standard observational causal inference framework. We assume access to i.i.d. samples $\{(x_i, a_i, y_i)\}_{i=1}^n \sim (X, A, Y)$. All hypothesis tests are conducted at a significance level of 0.05.

**Synthetic experiments** We synthetically generate covariates, continuous treatments, and outcomes under four scenarios adapted from [17, 28]. Scenario I (Null): $\pi = \pi'$, implying no distributional shift and $\nu(\pi) = \nu(\pi')$. Scenario II (Mean Shift): $\pi$ and $\pi'$ differ by small opposite shifts in their mean treatment assignments, changing the expected mean. Scenario III (Mixture): $\pi'$ is a stochastic mixture of two policies with the same mean as $\pi$, creating a bimodal treatment distribution that alters outcomes without affecting the mean. Scenario IV (Shifted Mixture): same as Scenario III but with an additional mean shift of $\pi'$ relative to $\pi$.

In all cases, treatments are drawn from a logging policy $\pi_0$, while outcome and propensity models are unknown. Propensities $\hat{\pi}_0(\cdot)$ are estimated via linear regression, and outcome regressions via conditional mean embeddings. We first assess the empirical calibration of DR-KPT and the Gaussian behavior of $T_{\pi,\pi'}^\dagger$ under the null. Figure 1 shows that DR-KPT achieves near-standard normal behavior and proper calibration in Scenario I. Figure 2 reports results for Scenarios II–IV. As baselines, we adapt the KTE method of Muandet et al. [17] into a Kernel Policy Test (KPT) with estimated propensities and include a linear-kernel variant (PT-linear) testing only mean shifts. DR-KPT consistently outperforms all methods, including under pure mean shifts, where KPT and PT-linear degrade due to propensity estimation. Overall, DR-KPT reliably detects distributional changes, exhibits strong power across scenarios, and remains computationally efficient (see Appendix 14).

**Warfarin dataset** We use the publicly available dataset on Warfarin dosage [68], which contains patient covariates and expert-prescribed therapeutic doses. The treatment corresponds to a continuous dosage level, making this dataset well suited for off-policy evaluation of continuous treatment policies. Although the data are fully supervised, we simulate an off-policy bandit environment (see Appendix 14) by defining a reward function that is maximal when the assigned dose $a$ lies within $\pm 10\%$ of the expert's prescription, following Kallus and Zhou [4], Zenati et al. [69]; logging and target policies are modeled as Gaussian distributions.

We mirror the synthetic testing protocol of the previous experiment and evaluate four scenarios—(I) Null, (II) Mean Shift, (III) Mixture, and (IV) Shifted Mixture—each introducing distinct shifts in the treatment and outcome distributions. Both outcome models and propensity scores are learned from data. We compare our Doubly Robust Kernel Policy Test (DR-KPT) with baseline KPT estimators using linear, RBF, and polynomial kernels. The results in Table 1 show that DR-KPT is well-calibrated under the null (Scenario I) with near-nominal rejection rates. Across all alternative scenarios (II–IV), DR-KPT consistently outperforms or matches the best baseline.

**dSprites (Structured Outcomes).** We perform experiments on the dSprites dataset [70, 71], which enables evaluation on structured image outcomes. Unlike scalar outcomes in our other experiments, here the counterfactual effect of a policy is evaluated on rendered $64 \times 64$ images generated from latent variables. The structural causal model is defined by latent contexts $x \sim \mathcal{U}([0,1]^2)$, actions $a \sim \pi(\cdot \mid x)$, and outcomes $y := g(x, a) \in \mathbb{R}^{64 \times 64}$, where $g$ maps each context–action pair to an image via the fixed dSprites renderer. All other latent factors (shape, scale, and orientation) are held

Table 1: Rejection rates for the Warfarin dataset across four scenarios.

| Scenario | KPT-linear | KPT-rbf | KPT-poly | DR-KPT-rbf | DR-KPT-poly |
|----------|------------|---------|----------|------------|-------------|
| I        | 0.00       | 0.00    | 0.00     | **0.02**   | **0.06**    |
| II       | 0.77       | 0.01    | 0.29     | **0.80**   | 0.66        |
| III      | **1.00**   | 0.00    | 0.66     | **0.99**   | 0.95        |
| IV       | 0.24       | 0.00    | 0.11     | **0.76**   | 0.55        |

constant. As in previous experiments, the logging and target policies $\pi, \pi'$ are contextual Gaussians $\mathcal{N}(\mu(U), \sigma^2 I)$, where $\mu(U)$ encodes a rotated and shifted transformation of the context. We focus on two scenarios: (I) *Null*, where outcome distributions coincide, and (IV) *Shifted Mixture*, where they differ due to policy-induced shifts. Both outcome models and propensity scores are learned from data, and the evaluation follows the same procedure as in the Warfarin experiment.

Table 2: Rejection rates for the dSprites dataset under structured outcomes.

| Scenario | KPT-linear | KPT-rbf | KPT-poly | DR-KPT-rbf | DR-KPT-poly |
|----------|------------|---------|----------|------------|-------------|
| I        | 0.394      | 0.401   | 0.375    | **0.024**  | 0.000       |
| IV       | 0.081      | 0.054   | 0.073    | **0.656**  | 0.502       |

The results highlight the poor calibration of baseline methods under the null (Scenario I), with inflated rejection rates approaching $40\%$, while DR-KPT maintains near-nominal levels. In the alternative scenario (IV), DR-KPT achieves substantially higher power than all baselines, confirming its robustness and sensitivity in detecting structured distributional shifts with complex outcomes.

## 6.2 Sampling

We also perform an experiment in which we generate samples from Algorithm 2 with both the plug-in and DR estimators of the CPME under multiple scenarios in which we vary the design of the logging policy (uniform and logistic) and the outcome function (quadratic and sinusoidal) - see Appendix 14. In Figure 3 we illustrate an example of the outcome distribution from logged samples, the oracle counterfactual outcome distribution and the empirical distribution obtained from two kernel herding algorithms. Appendix 14 reports MMD and Wasserstein distances between the counterfactual and oracle distributions, illustrating that the DR variant generally attains lower distances in our synthetic setting.

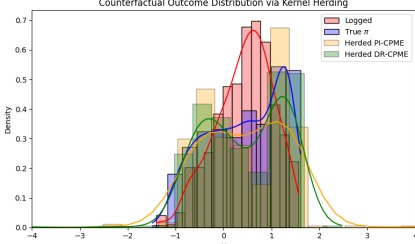

Figure 3: Logistic logging policy, nonlinear outcome function.

## 7 Discussion

In this paper, we presented a method for estimating the Counterfactual Policy Mean Embedding (CPME), the outcome distribution mean embedding of counterfactual policies. We proposed a nonparametric plug-in estimator together with a doubly robust, efficient influence function-based variant enabling a computationally efficient kernel test. Our framework also supports sampling from counterfactual outcome distributions. Recent advances suggest more scalable extensions based on MMD gradient flows [72, 73], which we view as a promising direction for future work. Finally, our analysis relies on standard identification assumptions such as positivity and exchangeability; relaxing these toward weaker or partially identifiable settings is another important avenue for future research. settings is an important direction for future work.

## Acknowledgments and Disclosure of Funding

The authors thank Dimitri Meunier for his valuable discussions and insightful comments. The authors also thank Liyuan Xu for early discussions. All authors acknowledge support from the Gatsby Charitable Foundation. The authors are grateful for the constructive feedback and insightful discussion provided by the anonymous reviewers of NeurIPS 2025.

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

# Appendix

This appendix is organized as follows:

All the code to reproduce our numerical simulations is provided in the supplementary material and will be open-sourced upon acceptance of the manuscript.

# 8 Notations

In this appendix, we recall for clarity some useful notations that are used throughout the paper.

**Notations for distributional off-policy evaluation setting and finite samples**

- $y_i, a_i, x_i$ are realizations of the outcome, action, and context random variables $Y, A, X$ for $i \in \{1, \dots n\}$. Potential outcomes are written $\{Y(a)\}_{a \in \mathcal{A}}$.
- The distribution on the context space is written $P_X$, the distribution on outcomes is conditional to actions and contexts and is written $P_{Y|X,A}$. Distributions on actions $A$ are policies $\pi$ belonging to a set $\Pi$. In the logged dataset, actions are drawn from a logging policy $\pi_0$. Resulting triplet distribution is written $P_\pi = P_{Y|X,A} \times \pi \times P_X$.
- The distribution $\nu(\pi)$ represents the marginal distribution of outcomes over $\pi \times P_X$.

**Notations related to the kernel-based representations used to embed counterfactual outcome distributions**

- $\mathcal{H}_\mathcal{F}$ is a generic RKHS associated with a domain $\mathcal{F}$.
- $\mathcal{H}_{\mathcal{A}\mathcal{X}}$: RKHS on $\mathcal{A} \times \mathcal{X}$ with kernel $k_{\mathcal{A}\mathcal{X}}$ and feature map $\phi_{\mathcal{A}\mathcal{X}}(a, x) = k_{\mathcal{A}\mathcal{X}}(\cdot, (a, x))$. Inner product: $\langle \cdot, \cdot \rangle_{\mathcal{H}_{\mathcal{A}\mathcal{X}}}$.
- $\mathcal{H}_\mathcal{Y}$: RKHS on $\mathcal{Y}$ with kernel $k_\mathcal{Y}$ and feature map $\phi_\mathcal{Y}(y) = k_\mathcal{Y}(\cdot, y)$. Inner product: $\langle \cdot, \cdot \rangle_{\mathcal{H}_\mathcal{Y}}$.
- Given a distribution $P$ over $\mathcal{F}$, the kernel mean embedding is $\mu_F = \mathbb{E}_P[\phi_\mathcal{F}(F)] \in \mathcal{H}_\mathcal{F}$.
- For conditional $P_{F|G}$, the conditional mean embedding is $\mu_{F|G}(g) = \mathbb{E}[\phi_\mathcal{F}(F) \mid G = g] \in \mathcal{H}_\mathcal{F}$.
- The counterfactual policy mean embedding (CPME): $\chi(\pi) = \mathbb{E}_{P_\pi}[\phi_\mathcal{Y}(Y(a))]$.
- $\kappa_{ax}, \kappa_y$: bounds on kernels: $\sup_{a,x} \|\phi_{\mathcal{A}\mathcal{X}}(a, x)\|_{\mathcal{H}_{\mathcal{A}\mathcal{X}}} \leq \kappa_{a,x}$, $\sup_y \|\phi_\mathcal{Y}(y)\|_{\mathcal{H}_\mathcal{Y}} \leq \kappa_y$
- $S_2(\mathcal{H}_{\mathcal{A}\mathcal{X}}, \mathcal{H}_\mathcal{Y})$ denotes the Hilbert space of the Hilbert-Schmidt operators from $\mathcal{H}_{\mathcal{A}\mathcal{X}}$ to $\mathcal{H}_\mathcal{Y}$.
- $\mathcal{C}_{Y|A,X} \in S_2(\mathcal{H}_{\mathcal{A}\mathcal{X}}, \mathcal{H}_\mathcal{Y})$ is the conditional mean operator.
- $\mu_\pi$: the kernel policy embedding in $\mathcal{H}_{\mathcal{A}\mathcal{X}}$.
- $c, b$: source condition and spectral decay parameters.
- $\lambda$: regularization parameter for learning $\mathcal{C}_{Y|A,X}$.
- $L$: Kernel integral operator $Lh := \int k(\cdot, w)h(w) \, d\rho(w)$, mapping $L^2(\rho) \to L^2(\rho)$.
- $\{\eta_j\}_{j \geq 1}$: Eigenvalues of $L$, ordered decreasingly, assumed to satisfy a spectral decay assumption $\eta_j \leq Cj^{-b}$.
- $\{\varphi_j\}_{j \geq 1}$: Orthonormal eigenfunctions of $L$ in $L^2(\rho)$, satisfying $L\varphi_j = \eta_j \varphi_j$.
- $\mathcal{H}^c$: Interpolation space of order $c$, defined as $\mathcal{H}^c := \left\{ f = \sum_j h_j \varphi_j \mid \sum_j h_j^2/\eta_j^c < \infty \right\}$.

**Notations related to estimators and asymptotic analysis**

- $\hat{\chi}_{pi}(\pi)$ and $\hat{\chi}_{dr}(\pi)$: plug-in and doubly robust estimators of CPME.
- $\hat{\mu}_{Y|A,X}(a, x)$: estimator of the conditional mean embedding.

- $\hat{\pi}_0(a|x)$: estimator of the logging policy.
- $r_C(n, b, c)$: convergence rate of $\hat{\mathcal{C}}_{Y|A,X}$.
- $r_{\pi_0}(n)$: convergence rate of $\hat{\pi}_0$.
- $o_p(1), O_p(1)$: standard probabilistic asymptotic notations.

## Notations for differentiability and statistical models

- $\mathcal{P}$: statistical model on $\mathcal{Z} = \mathcal{Y} \times \mathcal{A} \times \mathcal{X}$.
- $L^2(\rho)$: space of square-integrable real-valued functions w.r.t. measure $\rho$.
- $L^2(P; \mathcal{H})$: Bochner space of $\mathcal{H}$-valued functions with norm $\|f\|_{L^2(P;\mathcal{H})} = \left( \int \|f(z)\|_{\mathcal{H}}^2 \, dP(z) \right)^{1/2}$.
- $\Pi_{\mathcal{H}}[h \mid \mathcal{W}]$: orthogonal projection of $h$ onto closed subspace $\mathcal{W} \subset \mathcal{H}$.
- $\mathcal{P}_{\pi_0}$: submodel of $\mathcal{P}$ with fixed treatment policy $\pi_0$.
- $\dot{\mathcal{P}}_P$: tangent space at $P$.
- $\mathscr{P}(P, \mathcal{P}, s)$: smooth submodels of $\mathcal{P}$ at $P$ with score $s$.
- $s, s_X, s_{Y|A,X}, s_{A|X}$: score functions.
- $\chi(\pi)(P)$: value of the CPME at $P$.
- $\dot{\chi}_P^{\pi}$: local parameter of $\chi(\pi)$ at $P$.
- $\dot{\chi}_P^{\pi,*}$: adjoint (efficient influence operator).
- $\dot{\mathcal{H}}_P$: image of $\dot{\chi}_P^{\pi,*}$.

## Notations for efficient influence functions

- $\psi_P^{\pi}$: efficient influence function (EIF) at $P$.
- $\tilde{\psi}_P^{\pi}$: candidate EIF: $\tilde{\psi}_P^{\pi}(y, a, x) = \dot{\chi}_P^{\pi,*}(\phi_{\mathcal{Y}})(y, a, x)$.

## Error decomposition of the one-step estimator

- $P_n$: empirical distribution of the sample $\{z_i\}_{i=1}^n$.
- $\hat{P}_n$: estimated distribution using nuisance estimators.
- $\mathcal{S}_n = (P_n - P)\psi^{\pi}$: empirical average term.
- $\mathcal{T}_n = (P_n - P)(\hat{\psi}_n^{\pi} - \psi^{\pi})$: empirical process term.
- $\mathcal{R}_n = \chi(\hat{P}_n) + P\hat{\psi}_n^{\pi} - \chi(\pi)$: remainder term.

## Notations for empirical processes and equicontinuity

- $\mathcal{T}_n(\varphi) := \sqrt{n}(P_n - P)(\varphi)$: empirical process acting on $\varphi$.
- $\mathcal{G}$: class of $\mathcal{H}_{\mathcal{Y}}$-valued functions (e.g. $\hat{\psi}_n^{\pi} - \psi^{\pi}$).

## Notations for hypothesis testing

- $H_0$: null hypothesis — $\nu(\pi) = \nu(\pi')$.
- $H_1$: alternative hypothesis — $\nu(\pi) \neq \nu(\pi')$.
- $\varphi_{\pi,\pi'}$: difference of EIFs for policies $\pi$ and $\pi'$.
- $\hat{\varphi}_{\pi,\pi'}, \tilde{\varphi}_{\pi,\pi'}$: estimates of $\varphi_{\pi,\pi'}$ over disjoint subsets.
- $\hat{\beta}_{\pi}(x) := \int \hat{\mu}_{Y|A,X}(a, x)\pi(da \mid x)$: estimated conditional policy mean.
- $f_{\pi,\pi'}^{\dagger}(y, a, x)$: cross-U-statistic kernel.
- $\bar{f}_{\pi,\pi'}^{\dagger}, S_{\pi,\pi'}^{\dagger}$: empirical mean and std of $f^{\dagger}$.
- $T_{\pi,\pi'}^{\dagger}$: normalized test statistic.
- $\mathbb{H}$: limiting Gaussian process in $\mathcal{H}_{\mathcal{Y}}$.
- $\langle \mathbb{H}, h \rangle_{\mathcal{H}_{\mathcal{Y}}}$: projection onto direction $h$.
- $\Phi$: CDF of standard normal.
- $p = 1 - \Phi(T_{\pi,\pi'}^{\dagger})$: p-value.

## Notations for sampling from counterfactual distributions

- $(\tilde{y}_j)_{j=1}^m$: deterministic samples generated via kernel herding.
- $\tilde{P}_Y^m$: empirical distribution over the $\tilde{y}_j$.
- $\tilde{P}_{Y,dr}^m, \tilde{P}_{Y,pi}^m$: empirical distributions generated from $\hat{\chi}_{dr}(\pi)$ and $\hat{\chi}_{pi}(\pi)$.

# 9 Review of Counterfactual Mean Embeddings

In this appendix, we provide a background section on counterfactual mean embeddings [17] and distributional treatment effects.

## 9.1 Reproducing kernel hilbert spaces and kernel mean embeddings

A scalar-valued RKHS $\mathcal{H}_{\mathcal{W}}$ is a Hilbert space of functions $h : \mathcal{W} \to \mathbb{R}$. The RKHS is fully characterized by its feature map, which takes a point $w$ in the original space $\mathcal{W}$ and maps it to a feature $\phi_{\mathcal{W}}(w)$ in RKHS $\mathcal{H}_{\mathcal{W}}$. The closure of $\mathrm{span}\{\phi_{\mathcal{W}}(w)\}_{w \in \mathcal{W}}$ is RKHS $\mathcal{H}_{\mathcal{W}}$. In other words, $\{\phi_{\mathcal{W}}(w)\}_{w \in \mathcal{W}}$ can be viewed as the dictionary of basis functions for RKHS $\mathcal{H}_{\mathcal{W}}$. The kernel $k_{\mathcal{W}} : \mathcal{W} \times \mathcal{W} \to \mathbb{R}$ is the inner product of features $\phi_{\mathcal{W}}(w)$ and $\phi_{\mathcal{W}}(w')$.

$$k_{\mathcal{W}}(w, w') = \langle \phi_{\mathcal{W}}(w), \phi_{\mathcal{W}}(w') \rangle_{\mathcal{H}_{\mathcal{W}}}. \tag{15}$$

A real-valued kernel $k$ is continuous, symmetric and positive definite. The essential property of a function $h$ in an RKHS $\mathcal{H}_{\mathcal{W}}$ is the eponymous reproducing property:

$$h(w) = \langle h, \phi_{\mathcal{W}}(w) \rangle_{\mathcal{H}_{\mathcal{W}}} \tag{16}$$

In other words, to evaluate $h$ at $w$, we take the RKHS inner product between $h$ and the features $\phi_{\mathcal{W}}(w)$ for $\mathcal{H}_{\mathcal{W}}$. The reproducing property, importantly, allows to separate function $h$ from features $\phi_{\mathcal{W}}(w)$ and thereby decouple the steps of nonparametric causal estimation. Notably, the RKHS is a practical hypothesis space for nonparametric regression.

**Example 9.1.** *(Nonparametric regression) Consider the output $y \in \mathbb{R}$, the input $w \in \mathcal{W}$ and the goal of estimating the conditional expectation function $h(w) = \mathbb{E}(Y \mid W = w)$. A kernel ridge regression estimator of $h$ is*

$$\hat{h} = \operatorname*{arg\,min}_{h \in \mathcal{H}} \frac{1}{n} \sum_{i=1}^{n} \{ y_i - \langle h, \phi_{\mathcal{W}}(w_i) \rangle_{\mathcal{H}} \}^2 + \lambda \|h\|_{\mathcal{H}}^2, \tag{17}$$

*where $\lambda > 0$ is a hyperparameter on the ridge penalty $\|h\|_{\mathcal{H}}^2$, which imposes smoothness in estimation. The solution to the optimization problem has a well-known closed form:*

$$\hat{h}(w) = Y^{\mathrm{T}} (K_{WW} + n\lambda I)^{-1} K_{Ww}. \tag{18}$$

*The closed-form solution involves the kernel matrix $K_{WW} \in \mathbb{R}^{n \times n}$ with $(i, j)$ th entry $k_{\mathcal{W}}(w_i, w_j)$, and the kernel vector $K_{Ww} \in \mathbb{R}^n$ with $i$ th entry $k_{\mathcal{W}}(w_i, w)$.*

In this work, we use kernels and RKHSs to represent, compare, and estimate probability distributions. This is enabled by the approach known as kernel mean embedding (KME) of distributions [21], which we briefly review here. Let $\mathcal{H}_{\mathcal{W}}$ be a RKHS with kernel $k_{\mathcal{W}}$ defined on a space $\mathcal{W}$, and assume that $\sup_{w \in \mathcal{W}} k_{\mathcal{W}}(w, w) < \infty$. Then, for a probability distribution $P$ over $\mathcal{W}$, the kernel mean embedding is defined as the Bochner integral[3]:

$$\mu : \mathcal{P} \to \mathcal{H}_{\mathcal{W}}, \quad P \mapsto \mu_P := \int k_{\mathcal{W}}(\cdot, w) \, dP(w).$$

The embedded element $\mu_P$, also written $\mu_W$ when $W \sim P$, serves as a representation of $P$ in $\mathcal{H}_{\mathcal{W}}$. If $\mathcal{H}_{\mathcal{W}}$ is characteristic [76, 23, 77], this mapping is injective: $\mu_P = \mu_Q$ if and only if $P = Q$. Thus, $\mu_P$ uniquely identifies $P$, preserving all distributional information. Common examples of characteristic kernels on $\mathbb{R}^d$ include Gaussian, Matérn, and Laplace kernels [23, 77], while linear and polynomial kernels are not characteristic due to their finite-dimensional RKHSs.

The kernel mean embedding induces a popular distance between probability measures known as the maximum mean discrepancy (MMD) [78, 79, 24]. For distributions $P$ and $Q$, it is defined by:

$$\mathrm{MMD}[\mathcal{H}_{\mathcal{W}}, P, Q] := \|\mu_P - \mu_Q\|_{\mathcal{H}_{\mathcal{W}}} = \sup_{h \in \mathcal{H}_{\mathcal{W}}, \|h\| \leq 1} \left| \int h \, dP - \int h \, dQ \right|.$$

---

[3]See, e.g., [74, Chapter 2] and [75, Chapter 1] for the definition of the Bochner integral.

The second equality follows from the reproducing property and the structure of RKHSs as vector spaces [24, Lemma 4]. If $\mathcal{H}_\mathcal{W}$ is characteristic, then $\mathrm{MMD}[\mathcal{H}_\mathcal{W}, P, Q] = 0$ implies $P = Q$, so MMD defines a proper metric on distributions.

Given an i.i.d. sample $\{w_i\}_{i=1}^n$ from $P$, the kernel mean embedding can be estimated via the empirical average:

$$\hat{\mu}_P := \frac{1}{n} \sum_{i=1}^n k_\mathcal{W}(\cdot, w_i).$$

This estimator is $\sqrt{n}$-consistent: $\|\mu_P - \hat{\mu}_P\|_{\mathcal{H}_\mathcal{W}} = O_p(n^{-1/2})$ under mild assumptions [24, 80, 81].

Given a second i.i.d. sample $\{w_j'\}_{j=1}^m$ from $Q$, the squared empirical MMD is

$$\widehat{\mathrm{MMD}}^2[\mathcal{H}_\mathcal{W}, P, Q] = \|\hat{\mu}_P - \hat{\mu}_Q\|_{\mathcal{H}_\mathcal{W}}^2$$
$$= \frac{1}{n^2} \sum_{i,j=1}^n k_\mathcal{W}(w_i, w_j) - \frac{2}{nm} \sum_{i,j=1}^{n,m} k_\mathcal{W}(w_i, w_j') + \frac{1}{m^2} \sum_{i,j=1}^m k_\mathcal{W}(w_i', w_j').$$

This estimator is consistent and converges at the parametric rate $O_p(n^{-1/2} + m^{-1/2})$. It is biased but simple to compute; an unbiased version is also available [24, Eq. 3].

The KME framework extends naturally to conditional distributions [47, 82, 52, 83]. Let $(W, V)$ be a random variable on $\mathcal{W} \times \mathcal{V}$ with joint distribution $P_{WV}$. Using kernels $k_\mathcal{W}$ and $k_\mathcal{V}$ with RKHSs $\mathcal{H}_\mathcal{W}, \mathcal{H}_\mathcal{V}$, the conditional mean embedding of $P_{V|W=w}$ is defined as:

$$\mu_{V|W=w} := \int k_\mathcal{V}(\cdot, v) \, dP(v|w) \in \mathcal{H}_\mathcal{V}.$$

This representation preserves all information if $\mathcal{H}_\mathcal{V}$ is characteristic. Given a sample $\{(w_i, v_i)\}_{i=1}^n$, the conditional embedding can be estimated as

$$\hat{\mu}_{V|W=w} := \sum_{i=1}^n \beta_i(w) k_\mathcal{V}(\cdot, v_i),$$

with weights

$$\boldsymbol{\beta}(w) = (K + n\lambda I)^{-1} k_\mathcal{W}(w), \quad k_\mathcal{W}(w) = (k_\mathcal{W}(w, w_1), \ldots, k_\mathcal{W}(w, w_n))^\top.$$

Here, $K$ is the $n \times n$ kernel matrix with entries $K_{ij} = k_\mathcal{W}(w_i, w_j)$, and $\lambda > 0$ is a regularization parameter. This estimator corresponds to kernel ridge regression from $\mathcal{W}$ into $\mathcal{H}_\mathcal{V}$, where the target functions are feature maps $k_\mathcal{V}(\cdot, v_i)$. To guarantee convergence, $\lambda$ must decay appropriately as $n \to \infty$ [48, 51].

Finally, we make use of the Hilbert space $S_2(\mathcal{H}_{\mathcal{W}_1}, \mathcal{H}_{\mathcal{W}_2})$ of Hilbert-Schmidt operators between RKHSs. The conditional expectation operator $C : \mathcal{H}_{\mathcal{W}_1} \to \mathcal{H}_{\mathcal{W}_2}$ given by $h(\cdot) \mapsto \mathbb{E}[h(W_1)|W_2 = \cdot]$ is assumed to lie in $S_2(\mathcal{H}_{\mathcal{W}_1}, \mathcal{H}_{\mathcal{W}_2})$ and is estimated via ridge regression, by regressing $\phi_{\mathcal{W}_1}(W_1)$ on $\phi_{\mathcal{W}_2}(W_2)$ in $\mathcal{H}_{\mathcal{W}_2}$.

## 9.2 Assumptions for consistency

To prove consistency of our estimator, we rely on two standard approximation assumptions from RKHS learning theory: *smoothness* of the target function and *spectral decay* of the kernel operator. These are naturally formulated through the eigendecomposition of an associated integral operator, which we introduce below. The results may be found in [84].

**Kernel smoothing operator**   Let $\mathcal{H}_\mathcal{W}$ be a reproducing kernel Hilbert space (RKHS) over a space $\mathcal{W}$, with reproducing kernel with kernel $k_\mathcal{W} : \mathcal{W} \times \mathcal{W} \to \mathbb{R}$ consisting of functions of the form $h : \mathcal{W} \to \mathbb{R}$. Let $\rho$ be any Borel measure on $\mathcal{W}$. Let $L^2(\rho)$ be the space of square integrable functions with respect to measure $\rho$. We define the integral operator $L$ associated with the kernel $k_\mathcal{W}$ and the measure $\rho$ as:

$$L : L^2(\rho) \to L^2(\rho), h \mapsto \int k_\mathcal{W}(\cdot, w) h(w) \mathrm{d}\rho(w) \tag{19}$$

Intuitively, this operator smooths a function $h$ by averaging it with respect to the kernel $k_{\mathcal{W}}$ and the distribution $\rho$.

**Remark 10.** *(L as convolution). If the kernel $k_{\mathcal{W}}$ is defined on $\mathcal{W} \subset \mathbb{R}^d$ and shift invariant, then $L$ is a convolution of $k_{\mathcal{W}}$ and $h$. If $k_{\mathcal{W}}$ is smooth, then $Lh$ is a smoothed version of $h$.*

**Spectral properties of the kernel smoothing operator** The operator $L$ is *compact*, *self-adjoint*, and *positive semi-definite*. Therefore, by the *spectral theorem*, $L$ admits an orthonormal basis of eigenfunctions $(\varphi_j)_\rho$ in $\mathbb{L}^2_\rho(\mathcal{W})$, with corresponding non-negative eigenvalues $(\eta_j)$.

**Assumption 11.** *(Nonzero eigenvalues). For simplicity, we assume $(\eta_j) > 0$ in this discussion; see [85, Remark 3] for the more general case.*

Thus, for any $h \in L^2(\rho)$, we can write:

$$Lh = \sum_{j=1}^{\infty} \eta_j \langle \varphi_j, h \rangle_{\mathbb{L}^2_\rho} \varphi_j,$$

where each $\varphi_j$ is defined up to $\rho$-almost-everywhere equivalence.

**Feature map representation** The following observations help to interpret this eigendecomposition.

**Theorem 12.** *[86, Corollary 3.5] (Mercer's Theorem). The kernel $k_{\mathcal{W}}$ can be expressed as $k_{\mathcal{W}}(w, w') = \sum_{j=1}^{\infty} \eta_j \varphi_j(w) \varphi_j(w')$, where $(w, w')$ are in the support of $\rho$, $\varphi_j$ is a continuous element in the equivalence class $(\varphi_j)_\rho$, and the convergence is absolute and uniform.*

Since the kernel $k_{\mathcal{W}}$ can be decomposed as:

$$k_{\mathcal{W}}(w, w') = \sum_{j=1}^{\infty} \eta_j \varphi_j(w) \varphi_j(w'),$$

with absolute and uniform convergence on compact subsets of the support of $\rho$, we can express the *feature map $\phi_{\mathcal{W}}(w)$* associated with the RKHS as:

$$\phi_{\mathcal{W}}(w) = \left( \sqrt{\eta_1} \varphi_1(w), \sqrt{\eta_2} \varphi_2(w), \dots \right).$$

Thus, the inner product $\langle \phi_{\mathcal{W}}(w), \phi_{\mathcal{W}}(w') \rangle_{\mathcal{H}_{\mathcal{W}}}$ reproduces the kernel value $k_{\mathcal{W}}(w, w')$.

Both $L^2(\rho)$ and the RKHS $\mathcal{H}$ can be described using the same orthonormal basis $(\varphi_j)$, but with different norms.

**Remark 13.** *(Comparison between $\mathcal{H}$ and $\mathbb{L}^2_\rho(\mathcal{W})$). A function $h \in L^2(\rho)$ has an expansion $h = \sum_j h_j \varphi_j$, and:*

$$\|h\|^2_{L^2(\rho)} = \sum_{j=1}^{\infty} h_j^2.$$

*A function $h \in \mathcal{H}$ has the same expansion, but the RKHS norm is:*

$$\|h\|^2_{\mathcal{H}} = \sum_{j=1}^{\infty} \frac{h_j^2}{\eta_j}.$$

*This means that functions with large coefficients on eigenfunctions associated with small eigenvalues are heavily penalized in $\mathcal{H}$, which enforces a notion of smoothness.*

To summarize, the space $\mathbb{L}^2_\rho$ contains all square-integrable functions with respect to the measure $\rho$. In contrast, the RKHS $\mathcal{H}$ is a subspace of $\mathbb{L}^2_\rho$ consisting of *smoother* functions—those whose spectral expansions put less weight on high-frequency eigenfunctions (i.e., those associated with small eigenvalues $\eta_j$).

This motivates two classical assumptions from statistical learning theory: the *smoothness assumption*, which constrains the target function via its spectral decay profile, and the *spectral decay assumption*, which characterizes the approximation capacity of the RKHS.

**Remark 14.** *The smoothness assumption governs the approximation error (bias), while the spectral decay controls the estimation error (variance). These assumptions together determine the learning rate of kernel methods.*

**Source condition**   To control the *bias* introduced by ridge regularization, we assume that the target function lies in a smoother subspace of the RKHS. This is formalized by a *source condition*, a common assumption in inverse problems and kernel learning theory [55, 87, 88].

**Assumption 15.** *(Source Condition) There exists $c \in (1, 2]$ such that the target function $h$ belongs to the subspace*

$$\mathcal{H}^c = \left\{ f = \sum_{j=1}^\infty h_j \varphi_j \; : \; \sum_{j=1}^\infty \frac{h_j^2}{\eta_j^c} < \infty \right\} \subset \mathcal{H}.$$

When $c = 1$, this corresponds to assuming only that $h \in \mathcal{H}$. Larger values of $c$ imply greater smoothness: the function $h$ can be well-approximated using only the leading eigenfunctions. Intuitively, smoother targets lead to smaller bias and enable faster convergence of the estimator $\hat{h}$.

**Variance and spectral decay**   To control the *variance* of kernel ridge regression, we must also constrain the complexity of the RKHS. This is done via a *spectral decay assumption*, which controls the effective dimension of the RKHS by quantifying how quickly the eigenvalues $\eta_j$ of the kernel operator vanish.

**Assumption 16.** *(Spectral Decay) We assume that there exists a constant $C > 0$ such that, for all $j$,*

$$\eta_j \leq Cj^{-b}, \quad \textit{for some } b \geq 1.$$

This polynomial decay condition ensures that the contributions of high-frequency components decrease rapidly. A bounded kernel implies that $b \geq 1$ [56, Lemma 10]. In the limit $b \to \infty$, the RKHS becomes finite-dimensional. Intermediate values of $b$ define how "large" or complex the RKHS is, relative to the underlying measure $\rho$. A larger $b$ corresponds to a smaller effective dimension and thus a lower variance in estimation.

**Space regularity**   We can also require an additional assumption on the regularity of the domains.

**Assumption 17.** *(Original Space Regularity Conditions) Assume that $\mathcal{A}, \mathcal{X}$ (and $\mathcal{Y}$) are Polish spaces.*

A Polish space is a separable, completely metrizable topological space. This assumption covers a broad range of settings, including discrete, continuous, and infinite-dimensional cases. When the outcome $Y$ is bounded, the moment condition is automatically satisfied.

### 9.3   Further details on Counterfactual Policy Mean Embeddings

To justify Proposition 2, we rely on the classical identification strategy established by Rosenbaum and Rubin [44] and Robins [45]. Recall that the counterfactual policy mean embedding is defined as

$$\chi(\pi) := \mathbb{E}_{\pi \times P_X} \left[ \phi_{\mathcal{Y}}(Y(a)) \right],$$

which involves the unobserved potential outcome $Y(a)$. Under Assumption 1, we proceed to express this quantity in terms of observed data.

First, by the consistency assumption, we have that for any realization where $A = a$, the observed outcome satisfies $Y = Y(a)$. Second, by conditional exchangeability, we have that $Y(a) \perp A \mid X$, which implies that the conditional distribution of $Y(a)$ given $X = x$ is equal to the conditional distribution of $Y$ given $A = a, X = x$. That is,

$$\mathbb{E}[\phi_{\mathcal{Y}}(Y(a)) \mid X = x] = \mathbb{E}[\phi_{\mathcal{Y}}(Y) \mid A = a, X = x] = \mu_{Y|A,X}(a, x).$$

Finally, under the strong positivity assumption, the conditional density $\pi_0(a \mid x)$ is strictly bounded away from zero for all $a \in \mathcal{A}$, $x \in \mathcal{X}$, ensuring that the conditional expectation $\mu_{Y|A,X}(a, x)$ is identifiable throughout the support of $\pi \times P_X$. It follows that

$$\chi(\pi) = \mathbb{E}_{\pi \times P_X} \left[ \mathbb{E}[\phi_{\mathcal{Y}}(Y(a)) \mid X = x] \right] = \mathbb{E}_{\pi \times P_X} \left[ \mu_{Y|A,X}(a, x) \right],$$

which completes the identification argument.

## 10 Details and Analysis of the Plug-in Estimator

In this appendix, we provide further details on the analysis of the plug-in estimator proposed in Section 3.

### 10.1 Decoupling

We propose a plug-in estimator based on conditional mean operators for the nonparametric distribution of the outcome under policy a target policy $\pi$. Due to a decomposition property specific to the reproducing kernel Hilbert space, our plug-in estimator has a simple closed form solution.

**Proposition 4** ((Decoupling via kernel mean embedding))**.** *Suppose Assumptions 1 and 3 hold. Then, the counterfactual policy mean embedding can be expressed as:*

$$\chi(\pi) = C_{Y|A,X}\mu_\pi$$

*Proof.* In Assumption 3, we impose that the scalar kernels are bounded. This assumption has several implications. First, the feature maps are Bochner integrable [84, see Definition A.5.20]. Bochner integrability permits us to interchange the expectation and inner product. Second, the mean embeddings exist. Third, the product kernel is also bounded and hence the tensor product RKHS inherits these favorable properties. By Proposition 2 and the linearity of expectation,

$$\chi(\pi) = \int \mu_{Y|A,X}(a,x)\mathrm{d}\pi(a|x)\mathrm{d}P(x)$$

$$= \int \mathcal{C}_{Y|A,X}\{\phi_{\mathcal{A}}(a) \otimes \phi_{\mathcal{X}}(x)\}\mathrm{d}\pi(a|x)\mathrm{d}P(x)$$

$$= \mathcal{C}_{Y|A,X}\int \phi_{\mathcal{A}}(a) \otimes \phi_{\mathcal{X}}(x)\mathrm{d}\pi(a|x)\mathrm{d}P(x)$$

$$= \mathcal{C}_{Y|A,X}\mu_\pi.$$

$\square$

### 10.2 Analysis of the plug-in estimator

We will now present technical lemmas for kernel mean embeddings and conditional mean embeddings.

**Kernel mean embedding** For expositional purposes, we summarize classic results for the kernel mean embedding estimator $\hat{\mu}_z$ for $\mu_z = E\{\phi(Z)\}$.

**Lemma 10.1.** *(Bennett inequality; Lemma 2 of Smale and Zhou [88]) Let $(\xi_i)$ be i.i.d. random variables drawn from the distribution $P$ taking values in a real separable Hilbert space $\mathcal{K}$. Suppose there exists $M$ such that $\|\xi_i\|_{\mathcal{K}} \leq M < \infty$ almost surely and $\sigma^2(\xi_i) = E\left(\|\xi_i\|_{\mathcal{K}}^2\right)$. Then for all $n \in \mathbb{N}$ and for all $\delta \in (0,1)$,*

$$\mathrm{pr}\left[\left\|\frac{1}{n}\sum_{i=1}^n \xi_i - E(\xi)\right\|_{\mathcal{K}} \leq \frac{2M\log(2/\delta)}{n} + \left\{\frac{2\sigma^2(\xi)\log(2/\delta)}{n}\right\}^{1/2}\right] \geq 1 - \delta$$

We next provide a convergence result for the mean embedding, following from the above. This is included to make the paper self contained, however see [57, Proposition A.1] for an improved constant and a proof that the rate is minimax optimal.

**Proposition 18.** *(Mean embedding Rate). Suppose Assumptions 3 and 17 hold. Then with probability $1 - \delta$,*

$$\|\hat{\mu}_\pi - \mu_\pi\|_{\mathcal{H}} \leq r_{\mu_\pi}(n,\delta) = \frac{4\kappa_z\log(2/\delta)}{n^{1/2}}$$

*Proof.* The result follows from Lemma 10.1 with $\xi_i = \phi(Z_i)$, since

$$\left\|n^{-1}\sum_{i=1}^n \phi(Z_i) - E_{P_\pi}\{\phi(Z)\}\right\|_{\mathcal{H}_{\mathcal{Z}}} \leq \frac{2\kappa_z\log(2/\delta)}{n} + \left\{\frac{2\kappa_z^2\log(2/\delta)}{n}\right\}^{1/2} \leq \frac{4\kappa_z\log(2/\delta)}{n^{1/2}}$$

See [21, Theorem 2] for an alternative argument via Rademacher complexity. $\square$

**Conditional mean embeddings** Below, we restate Assumptions 15 and 16 for the RKHS $\mathcal{H}_{\mathcal{A}\mathcal{X}}$, which are used to establish the convergence rate of learning the conditional mean operator $C_{Y|A,X}$. Our formulation of Assumption 15 differs slightly from the one in Appendix 9.2, but they are equivalent due to [55, Remark 2].

**Assumption 15** (Source condition.). *We define the (uncentered) covariance operator* $\Sigma_{AX} = \mathbb{E}[\phi_{\mathcal{A}\mathcal{X}}(A, X) \otimes \phi_{\mathcal{A}\mathcal{X}}(A, X)]$. *There exists a constant* $B < \infty$ *such that for a given* $c \in (1, 3]$,

$$\|C_{Y|A,X}\Sigma_{AX}^{-\frac{c-1}{2}}\|_{S_2(\mathcal{H}_{\mathcal{A}\mathcal{X}}, \mathcal{H}_{\mathcal{Y}})} \leq B$$

In the above assumption, the smoothness parameter is allowed to range up to $c \leq 3$, in contrast to prior work on kernel ridge regression, which typically restricts it to $c \leq 2$ [e.g. 56]. This extension is justified by Meunier et al. [89, Remark 7 and Proposition 7], who showed that the saturation effect of Tikhonov regularization can be extended to $c \leq 3$ when the error is measured in the RKHS norm, as in Theorem 19, rather than the $L^2$ norm.

**Assumption 16** (Eigenvalue decay.). *Let* $(\lambda_{1,i})_{i \geq 1}$ *be the eigenvalues of* $\Sigma_{AX}$. *For some constant* $B > 0$ *and parameter* $b \in (0, 1]$ *and for all* $i \geq 1$,

$$\lambda_{1,i} \leq Ci^{-b}.$$

**Theorem 19.** *(Theorem 3 [51]) Suppose Assumptions, 3, 15, 16 and 17, hold and take* $\lambda_1 = \Theta\left(n^{-\frac{1}{c+1/b}}\right)$. *There is a constant* $J_1 > 0$ *independent of* $n \geq 1$ *and* $\delta \in (0, 1)$ *such that*

$$\left\|\hat{C}_{Y|A,X} - C_{Y|A,X}\right\|_{S_2(\mathcal{H}_{\mathcal{A}\mathcal{X}}, \mathcal{H}_{\mathcal{Y}})} \leq J_1 \log(4/\delta) \left(\frac{1}{\sqrt{n}}\right)^{\frac{c-1}{c+1/b}} =: r_C(\delta, n, b, c)$$

*is satisfied for sufficiently large* $n \geq 1$ *with probability at least* $1 - \delta$.

We will now appeal to these previous lemmas to prove the consistency of the causal function.

**Theorem 5** ((Consistency of the plug-in estimator.).). *Suppose Assumptions 1, 3, 15, 16 and 17. Set* $\lambda = n^{-1/(c+1/b)}$, *which is rate optimal regularization. Then, with high probability,*

$$\|\hat{\chi}_{pi}(\pi) - \chi(\pi)\|_{\mathcal{H}_{\mathcal{Y}}} = O[r_C(n, \delta, b, c)] = O\left[n^{-(c-1)/\{2(c+1/b)\}}\right]$$

*Proof of Theorem 5.* We note that

$$\hat{\chi}_{pi}(\pi) - \chi(\pi) = \hat{C}_{Y|A,X}\hat{\mu}_\pi - C_{Y|A,X}\mu_\pi$$

$$= \hat{C}_{Y|A,X}(\hat{\mu}_\pi - \mu_\pi) + \left(\hat{C}_{Y|A,X} - C_{Y|A,X}\right)\mu_\pi$$

$$= \left(\hat{C}_{Y|A,X} - C_{Y|A,X}\right)(\hat{\mu}_\pi - \mu_\pi) + C_{Y|A,X}(\hat{\mu}_\pi - \mu_\pi) + \left(\hat{C}_{Y|A,X} - C_{Y|A,X}\right)\mu_\pi.$$

Therefore we can write with Cauchy-Schwartz inequality:

$$|\hat{\chi}_{pi}(\pi) - \chi(\pi)| \leq \left\|\hat{C}_{Y|A,X} - C_{Y|A,X}\right\|_{S_2(\mathcal{H}_{\mathcal{A}\mathcal{X}}, \mathcal{H}_{\mathcal{Y}})} \|\hat{\mu}_\pi - \mu_\pi\|_{\mathcal{H}}$$

$$+ \left\|C_{Y|A,X}\right\|_{S_2(\mathcal{H}_{\mathcal{A}\mathcal{X}}, \mathcal{H}_{\mathcal{Y}})} \|\hat{\mu}_\pi - \mu_\pi\|_{\mathcal{H}}$$

$$+ \left\|\hat{C}_{Y|A,X} - C_{Y|A,X}\right\|_{S_2(\mathcal{H}_{\mathcal{A}\mathcal{X}}, \mathcal{H}_{\mathcal{Y}})} \|\mu_\pi\|_{\mathcal{H}}$$

Therefore by Theorems 19 and 18, with probability $1 - 2\delta$,

$$|\hat{\chi}_{pi}(\pi) - \chi(\pi)| \leq \cdot r_C(n, \delta, b, c) \cdot r_\mu(n, \delta) + \left\|C_{Y|A,X}\right\|_{S_2(\mathcal{H}_{\mathcal{A}\mathcal{X}}, \mathcal{H}_{\mathcal{Y}})} \cdot r_\mu(n, \delta) + \kappa_{a,x} \cdot r_C(n, \delta, b, c).$$

Using Assumption 15, we observe that $\left\|C_{Y|A,X}\right\|_{S_2(\mathcal{H}_{\mathcal{A}\mathcal{X}}, \mathcal{H}_{\mathcal{Y}})} \leq B\kappa^{c-1}$. As a result, the above bound readily gives

$$|\hat{\chi}_{pi}(\pi) - \chi(\pi)| \lesssim n^{-\frac{1}{2}\frac{c-1}{c+1/b}}.$$

$\square$

## 10.3 Further details and Estimation strategies for the kernel policy mean embedding

**Discrete Action Spaces.** When the action space $\mathcal{A}$ is discrete, we can directly compute the kernel policy mean embedding by exploiting the known form of the target policy $\pi(a \mid x)$. For each logged context $x_i$, we compute a convex combination of the feature maps $\phi_{\mathcal{A}\mathcal{X}}(a, x_i)$, weighted by the policy $\pi(a \mid x_i)$. This leads to the following empirical estimator:

$$\hat{\mu}_\pi = \frac{1}{n} \sum_{i=1}^n \sum_{a \in \mathcal{A}} \pi(a \mid x_i)\phi_{\mathcal{A}\mathcal{X}}(a, x_i).$$

The plug-in estimator for the counterfactual policy mean embedding then admits the following matrix expression:

$$\hat{\chi}(\pi) = \hat{C}_{Y|A,X}\hat{\mu}_\pi$$
$$= (K_{AA} \odot K_{XX} + n\lambda I)^{-1} (\Phi_{\mathcal{A}} \otimes \Phi_{\mathcal{X}}) \left( \frac{1}{n} \sum_{i=1}^n \sum_{a \in \mathcal{A}} \pi(a \mid x_i)\phi_{\mathcal{A}\mathcal{X}}(a, x_i) \right)$$
$$= (K_{AA} \odot K_{XX} + n\lambda I)^{-1} \underbrace{(\Phi_{\mathcal{A}} \otimes \Phi_{\mathcal{X}})(\Phi_\pi \otimes \Phi_{\mathcal{X}})}_{K_\pi \odot K_{XX}} \mathbf{1}\frac{1}{n}$$
$$= (K_{AA} \odot K_{XX} + n\lambda I)^{-1} (K_\pi \odot K_{XX}) \mathbf{1}\frac{1}{n},$$

where $K_\pi[i,j] = \sum_{a \in \mathcal{A}} k_{\mathcal{A}}(a_i, a)\pi(a \mid x_j)$, and $\Phi_\pi$ denotes the policy-weighted features.

---

**Algorithm 3** Plug-in estimator of the CPME (Discrete actions)

---

**Require:** Kernels $k_{\mathcal{X}}$, $k_{\mathcal{A}}$, $k_{\mathcal{Y}}$, and regularization constant $\lambda > 0$.
**Input:** Logged data $(x_i, a_i, y_i)_{i=1}^n$, target policy $\pi(a \mid x)$.
1: Compute empirical kernel matrices $K_{AA}, K_{XX} \in \mathbb{R}^{n \times n}$ from the samples $\{(a_i, x_i)\}_{i=1}^n$
2: Compute the kernel outcome matrix $K_{yY} = [k_{\mathcal{Y}}(y_1, y), \dots, k_{\mathcal{Y}}(y_n, y)]$
3: Compute $K_\pi \in \mathbb{R}^{n \times n}$ with entries $K_\pi[i,j] = \sum_{a \in \mathcal{A}} \pi(a \mid x_j) \cdot k_{\mathcal{A}\mathcal{X}}((a_i, x_i), (a, x_j))$
4: Set $\tilde{K} = K_\pi \cdot \frac{1}{n} \cdot (1 \dots 1)^\top$
**Output:** An estimate $\hat{\chi}_{pi}(\pi)(y) = K_{yY} (K_{AA} \odot K_{XX} + n\lambda I)^{-1} \tilde{K}$.

---

**Continuous Actions via Resampling.** When $\mathcal{A}$ is continuous and no closed-form sum over actions is available, we instead approximate the kernel policy mean embedding by resampling from $\pi(\cdot \mid x_i)$. Specifically, for each logged covariate $x_i$, we sample $\tilde{a}_i \sim \pi(\cdot \mid x_i)$, and form the empirical estimate:

$$\hat{\mu}_\pi = \frac{1}{n} \sum_{i=1}^n \phi_{\mathcal{A}\mathcal{X}}(\tilde{a}_i, x_i).$$

This leads to the following expression for the plug-in estimator:

$$\hat{\chi}(\pi) = \hat{C}_{Y|A,X}\hat{\mu}_\pi$$
$$= (K_{AA} \odot K_{XX} + n\lambda I)^{-1} (\Phi_{\mathcal{A}} \otimes \Phi_{\mathcal{X}})\frac{1}{n} \sum_{i=1}^n \phi_{\mathcal{A}\mathcal{X}}(\tilde{a}_i, x_i)$$
$$= (K_{AA} \odot K_{XX} + n\lambda I)^{-1} \underbrace{(\Phi_{\mathcal{A}} \otimes \Phi_{\mathcal{X}})(\Phi_{\tilde{A}} \otimes \Phi_{\mathcal{X}})}_{K_{A\tilde{A}} \odot K_{XX}} \mathbf{1}\frac{1}{n}$$
$$= (K_{AA} \odot K_{XX} + n\lambda I)^{-1} (K_{A\tilde{A}} \odot K_{XX}) \mathbf{1}\frac{1}{n},$$

where $K_{A\tilde{A}}[i,j] = k_{\mathcal{A}}(a_i, \tilde{a}_j)$, and $\tilde{a}_j$ is drawn from $\pi(\cdot \mid x_j)$.

---

**Algorithm 4** Plug-in estimator of the CPME

---

**Require:** Kernels $k_{\mathcal{X}}$, $k_{\mathcal{A}}$, $k_{\mathcal{Y}}$, and regularization constant $\lambda > 0$.
**Input:** Logged data $(x_i, a_i, y_i)_{i=1}^n$, the target policy $\pi$,
  1: Compute empirical kernel matrices $K_{AA}, K_{XX} \in \mathbb{R}^{T \times T}$ from the empirical samples
  2: Compute the kernel outcome matrix $K_{yY} = [k_{\mathcal{Y}}(y_1, y), \dots, k_{\mathcal{Y}}(y_n, y)]$
  3: Compute $\tilde{K}$ with resampling, $\tilde{K} = (K_{A\tilde{A}} \odot K_{XX}).(1 \dots 1)^\top \frac{1}{n}$ and $\tilde{A} \sim \pi(\cdot|X)$.
**Output:** An estimate $\hat{\chi}_{pi}(\pi)(y) = K_{yY}(K_{AA} \odot K_{XX} + n\lambda I)^{-1} \tilde{K}$.

---

**Importance Sampling** This resampling procedure can be quite cumbersome however, and not appropriate for off-policy learning. When propensity scores are known, an optional alternative is to invoke an inverse propensity scoring method [90], which expresses the embedding under the target policy $\pi$ as a reweighting of the observational distribution:

$$\mu_\pi = \mathbb{E}_{\pi_0 \times P_X}\left[\frac{\pi(a \mid x)}{\pi_0(a \mid x)}\phi_{\mathcal{A}\mathcal{X}}(a, x)\right]. \tag{20}$$

This formulation enables a direct estimator of $\mu_\pi$ from logged data $\{(x_i, a_i, y_i)\}_{i=1}^n$, using the known logging policy $\pi_0$:

$$\hat{\mu}_\pi = \frac{1}{n}\sum_{i=1}^n \frac{\pi(a_i \mid x_i)}{\pi_0(a_i \mid x_i)}\phi_{\mathcal{A}\mathcal{X}}(a_i, x_i). \tag{21}$$

Let $W_\pi \in \mathbb{R}^n$ be the vector of importance weights $W_\pi[i] = \frac{\pi(a_i|x_i)}{\pi_0(a_i|x_i)}$, and let $\Phi_{\mathcal{A}\mathcal{X}} = [\phi_{\mathcal{A}\mathcal{X}}(a_1, x_1), \dots, \phi_{\mathcal{A}\mathcal{X}}(a_n, x_n)]$. Then the estimator admits the vectorized form:

$$\hat{\mu}_\pi = \Phi_{\mathcal{A}\mathcal{X}}\left(\frac{1}{n}W_\pi\right). \tag{22}$$

Accordingly, the closed-form expression for the plug-in estimator becomes:

$$\hat{\chi}(\pi) = \hat{C}_{Y|A,X}\hat{\mu}_\pi$$
$$= (K_{AA} \odot K_{XX} + n\lambda I)^{-1}(\Phi_{\mathcal{A}} \otimes \Phi_{\mathcal{X}}) \cdot \Phi_{\mathcal{A}\mathcal{X}}\left(\frac{1}{n}W_\pi\right)$$
$$= (K_{AA} \odot K_{XX} + n\lambda I)^{-1}(K_{AA} \odot K_{XX})W_\pi \cdot \frac{1}{n}.$$

This estimator leverages all observed samples without requiring resampling or external sampling procedures, and is especially suited to settings where both the logging and target policies are known or estimable. However, its stability critically depends on the variance of the importance weights $W_\pi$, which may require regularization or clipping in practice. Moreover, this estimator is not compatible with the doubly robust estimator proposed in the next section.

## 11  Details and Analysis of the Efficient Score Function based Estimator

In this appendix, we provide background definitions and lemmas on the pathwise differentiability of RKHS-valued parameters [34, 41], followed by the derivation and analysis of a one-step estimator for the counterfactual policy mean embedding (CPME).

As stated in Assumption 17, we work on a Polish space $(\mathcal{Z}, \mathcal{B})$ with $\mathcal{Z} = \mathcal{Y} \times \mathcal{A} \times \mathcal{X}$ and consider a collection of distributions $\mathcal{P}$ defined on $(\mathcal{Z}, \mathcal{B})$. Let $z_1, \dots, z_n \sim P_0$ be an i.i.d. sample from some $P_0 \in \mathcal{P}$, and denote by $P_n$ the empirical distribution. Let $\widehat{P}_n \in \mathcal{P}$ be an estimate of $P_0$. For a measure $\rho$ on $(\mathcal{X}, \Sigma)$, the space $L^2(\rho)$ denotes the Hilbert space of $\rho$-almost surely equivalence classes of real-valued square-integrable functions, equipped with the inner product $\langle f, g \rangle_{L^2(\rho)} := \int fg\, d\rho$. For any Hilbert space $\mathcal{H}$, we write $L^2(P; \mathcal{H})$ for the space of Bochner-measurable functions $f : \mathcal{Z} \to \mathcal{H}$ with finite norm

$$\|f\|_{L^2(P;\mathcal{H})} := \left(\int \|f(z)\|_{\mathcal{H}}^2\, dP(z)\right)^{1/2}.$$

If $\mathcal{W}$ is a closed subspace of $\mathcal{H}$, we denote by $\Pi_{\mathcal{H}}[h \mid \mathcal{W}]$ the orthogonal projection of $h$ onto $\mathcal{W}$.

## 11.1 Background on pathwise differentiability of RKHS-valued parameters

We begin with a brief review of the formalism used to characterize the smoothness of RKHS-valued statistical parameters [34, 41]. Let $\mathcal{P}$ be a model, i.e., a collection of probability distributions on the Polish space $(\mathcal{Y} \times \mathcal{A} \times \mathcal{X}, \mathcal{B})$, dominated by a common $\sigma$-finite measure $\rho$.

**Definition 11.1.** *(Quadratic mean differentiability) A submodel $\{P_\epsilon : \epsilon \in [0, \delta)\} \subset \mathcal{P}$ is said to be quadratic mean differentiable at $P$ if there exists a score function $s \in L^2(P)$ such that*

$$\left\| p_\epsilon^{1/2} - p^{1/2} - \frac{\epsilon}{2} s p^{1/2} \right\|_{L^2(\rho)} = o(\epsilon),$$

*where $p = \frac{dP}{d\rho}$ and $p_\epsilon = \frac{dP_\epsilon}{d\rho}$.*

We denote by $\mathscr{P}(P, \mathcal{P}, s)$ the set of submodels at $P$ with score function $s$. The collection of such $s \in L^2(P)$ for which $\mathscr{P}(P, \mathcal{P}, s) \neq \varnothing$ is called the *tangent set*, and its closed linear span is the *tangent space* of $\mathcal{P}$ at $P$, denoted $\dot{\mathcal{P}}_P$.

We define $L_0^2(P) := \{s \in L^2(P) : \int s\, dP = 0\}$, the largest possible tangent space, and refer to models with $\dot{\mathcal{P}}_P = L_0^2(P)$ for all $P \in \mathcal{P}$ as *locally nonparametric*.

The parameter of interest is the counterfactual policy mean embedding and can written over the model $\mathcal{P}$ as $\chi(\pi) : \mathcal{P} \to \mathcal{H}_\mathcal{Y}$, such that

$$\chi(\pi)(P) = \iint \mathbb{E}_P\left[\phi_\mathcal{Y}(Y) \mid A = a, X = x\right] \pi(da \mid x) P_X(dx). \tag{23}$$

**Definition 11.2.** *(Pathwise differentiability) The parameter $\chi(\pi)$ is pathwise differentiable at $P$ if there exists a continuous linear map $\dot{\chi}_P^\pi : \dot{\mathcal{P}}_P \to \mathcal{H}_\mathcal{Y}$ such that for all $\{P_\epsilon\} \in \mathscr{P}(P, \mathcal{P}, s)$,*

$$\|\chi(\pi)(P_\epsilon) - \chi(\pi)(P) - \epsilon \dot{\chi}_P^\pi(s)\|_{\mathcal{H}_\mathcal{Y}} = o(\epsilon).$$

*We refer to $\dot{\chi}_P^\pi$ as the local parameter of $\chi(\pi)$ at $P$, and its Hermitian adjoint $(\dot{\chi}_P^\pi)^* : \mathcal{H}_\mathcal{Y} \to \dot{\mathcal{P}}_P$ as the efficient influence operator. Its image, denoted $\dot{\mathcal{H}}_P$, is a closed subspace of $\mathcal{H}_\mathcal{Y}$ known as the local parameter space.*

Next, we go on defining the efficient influence function of the parameter $\chi(\pi)$.

**Definition 11.3.** *(Efficient influence function) We say that $\chi(\pi)$ has an efficient influence function (EIF) $\psi_P^\pi : \mathcal{Y} \times \mathcal{A} \times \mathcal{X} \to \mathcal{H}_\mathcal{Y}$ if there exists a $P$-almost sure set such that*

$$\dot{\chi}_P^{\pi,*}(h)(y, a, x) = \langle h, \psi_P^\pi(y, a, x) \rangle_{\mathcal{H}_\mathcal{Y}} \quad \text{for all } h \in \mathcal{H}_\mathcal{Y}.$$

By the Riesz representation theorem, $\chi(\pi)$ admits an EIF if and only if $\dot{\chi}_P^{\pi,*}(\cdot)(y, a, x)$ defines a bounded linear functional almost surely. In that case, $\psi_P^\pi(y, a, x)$ equals its Riesz representation in $\mathcal{H}_\mathcal{Y}$.

In our case, since $\mathcal{H}_\mathcal{Y}$ is an RKHS over a space $\mathcal{Y}$, the local parameter space $\dot{\mathcal{H}}_P$ is itself an RKHS over $\mathcal{Y}$, with associated feature map $\phi_\mathcal{Y}$. Define

$$\tilde{\psi}_P^\pi(y, a, x) := \dot{\chi}_P^{\pi,*}(\phi_\mathcal{Y})(y, a, x), \tag{24}$$

which serves as a candidate representation of the EIF. The following result will serve us to show that $\tilde{\psi}_P^\pi$ both provides the form of the EIF of $\chi$, when it exists, and also a sufficient condition that can be used to verify its existence.

**Proposition 20.** *[41, Theorem 1], Form of the efficient influence function Suppose $\chi$ is pathwise differentiable at $P$ and $\dot{\mathcal{H}}_P$ is an RKHS. Then:*

*i) If an EIF $\psi_P^\pi$ exists, then $\psi_P^\pi = \tilde{\psi}_P^\pi$ almost surely.*

*ii) If $\|\tilde{\psi}_P^\pi\|_{L^2(P;\mathcal{H}_\mathcal{Y})} < \infty$, then $\chi(\pi)$ admits an EIF at $P$.*

Prior to that, we state below a result to show a sufficient condition for pathwise differentiability.

**Lemma 11.1.** *(Sufficient condition for pathwise differentiability)* [91, Lemma 2] *The parameter* $\chi : \mathcal{P} \to \mathcal{H}_{\mathcal{Y}}$ *is pathwise differentiable at* $P$ *if:*

    *i)* $\dot{\chi}_P$ *is bounded and linear, and there exists a dense set of scores* $\mathcal{S}(P)$ *such that for all* $s \in \mathcal{S}(P)$, *a submodel* $\{P_\epsilon\} \in \mathscr{P}(P, \mathcal{P}, s)$ *satisfies*

$$\|\chi(P_\epsilon) - \chi(P) - \epsilon \dot{\chi}_P(s)\|_{\mathcal{H}_{\mathcal{Y}}} = o(\epsilon),$$

    *ii) and* $\chi$ *is locally Lipschitz at* $P$, *i.e., there exist* $(c, \delta) > 0$ *such that for all* $P_1, P_2 \in B_\delta(P)$,

$$\|\chi(P_1) - \chi(P_2)\|_{\mathcal{H}_{\mathcal{Y}}} \le c\, H(P_1, P_2),$$

    *where* $H(\cdot, \cdot)$ *denotes the Hellinger distance and* $B_\delta(P)$ *is the* $\delta$-*neighborhood of* $P$ *in Hellinger distance.*

Finally, we will show that under suitable conditions, an estimator of the form

$$\widehat{\chi}_n(\pi) := \chi(\pi)(\widehat{P}_n) + P_n \psi_n^\pi$$

achieves efficiency.

## 11.2 Derivation of the Efficient Influence Function

We now prove Lemma 4.1, which characterizes the existence and form of the efficient influence function (EIF) of the CPME. We begin by restating the lemma for convenience.

**Lemma 4.1** ((Existence and form of the efficient influence function).)**.** *Suppose Assumptions 1 and 17 hold. Then, the CPME* $\chi(\pi)$ *admits an EIF which is* $P$-*Bochner square integrable and takes the form*

$$\psi^\pi(y, a, x) = \frac{\pi(a \mid x)}{\pi_0(a \mid x)} \left\{ \phi_{\mathcal{Y}}(y) - \mu_{Y|A,X}(a, x) \right\} + \int \mu_{Y|A,X}(a', x) \pi(da' \mid x) - \chi(\pi).$$

The proof proceeds in two main steps. First, we establish that $\chi$ is pathwise differentiable in Lemma 11.2. Then, we derive the form of its EIF.

**Lemma 11.2.** $\chi$ *is pathwise differentiable relative to a locally nonparametric model* $\mathcal{P}$ *at any* $P \in \mathcal{P}$

*Proof.* Fix $\pi \in \Pi$. To prove this lemma, we apply Lemma 11.1 to establish the pathwise differentiability of $\chi$ relative to a restricted model $\mathcal{P}_{\pi_0}$. This model consists of all distributions $P'$ such that $\pi_{P'} = \pi_0$, and for which there exists $P \in \mathcal{P}$ with $P'_{Y|A,X} = P_{Y|A,X}$ and $P'_X = P_X$. Since the functional $\chi(\pi)$ does not depend on the treatment assignment mechanism, we may then extend pathwise differentiability from $\mathcal{P}_{\pi_0}$ to the full, locally nonparametric model $\mathcal{P}$.

Following the construction in Luedtke and Chung [41], we assume that for any $P \in \mathcal{P}$ and fixed $\delta > 0$, the model $\mathcal{P}$ contains submodels of the form $\{P_\epsilon : \epsilon \in [0, \delta)\}$, where the perturbations act only on the marginal of $X$ and the conditional of $Y \mid A, X$. Specifically,

$$\frac{dP_{\epsilon,X}}{dP_X}(x) = 1 + \epsilon s_X(x), \quad \frac{dP_{\epsilon,A|X}}{dP_{A|X}}(a \mid x) = 1, \quad \frac{dP_{\epsilon,Y|A,X}}{dP_{Y|A,X}}(y \mid a, x) = 1 + \epsilon s_{Y|A,X}(y \mid a, x),$$

where $s_X$ and $s_{Y|A,X}$ are measurable functions bounded in $(-\delta^{-1}, \delta^{-1})$, satisfying

$$\mathbb{E}_P[s_X(X)] = 0 \quad \text{and} \quad \mathbb{E}_P[s_{Y|A,X}(Y \mid A, X) \mid A, X] = 0 \quad \text{a.s..}$$

**Step 1: Boundedness and quadratic mean differentiability of the local parameter**    Let $\pi_0$ be such that $\pi_0 = \pi_{P'}$ for some fixed $P' \in \mathcal{P}$. The local parameter $\dot{\chi}_P^\pi(s)$ can be expressed as

$$\dot{\chi}_P^\pi(s) = \int \frac{\pi(a \mid x)}{\pi_0(a \mid x)} \phi_{\mathcal{Y}}(y) \left[ s_{Y|A,X}(y \mid a, x) + s_X(x) \right] P(dz). \tag{25}$$

**Boundedness.**    We first verify that $\dot{\chi}_P^\pi$ is a bounded operator. This will establish the first part of condition (i) of Lemma 11.1 for the model $\mathcal{P}$ at $P$.

Take any score function $s$ in the tangent space $\dot{\mathcal{P}}_P$. Define

$$s_{Y|A,X}(y \mid a, x) := s(x, a, y) - \mathbb{E}_P[s(X, A, Y) \mid A = a, X = x],$$

$$s_X(x) := \mathbb{E}_P[s(X, A, Y) \mid X = x].$$

It is straightforward to verify that $\mathbb{E}_P[s(X, A, Y) \mid A, X] - \mathbb{E}_P[s(X, A, Y) \mid X] = 0$ $P$-almost surely. Therefore, we have the decomposition $s = s_{Y|A,X} + s_X$. Since $s \in L^2(P)$, it follows that both $s_{Y|A,X}$ and $s_X$ are in $L^2(P)$ as well.

Now, under the strong positivity assumption and the boundedness of the kernel $\kappa$, the integrand

$$(x, a, y) \mapsto \frac{\pi(a \mid x)}{\pi_0(a \mid x)} \phi_{\mathcal{Y}}(y) \left[ s_{Y|A,X}(y \mid a, x) + s_X(x) \right]$$

belongs to $L^2(P; \mathcal{H}_{\mathcal{Y}})$. Hence, the local parameter $\dot{\chi}_P^\tau(s)$ is well-defined in $\mathcal{H}_{\mathcal{Y}}$.

To establish boundedness of the local parameter $\dot{\chi}_P^\tau$, we compute its squared RKHS norm:

$$\begin{aligned}
\|\dot{\chi}_P^\tau(s)\|_{\mathcal{H}_{\mathcal{Y}}}^2 &= \iint \frac{\pi(a \mid x)}{\pi_0(a \mid x)} \frac{\pi(a' \mid x')}{\pi_0(a' \mid x')} k_y(y, y') \left[ s_{Y|A,X}(y \mid a, x) + s_X(x) \right] \\
&\qquad \cdot \left[ s_{Y|A,X}(y' \mid a', x') + s_X(x') \right] P^2(dz, dz') \quad\quad (26) \\
&\leq \iint \frac{\pi(a \mid x)}{\pi_0(a \mid x)} \frac{\pi(a' \mid x')}{\pi_0(a' \mid x')} \sqrt{k_y(y, y) \, k_y(y', y')} \left| s_{Y|A,X}(y \mid a, x) + s_X(x) \right| \\
&\qquad \cdot \left| s_{Y|A,X}(y' \mid a', x') + s_X(x') \right| P^2(dz, dz') \quad\quad (27) \\
&= \left[ \int \frac{\pi(a \mid x)}{\pi_0(a \mid x)} \sqrt{k_y(y, y)} \left| s_{Y|A,X}(y \mid a, x) + s_X(x) \right| P(dz) \right]^2 \quad\quad (28) \\
&\leq \left[ \int \frac{\pi^2(a \mid x)}{\pi_0^2(a \mid x)} |k_y(y, y)| P(dz) \right] \cdot \left[ \int \left( s_{Y|A,X}(y \mid a, x) + s_X(x) \right)^2 P(dz) \right] \quad (29) \\
&\leq \frac{\sup_{a,x} \pi(a \mid x) \cdot \sup_{y \in \mathcal{Y}} |k_y(y, y)|}{\inf_{P' \in \mathcal{P}} \operatorname{ess\,inf}_{a,x} \pi_{P'}(a \mid x)} \cdot \int \left( s_{Y|A,X}(y \mid a, x) + s_X(x) \right)^2 P(dz) \quad (30) \\
&\leq \frac{\sup_{a,x} \pi(a \mid x) \cdot \sup_{y \in \mathcal{Y}} |k_y(y, y)|}{\inf_{P' \in \mathcal{P}} \operatorname{ess\,inf}_{a,x} \pi_{P'}(a \mid x)} \cdot \|s\|_{L^2(P)}^2. \quad\quad (31)
\end{aligned}$$

Here: the first inequality applies Jensen's inequality to pull absolute values inside, and Cauchy–Schwarz on the kernel $k_y$. The second applies Cauchy–Schwarz to split the integrals. The third uses Hölder's inequality with exponents $(1, \infty)$. The final inequality follows from decomposing $s = s_{Y|A,X} + s_X + s_{A|X}$, where

$$s_{A|X}(a \mid x) := \mathbb{E}_P[s(Z) \mid A = a, X = x] - \mathbb{E}_P[s(Z) \mid X = x].$$

We then use

$$\|s\|_{L^2(P)}^2 = \|s_{Y|A,X} + s_X\|_{L^2(P)}^2 + \|s_{A|X}\|_{L^2(P)}^2 \geq \|s_{Y|A,X} + s_X\|_{L^2(P)}^2.$$

Since the kernel $k_y$ is bounded and $\pi_0$ is uniformly bounded away from zero by the strong positivity assumption, the bound in (31) is finite. Therefore, $\dot{\chi}_P^\tau$ is a bounded linear operator.

**Quadratic mean differentiability.** We now establish that $\chi(\pi)$ is quadratic mean differentiable at $P$ with respect to the restricted model $\mathcal{P}_{\pi_0}$, assuming $\pi_0 = \pi_P$.

As in Luedtke and Chung [41], we consider a smooth submodel $\{P_\epsilon : \epsilon \in [0, \delta)\} \subset \mathcal{P}_{\pi_0}$ of the form:

$$\frac{dP_{\epsilon,X}}{dP_X}(x) = 1 + \epsilon s_X(x), \qquad \frac{dP_{\epsilon,A|X}}{dP_{A|X}}(a \mid x) = 1, \qquad \frac{dP_{\epsilon,Y|A,X}}{dP_{Y|A,X}}(y \mid a, x) = 1 + \epsilon s_{Y|A,X}(y \mid a, x),$$

where $s_X$ and $s_{Y|A,X}$ are bounded in $[-\delta^{-1}/2, \delta^{-1}/2]$, satisfy $\mathbb{E}_P[s_X(X)] = 0$ and $\mathbb{E}_P[s_{Y|A,X}(Y \mid A, X) \mid A, X] = 0$ almost surely. The score of this submodel at $\epsilon = 0$ is given by $s(x, a, y) = s_X(x) + s_{Y|A,X}(y \mid a, x)$, and its $L^2(P)$-closure spans the tangent space of $\mathcal{P}_{\pi_0}$ at $P$.

Letting $\dot\chi_P^\pi(s)$ be defined as in Equation (25), we compute

$$\|\chi(\pi)(P_\epsilon) - \chi(\pi)(P) - \epsilon\dot\chi_P^\pi(s)\|_{\mathcal{H}_\mathcal{Y}}^2$$

$$= \left\| \iiint \phi_\mathcal{Y}(y)(1 + \epsilon s_{Y|A,X}(y \mid a, x))(1 + \epsilon s_X(x))\, P_{Y|A,X}(dy \mid a, x)\pi(da \mid x)P_X(dx) \right.$$

$$- \iiint \phi_\mathcal{Y}(y)\, P_{Y|A,X}(dy \mid a, x)\pi(da \mid x)P_X(dx)$$

$$\left. - \epsilon \iiint \frac{\pi(a \mid x)}{\pi_0(a \mid x)}\phi_\mathcal{Y}(y)\left[s_{Y|A,X}(y \mid a, x) + s_X(x)\right] P_{Y|A,X}(dy \mid a, x)\pi_0(da \mid x)P_X(dx) \right\|_{\mathcal{H}_\mathcal{Y}}^2$$

$$= \epsilon^4 \left\| \iint \phi_\mathcal{Y}(y)s_{Y|A,X}(y \mid a, x)s_X(x)\, P_{Y|A,X}(dy \mid a, x)\pi(da \mid x)P_X(dx) \right\|_{\mathcal{H}_\mathcal{Y}}^2$$

$$= \epsilon^4 \left\| \int \frac{\pi(a \mid x)}{\pi_0(a \mid x)}\phi_\mathcal{Y}(y)s_{Y|A,X}(y \mid a, x)s_X(x)\, P(dz) \right\|_{\mathcal{H}_\mathcal{Y}}^2 .$$

This is $o(\epsilon^2)$ provided that the last $\mathcal{H}_\mathcal{Y}$-norm is finite. To verify this, observe that the integrand

$$(x, a, y) \mapsto \frac{\pi(a \mid x)}{\pi_0(a \mid x)}\phi_\mathcal{Y}(y)s_{Y|A,X}(y \mid a, x)s_X(x)$$

belongs to $L^2(P; \mathcal{H}_\mathcal{Y})$, since $k_\mathcal{Y}$, $s_{Y|A,X}$, and $s_X$ are bounded and $\pi_0$ satisfies the strong positivity assumption. Indeedm if we compute its squared norm:

$$\left\| \int \frac{\pi(a \mid x)}{\pi_0(a \mid x)}\phi_\mathcal{Y}(y)s_{Y|A,X}(y \mid a, x)s_X(x)\, P(dz) \right\|_{\mathcal{H}_\mathcal{Y}}^2$$

$$= \iint \frac{\pi(a \mid x)}{\pi_0(a \mid x)} \frac{\pi(a' \mid x')}{\pi_0(a' \mid x')}\, k_\mathcal{Y}(y, y')\, s_{Y|A,X}(y \mid a, x)s_X(x)\, s_{Y|A,X}(y' \mid a', x')s_X(x')\, P(dz)P(dz')$$

$$< \infty.$$

Thus, $\chi(\pi)$ is quadratic mean differentiable at $P$ relative to $\mathcal{P}_{\pi_0}$.

**Step 2: Local Lipschitzness.** Let $\pi_0 = \pi_{P'}$ for some fixed $P' \in \mathcal{P}$. We now verify that $\chi(\pi)$ is locally Lipschitz over the restricted model $\mathcal{P}_{\pi_0}$.

Fix any $P, \tilde P \in \mathcal{P}_{\pi_0}$. Define the $\pi$-reweighted distributions:

$$P^\pi(z) := \frac{\pi(a \mid x)}{\pi_0(a \mid x)}\, P(z), \qquad \tilde P^\pi(z) := \frac{\pi(a \mid x)}{\pi_0(a \mid x)}\, \tilde P(z),$$

where $z = (x, a, y)$. Then:

$$\|\chi(\pi)(P) - \chi(\pi)(\tilde P)\|_{\mathcal{H}_\mathcal{Y}}^2 = \iint k_\mathcal{Y}(y, y')\, (P^\pi - \tilde P^\pi)(dz)\, (P^\pi - \tilde P^\pi)(dz')$$

$$= \iint k_\mathcal{Y}(y, y') \frac{\pi(a \mid x)}{\pi_0(a \mid x)}\left(P - \tilde P\right)(dz)\, \frac{\pi(a' \mid x')}{\pi_0(a' \mid x')}\left(P - \tilde P\right)(dz')$$

$$= \iint k_\mathcal{Y}(y, y')\left[\sqrt{dP(z)} - \sqrt{d\tilde P(z)}\right]\left[\sqrt{dP(z')} - \sqrt{d\tilde P(z')}\right]$$

$$\times \left[\sqrt{dP(z)} + \sqrt{d\tilde P(z)}\right]\left[\sqrt{dP(z')} + \sqrt{d\tilde P(z')}\right]$$

$$\times \frac{\pi(a \mid x)}{\pi_0(a \mid x)}\frac{\pi(a' \mid x')}{\pi_0(a' \mid x')}.$$

Applying the Cauchy–Schwarz inequality yields:

$$\|\chi(\pi)(P) - \chi(\pi)(\tilde{P})\|^2_{\mathcal{H}_{\mathcal{Y}}} \leq \left(\iint k_{\mathcal{Y}}^2(y,y')\left[\frac{\pi(a\mid x)}{\pi_0(a\mid x)}\frac{\pi(a'\mid x')}{\pi_0(a'\mid x')}\right]^2\left[\sqrt{dP(z)}+\sqrt{d\tilde{P}(z)}\right]^2\right.$$

$$\left.\cdot\left[\sqrt{dP(z')}+\sqrt{d\tilde{P}(z')}\right]^2\right)^{1/2}$$

$$\cdot\left(\iint\left[\sqrt{dP(z)}-\sqrt{d\tilde{P}(z)}\right]^2\left[\sqrt{dP(z')}-\sqrt{d\tilde{P}(z')}\right]^2\right)^{1/2}$$

$$= (\Lambda)^{1/2}\cdot H^2(P,\tilde{P}).$$

Where $\Lambda = \iint k_{\mathcal{Y}}^2(y,y')\left[\frac{\pi(a\mid x)}{\pi_0(a\mid x)}\frac{\pi(a'\mid x')}{\pi_0(a'\mid x')}\right]^2\left[\sqrt{dP(z)}+\sqrt{d\tilde{P}(z)}\right]^2\left[\sqrt{dP(z')}+\sqrt{d\tilde{P}(z')}\right]^2$.
Using the inequality $(b+c)^2\leq 2(b^2+c^2)$ and applying Hölder's inequality:

$$\Lambda \leq 2\iint k_{\mathcal{Y}}^2(y,y')\left[\frac{\pi(a\mid x)}{\pi_0(a\mid x)}\right]^2\left[\frac{\pi(a'\mid x')}{\pi_0(a'\mid x')}\right]^2(P+\tilde{P})(dz)(P+\tilde{P})(dz')$$

$$\leq \frac{2\sup_{x,a}\pi^2(a\mid x)\cdot\sup_{y,y'}k_{\mathcal{Y}}^2(y,y')}{\inf_{P'\in\mathcal{P}}\inf_{x,a}\pi_{P'}^2(a\mid x)}.$$

This upper bound is finite under the strong positivity assumption and the boundedness of the kernel $k_{\mathcal{Y}}$. Therefore, $\chi(\pi)$ is locally Lipschitz over $\mathcal{P}_{\pi_0}$. This establishes part (ii) of Lemma 11.1 and therefore finishes the proof.

$\square$

Now that we have proved Lemma 11.2, we establish Lemma 4.1 and derive the form of the efficient influence function.

*Proof.* To prove Lemma 4.1, we first recall that the local parameter takes the form, for $s \in \dot{\mathcal{P}}_P$

$$\dot{\chi}_P^\pi(s) = \iiint \phi_{\mathcal{Y}}(y)\left[s_{Y\mid A,X}(y\mid a,x)+s_{A\mid X}(a\mid x)+s_X(x)\right]P_{Y\mid A,X}(dy\mid a,x)\pi(da\mid x)P_X(dx)$$

Therefore, the efficient influence operator takes the form for $h \in \mathcal{H}_{\mathcal{Y}}$

$$\dot{\chi}_P^{\pi,*}(h)(y,a,x) = \frac{\pi(a|x)}{\pi_0(a\mid x)}\{h(y)-\mathbb{E}_P[h(Y)\mid A=a,X=x]\} \tag{32}$$

$$+\int\mathbb{E}_P[h(Y)\mid A=a',X=x]\pi(da'\mid x) \tag{33}$$

$$-\iint\mathbb{E}_P[h(Y)\mid A=a',X=x']\,\pi(da'\mid x)P_X(dx') \tag{34}$$

By Proposition 20, the EIF is given by evaluating the efficient influence operator at the representer $\phi_{\mathcal{Y}}(y')$, that is

$$\psi_P^\pi(z)(y') = \dot{\chi}_P^{\pi,*}(\phi_{\mathcal{Y}}(y'))(y,a,x) = \frac{\pi(a|x)}{\pi_0(a\mid x)}\{\phi_{\mathcal{Y}}(y')-\mathbb{E}_P[\phi_{\mathcal{Y}}(y')\mid A=a,X=x]\}$$

$$+\int\mathbb{E}_P[\phi_{\mathcal{Y}}(y')\mid A=a',X=x]\pi(da'\mid x)$$

$$-\iint\mathbb{E}_P[\phi_{\mathcal{Y}}(y')\mid A=a',X=x']\,\pi(da'\mid x)P_X(dx').$$

Indeed this function belongs to $L^2(P; \mathcal{H}_\mathcal{Y})$. Recalling the definition of the conditional mean embedding $\mu_{Y|A,X}(a, x)$ in (3) and noting that $\mathbb{E}_P \left[ \mu_{Y|A,X}(a, x) \right] = \chi(\pi)(P)$, we can rewrite the above as follows:

$$\psi_P^\pi(z) = \frac{\pi(a \mid x)}{\pi_0(a \mid x)} \left[ \phi_\mathcal{Y}(y) - \mu_{Y|A,X}(a, x) \right] + \int \mu_{Y|A,X}(a', x) \pi(da' \mid x) - \chi(\pi)(P)$$

Finally, since the kernel $k_\mathcal{Y}$ is bounded and $\pi_0$ is bounded away from zero by Assumption 1, it follows that $\psi_P^\pi \in L^2(P; \mathcal{H}_\mathcal{Y})$.

$\square$

## 11.3 Analysis of the one-step estimator

In this section we provide the analysis of the one-step estimator. We start by restating Theorem 6.

**Theorem 6** ((Consistency of the doubly robust estimator).). *Suppose Assumptions 1, 3, 15, 16 and 17. Set $\lambda = n^{-1/(c+1/b)}$, which is rate optimal regularization. Then, with high probability,*

$$\|\hat{\chi}_{dr}(\pi) - \chi(\pi)\|_{\mathcal{H}_\mathcal{Y}} = \mathcal{O} \left[ n^{-1/2} + r_{\pi_0}(n).r_C(n, \delta, b, c) \right]$$

For this Theorem, we will begin by decomposing the error terms .

$$\hat{\chi}_{dr} - \chi(P) = \chi(\hat{P}_n) + P_n \hat{\psi}_n - \chi(P) = (P_n - P)\hat{\psi}_n + \chi(\hat{P}_n) + P\hat{\psi}_n - \chi(P) \quad (35)$$

$$= (P_n - P)\psi^\pi + (P_n - P)(\hat{\psi}_n^\pi - \psi^\pi) + \chi(\hat{P}_n) + P\hat{\psi}_n^\pi - \chi(\pi) \quad (36)$$

$$= \mathcal{S}_n + \mathcal{T}_n + \mathcal{R}_n \quad (37)$$

where $\mathcal{S}_n = (P_n - P)\psi^\pi$, $\mathcal{T}_n = (P_n - P)(\hat{\psi}_n^\pi - \psi^\pi)$ and $\mathcal{R}_n = \chi(\hat{P}_n) + P\hat{\psi}_n^\pi - \chi(\pi)$. $\mathcal{S}_n$ is a sample average of a fixed function. We call $\mathcal{R}_n$ the remainder terms and $\mathcal{T}_n$ the empirical process term. The remainder terms $\mathcal{R}_n$, quantify the error in the approximation of the one-step estimator across the samples. The following result provides a reasonable condition under which the drift terms will be negligible.

### 11.3.1 Bounding the empirical process term

As explained in Appendix 11.4, Luedtke and Chung [41] proposed a cross-fitted version of the one-step estimator. However, splitting the data may lead to a loss in power. We are therefore interested in identifying a sufficient condition under which the empirical term $\mathcal{T}_n$ becomes asymptotically negligible *without* sample splitting.

In the scalar-valued case, a Donsker class assumption ensures the empirical process term is asymptotically negligible [32]. However, directly extending this notion to $\mathcal{H}_\mathcal{Y}$-valued functions is not straightforward, since standard entropy-based arguments rely on the total ordering of $\mathbb{R}$ [65]. Fortunately, Park and Muandet [65] introduces a notion of *asymptotic equicontinuity* adapted to Banach- or Hilbert-space valued empirical processes, which we adopt in this setting.

**Definition 11.4.** *(Asymptotic equicontinuity).* *We say that the empirical process $\{\mathcal{T}_n(\varphi) = \sqrt{n} (P_n - P) \varphi : \varphi \in \mathcal{G}\}$ with values in $\mathcal{H}$ and indexed by $\mathcal{G}$ is asymptotic equicontinuous at $\varphi_0 \in \mathcal{G}$ if, for every sequence $\{\hat{\varphi}_n\} \subset \mathcal{G}$ with $\|\hat{\varphi}_n - \varphi_0\| \overset{p}{\to} 0$, we have*

$$\|\mathcal{T}_n(\hat{\varphi}_n) - \mathcal{T}_n(\varphi_0)\|_\mathcal{H} \overset{p}{\to} 0. \quad (38)$$

Note that (38) is equivalent to $\mathcal{T}_n = (P_n - P)(\hat{\psi}_n^\pi - \psi^\pi) = o_P \left( \frac{1}{\sqrt{n}} \right)$. Park and Muandet [65] gives sufficient conditions for asymptotic equicontinuity to hold that we will leverage to show asymptotic equicontinuity. First we state the following result on the convergence of the efficient influence function estimator.

**Assumption 21.** *(Estimated Positivity) There exists a constant $\eta > 0$ such that, with high probability as $n \to \infty$,*

$$\hat{\pi}_0(a \mid x) \geq \eta, \quad \text{for all } (a, x) \in \mathcal{A} \times \mathcal{X}.$$

**Lemma 11.3.** *(Influence Function Error). Suppose that the conditions of Lemma 4.1 hold, as well as Assumptions 3, 21. Then the following bound holds:*

$$\|\psi_P^\pi - \psi_0^\pi\|_{L^2(P_0;\mathcal{H}_Y)} = O_P\left(\left\|\frac{1}{\hat{\pi}_0} - \frac{1}{\pi_0}\right\|_{L^2(\pi P_X)} + \left\|\mu_{Y|A,X} - \hat{\mu}_{Y|A,X}\right\|_{L^2(\pi P_X;\mathcal{H}_Y)}.\right)$$

*Proof.* We expand the difference between the estimated and oracle influence functions:

$$\psi_P^\pi(z) - \psi_0^\pi(z) = \left(\frac{\pi(a \mid x)}{\hat{\pi}_0(a \mid x)} - \frac{\pi(a \mid x)}{\pi_0(a \mid x)}\right)\left(\phi_Y(y) - \mu_{Y|A,X}(a, x)\right)$$
$$+ \frac{\pi(a \mid x)}{\hat{\pi}_0(a \mid x)}\left(\mu_{Y|A,X}(a, x) - \hat{\mu}_{Y|A,X}(a, x)\right)$$
$$+ \int \left(\hat{\mu}_{Y|A,X}(a', x) - \mu_{Y|A,X}(a', x)\right)\pi(da' \mid x).$$

Taking the $L^2(P_0; \mathcal{H}_Y)$ norm and applying the triangle inequality yields:

$$\|\psi_P^\pi - \psi_0^\pi\|_{L^2(P_0;\mathcal{H}_Y)} \le \text{(I)} + \text{(II)} + \text{(III)},$$

where:

$$\text{(I)} = \left\|\left(\frac{\pi(a \mid x)}{\hat{\pi}_0(a \mid x)} - \frac{\pi(a \mid x)}{\pi_0(a \mid x)}\right)\left(\phi_Y(y) - \mu_{Y|A,X}(a, x)\right)\right\|_{L^2(P_0;\mathcal{H}_Y)},$$

$$\text{(II)} = \left\|\frac{\pi(a \mid x)}{\hat{\pi}_0(a \mid x)}\left(\mu_{Y|A,X}(a, x) - \hat{\mu}_{Y|A,X}(a, x)\right)\right\|_{L^2(P_0;\mathcal{H}_Y)},$$

$$\text{(III)} = \left\|\int \left(\hat{\mu}_{Y|A,X}(a', x) - \mu_{Y|A,X}(a', x)\right)\pi(da' \mid x)\right\|_{L^2(P_0;\mathcal{H}_Y)}.$$

First, we consider the term

$$\text{(I)} = \left\|\left(\frac{\pi(a \mid x)}{\hat{\pi}_0(a \mid x)} - \frac{\pi(a \mid x)}{\pi_0(a \mid x)}\right)\left(\phi_Y(y) - \mu_{Y|A,X}(a, x)\right)\right\|_{L^2(P_0;\mathcal{H}_Y)}.$$

Let $\Delta(a, x) := \frac{\pi(a|x)}{\hat{\pi}_0(a|x)} - \frac{\pi(a|x)}{\pi_0(a|x)}$ and $h(a, x, y) := \phi_Y(y) - \mu_{Y|A,X}(a, x) \in \mathcal{H}_Y$. Then,

$$\text{(I)}^2 = \int \|\Delta(a, x) \cdot h(a, x, y)\|_{\mathcal{H}_Y}^2 \, dP_0(a, x, y) = \int \Delta^2(a, x) \cdot \|h(a, x, y)\|_{\mathcal{H}_Y}^2 \, dP_0.$$

Applying the Cauchy–Schwarz inequality gives:

$$\text{(I)} \le \left(\int \Delta^2(a, x) \, dP_0\right)^{1/2}\left(\int \|\phi_Y(y) - \mu_{Y|A,X}(a, x)\|_{\mathcal{H}_Y}^2 \, dP_0\right)^{1/2}.$$

Noting that $P_0(da, dx) = \pi_0(a \mid x)P_X(dx)$ and using the change of measure:

$$\int r^2(a, x) \, dP_0 = \int \left(\pi_0(a \mid x)\pi(a \mid x)\left(\frac{1}{\hat{\pi}_0(a \mid x)} - \frac{1}{\pi_0(a \mid x)}\right)\right)^2 \pi(a \mid x)P_X(dx),$$

we obtain:

$$\text{(I)} \le \left\|\pi_0\pi\left(\frac{1}{\hat{\pi}_0} - \frac{1}{\pi_0}\right)\right\|_{L^2(\pi P_X)} \cdot \left(\int \|\phi_Y(y) - \mu_{Y|A,X}(a, x)\|_{\mathcal{H}_Y}^2 \, dP_0\right)^{1/2}.$$

Using that the kernel is bounded in Assumption 3, then the second factor is finite, and:

$$\text{(I)} = O_P\left(\left\|\left(\frac{1}{\hat{\pi}_0} - \frac{1}{\pi_0}\right)\right\|_{L^2(\pi P_X)}\right),$$

for some constant depending on the kernel and outcome variance.

Second, we analyze the term

$$\text{(II)} = \left\| \frac{\pi(a \mid x)}{\hat{\pi}_0(a \mid x)} \left( \mu_{Y|A,X}(a,x) - \hat{\mu}_{Y|A,X}(a,x) \right) \right\|_{L^2(P_0;\mathcal{H}_Y)}.$$

By definition of the $L^2(P_0;\mathcal{H}_Y)$ norm, we have:

$$\text{(II)}^2 = \int \left\| \frac{\pi(a \mid x)}{\hat{\pi}_0(a \mid x)} \left( \mu_{Y|A,X}(a,x) - \hat{\mu}_{Y|A,X}(a,x) \right) \right\|_{\mathcal{H}_Y}^2 dP_0(a,x)$$

$$= \int \left( \frac{\pi(a \mid x)}{\hat{\pi}_0(a \mid x)} \right)^2 \left\| \mu_{Y|A,X}(a,x) - \hat{\mu}_{Y|A,X}(a,x) \right\|_{\mathcal{H}_Y}^2 \pi_0(a \mid x) P_X(dx).$$

Changing the measure to $\pi(a \mid x) P_X(dx)$ and bounding the weight by positivity assumptions yields:

$$\text{(II)}^2 = \int \left\| \mu_{Y|A,X}(a,x) - \hat{\mu}_{Y|A,X}(a,x) \right\|_{\mathcal{H}_Y}^2 \cdot w(a,x) \cdot \pi(a \mid x) P_X(dx),$$

where $w(a,x) := \frac{\pi_0(a|x)\pi(a|x)}{\hat{\pi}_0^2(a|x)}$. If $\hat{\pi}_0 \geq \eta > 0$, because of Assumption 21, then

$$\text{(II)} = O_P\left( \left\| \mu_{Y|A,X} - \hat{\mu}_{Y|A,X} \right\|_{L^2(\pi P_X;\mathcal{H}_Y)} \right),$$

for some constant depending on the inverse propensity bound.

Eventually, we bound the term

$$\text{(III)} = \left\| \int \left( \hat{\mu}_{Y|A,X}(a',x) - \mu_{Y|A,X}(a',x) \right) \pi(da' \mid x) \right\|_{L^2(P_0;\mathcal{H}_Y)}.$$

Simply, the interm does Using Jensen's inequality in the Hilbert space $\mathcal{H}_Y$ [92, Chapter 6], for each fixed $x$, we have:

$$\left\| \int \left( \hat{\mu}(a',x) - \mu(a',x) \right) \pi(da' \mid x) \right\|_{\mathcal{H}_Y} \leq \int \left\| \hat{\mu}(a',x) - \mu(a',x) \right\|_{\mathcal{H}_Y} \pi(da' \mid x).$$

Now square both sides and integrate over $P_0(a,x) = \pi_0(a \mid x) P_X(dx)$. Since the integrand is independent of $a$, this is equivalent to integrating over $P_X$ with the density $\pi_0(a \mid x)$ marginalized out:

$$\text{(III)}^2 = \int \left\| \int \left( \hat{\mu}(a',x) - \mu(a',x) \right) \pi(da' \mid x) \right\|_{\mathcal{H}_Y}^2 \pi_0(a \mid x) P_X(dx)$$

$$= \int \left\| \int \left( \hat{\mu}(a',x) - \mu(a',x) \right) \pi(da' \mid x) \right\|_{\mathcal{H}_Y}^2 P_X(dx)$$

$$\leq \int \left( \int \left\| \hat{\mu}(a',x) - \mu(a',x) \right\|_{\mathcal{H}_Y} \pi(da' \mid x) \right)^2 P_X(dx)$$

$$\leq \int \int \left\| \pi(a'|x)(\hat{\mu}(a',x) - \mu(a',x)) \right\|_{\mathcal{H}_Y}^2 \pi(da' \mid x) P_X(dx),$$

Therefore, using that $\pi$ is bounded

$$\text{(III)} \leq \left\| \sup_{x,a} \pi(a \mid x) \right\| \left\| \hat{\mu}_{Y|A,X} - \mu_{Y|A,X} \right\|_{L^2(\pi P_X;\mathcal{H}_Y)}.$$

Combining the bounds yields the desired result. $\qquad\square$

Then, we are now in position to state:

**Lemma 11.4.** *(Asymptotic equicontinuity of the empirical process term) Suppose that Assumptions 1, 3, 15, 17, 21 hold. Moreover, assume $k_\mathcal{Y}$ is a $C^\infty$ Mercer kernel. Then the empirical process term satisfies $\|\mathcal{T}_n\|_{\mathcal{H}_Y} = o_P(n^{-1/2})$.*

*Proof.* Under Assumptions 1, 3, 15, 17, 21, the functions $\hat{\psi}_n^\pi(y,a,x) - \psi^\pi(y,a,x)$ lie in a finite and shrinking ball of the RKHS $\mathcal{H}_Y$, therefore if $k_{\mathcal{Y}}$ is a $C^\infty$ Mercer kernel, we can apply [85, Theorem D] on the class $\mathcal{G} := \{\hat{\psi}_n^\pi - \psi^\pi\} \subset L^2(P; \mathcal{H}_Y)$ to verify the conditions of Theorem 6 of Park and Muandet [65].

Then, by Lemma 11.3 and with consistency of the nuisance parameters, $\|\hat{\psi}_n^\pi - \psi^\pi\|_{L^2(P;\mathcal{H}_Y)} \to 0$, and by their stochastic equicontinuity result in Corollary 8, [65], we readily have:

$$\|(P_n - P)(\hat{\psi}_n^\pi - \psi^\pi)\|_{\mathcal{H}_Y} \to 0 \quad \text{in probability.}$$

Hence, $\|\mathcal{T}_n\|_{\mathcal{H}_Y} = o_P(n^{-1/2})$, completing the proof. $\qquad\square$

### 11.3.2 Bounding the remainder term

**Lemma 11.5.** *(Remainder term bound).* Assumptions 1, 3, 15,16, 17, 21, then $\|\mathcal{R}_n\|_{\mathcal{H}_{\mathcal{Y}}} = O_p\left(r_C(n,\delta,b,c)r_{\pi_0}(n)\right)$.

*Proof.* From the definitions, the remainder term can be written as

$$
\begin{aligned}
\mathcal{R}_n &= \chi(\hat{P}_n) + P_0\hat{\psi}_n^\pi - \chi(\pi) \\
&= \mathbb{E}_{P_0}\left[\frac{\pi(a\mid x)}{\hat{\pi}_0(a\mid x)}\left(\phi_{\mathcal{Y}}(y) - \hat{\mu}_{Y\mid A,X}(a,x)\right) + \int \hat{\mu}_{Y\mid A,X}(a',x)\pi(da'\mid x)\right] \\
&\quad - \mathbb{E}_{P_0}\left[\frac{\pi(a\mid x)}{\pi_0(a\mid x)}\left(\phi_{\mathcal{Y}}(y) - \mu_{Y\mid A,X}(a,x)\right) + \int \mu_{Y\mid A,X}(a',x)\pi(da'\mid x)\right] \\
&= \mathbb{E}_{P_0}\left[\frac{\pi(a\mid x)}{\hat{\pi}_0(a\mid x)}\left(\mathbb{E}\left[\phi_{\mathcal{Y}}(y)\mid A=a, X=x\right] - \hat{\mu}_{Y\mid A,X}(a,x)\right)\right] \\
&\quad + \mathbb{E}_{P_0}\left[+\int \left(\hat{\mu}_{Y\mid A,X}(a',x) - \mu_{Y\mid A,X}(a',x)\right)\pi(da'\mid x)\right] \\
&\quad - \mathbb{E}_{P_0}\left[\frac{\pi(a\mid x)}{\pi_0(a\mid x)}\left(\mathbb{E}\left[\phi_{\mathcal{Y}}(y)\mid A=a, X=x\right] - \mu_{Y\mid A,X}(a,x)\right)\right] \\
&= \mathbb{E}_{P_0}\left[\frac{\pi(a\mid x)}{\hat{\pi}_0(a\mid x)}\left(\mu_{Y\mid A,X}(a,x) - \hat{\mu}_{Y\mid A,X}(a,x)\right) + \int \left(\hat{\mu}_{Y\mid A,X}(a',x) - \mu_{Y\mid A,X}(a',x)\right)\pi(da'\mid x)\right]
\end{aligned}
$$

We can expand the expectation into the following:

$$
\begin{aligned}
\mathcal{R}_n &= \iint \left[\frac{\pi(a\mid x)}{\hat{\pi}_0(a\mid x)}\left(\mu_{Y\mid A,X}(a,x) - \hat{\mu}_{Y\mid A,X}(a,x)\right)\right]\pi_0(da\mid x)P_X(dx) \\
&\quad + \iiint \left(\hat{\mu}_{Y\mid A,X}(a',x) - \mu_{Y\mid A,X}(a',x)\right)\pi(da'\mid x)\pi_0(da\mid x)P_X(dx) \\
&= \iint \frac{\pi_0(a\mid x)}{\hat{\pi}_0(a\mid x)}\left(\mu_{Y\mid A,X}(a,x) - \hat{\mu}_{Y\mid A,X}(a,x)\right)\pi(a\mid x)P_X(dx) \\
&\quad + \iint \left(\hat{\mu}_{Y\mid A,X}(a',x) - \mu_{Y\mid A,X}(a',x)\right)\pi(da'\mid x)P_X(dx) \\
&= \iint \left(\frac{\pi_0(a\mid x)}{\hat{\pi}_0(a\mid x)} - 1\right)\left(\mu_{Y\mid A,X}(a,x) - \hat{\mu}_{Y\mid A,X}(a,x)\right)\pi(da\mid x)P_X(dx) \\
&= \iint \pi_0(a\mid x)\left(\frac{1}{\hat{\pi}_0(a\mid x)} - \frac{1}{\pi_0(a\mid x)}\right)\left(\mu_{Y\mid A,X}(a,x) - \hat{\mu}_{Y\mid A,X}(a,x)\right)\pi(da\mid x)P_X(dx)
\end{aligned}
$$

By the Cauchy–Schwarz inequality, we have

$$
\left\|\iint \pi_0(a\mid x)\left(\frac{1}{\hat{\pi}_0(a\mid x)} - \frac{1}{\pi_0(a\mid x)}\right)\left(\mu_{Y\mid A,X}(a,x) - \hat{\mu}_{Y\mid A,X}(a,x)\right)\pi(da\mid x)P_X(dx)\right\|_{\mathcal{H}_Y}
$$

$$\leq \left( \iint \pi_0^2(a \mid x) \left( \frac{1}{\hat{\pi}_0(a \mid x)} - \frac{1}{\pi_0(a \mid x)} \right)^2 \pi(da \mid x) P_X(dx) \right)^{1/2} \left\| \mu_{Y|A,X} - \hat{\mu}_{Y|A,X} \right\|_{\mathcal{H}_Y} \cdot$$

$$\leq \left\| \pi_0 \left( \frac{1}{\hat{\pi}_0} - \frac{1}{\pi_0} \right) \right\|_{L^2(\pi P_X)} \cdot \left\| \mu_{Y|A,X} - \hat{\mu}_{Y|A,X} \right\|_{\mathcal{H}_Y}$$

If we write $r_{\pi_0}(n) = \left\| \frac{1}{\hat{\pi}_0} - \frac{1}{\pi_0} \right\|_{L^2(\pi P_X)}$ an error bound on the estimation of the inverse propensity scores, and noting that by Theorem 19, the regression error on $\left\| \mu_{Y|A,X} - \hat{\mu}_{Y|A,X} \right\|_{\mathcal{H}_Y}$ is $O_p(r_C(n,\delta,b,c))$, and we conclude the proof.

$\square$

### 11.3.3 Consistency proof

We are now in position to prove Theorem 6.

*Proof.* The decomposition in Eq (37) provides:

$$\| \hat{\chi}_{dr} - \chi(P_0) \|_{\mathcal{H}} \leq \| \mathcal{T}_n \|_{\mathcal{H}} + \| \mathcal{S}_n \|_{\mathcal{H}} + \| \mathcal{R}_n \|_{\mathcal{H}}, \tag{39}$$

The sample average $\mathcal{S}_n$ converges to 0 by the central limit theorem for Hilbert-valued random variables (see [93], see also Examples 1.4.7 and 1.8.5 in [94]), that is $\| \mathcal{S}_n \|_{\mathcal{H}_Y} = o_P(n^{-1/2})$.

Then by combining the results of Lemma 11.4 (or Lemma 11.6) and Lemma 11.5, we obtain readily that:

$$\| \hat{\chi}_{dr} - \chi(P_0) \|_{\mathcal{H}} = O_p \left( n^{-1/2} + r_C(n,\delta,b,c) r_{\pi_0}(n) \right).$$

$\square$

### 11.4 Additional details on the cross-fitted estimator

We now describe how cross-fitting [40, 37, 38, 95], can be used for our one-step estimator, following Luedtke and Chung [41]. Let $P_n^j$ denote the empirical distribution on the $j$-th fold of the samples and let $\widehat{P}_n^j \in \mathcal{P}$ denote an estimate of $P_0$ based on the remaining $j-1$ folds. The cross-fitted one-step estimator takes the form

$$\bar{\chi}_{dr}(\pi) = \frac{1}{k} \sum_{j=1}^{k} \left[ \chi \left( \widehat{P}_n^j \right) + P_n^j \hat{\psi}_n^j \right]. \tag{40}$$

Using a similar decomposition as in Eq. (37), we obtain:

$$\bar{\chi}_{dr}(\pi) - \chi(\pi)(P) = \frac{1}{k} \sum_{j=1}^{k} (P_n^j - P)\psi^\pi + \frac{1}{k} \sum_{j=1}^{k} (P_n^j - P)(\hat{\psi}_n^{j,\pi} - \psi^\pi) \tag{41}$$

$$+ \frac{1}{k} \sum_{j=1}^{k} (\chi(\hat{P}_n^j) + P\hat{\psi}_n^{j,\pi}) - \chi(\pi)(P) \tag{42}$$

Then, to prove the consistency of the estimator, we use the following triangular inequality.

$$\| \bar{\chi}_{dr}(\pi) - \chi(\pi)(P) \|_{\mathcal{H}} \leq \max_j \left\| \mathcal{T}_n^j \right\|_{\mathcal{H}} + \max_j \left\| \mathcal{S}_n^j \right\|_{\mathcal{H}} + \max_j \left\| \mathcal{R}_n^j \right\|_{\mathcal{H}}, \tag{43}$$

where $\mathcal{S}_n^j := (P_n^j - P)\psi^\pi$, $\mathcal{T}_n^j := (P_n^j - P)(\hat{\psi}_n^{j,\pi} - \psi^\pi)$, $\mathcal{R}_n^j = \chi(\hat{P}_n^j) + P\hat{\psi}_n^{j,\pi} - \chi(\pi)$ We call $\mathcal{R}_n^j$ the remainder terms and $\mathcal{T}_n^j$ the empirical process terms, $j \in \{1, k\}$.

**Lemma 11.6.** *[41, Lemma 3](Sufficient condition for negligible empirical process terms). Suppose that $\chi$ is pathwise differentiable at $P_0$ with EIF $\psi_0$. For each $j \in \{1, k\}$, $\left\| \psi_n^j - \psi_0 \right\|_{L^2(P;\mathcal{H})} = o_p(1)$ implies that $\left\| \mathcal{T}_n^j \right\|_{\mathcal{H}} = o_p\left(n^{-1/2}\right)$.*

Luedtke and Chung [91] proves this lemma via a conditioning argument that makes use of Chebyshev's inequality for Hilbert-valued random variables [96] and the dominated convergence theorem.

Then, to prove the sufficient condition, we recall the result of Lemma 11.3, which now allows to show that the cross-fitted CPME is consistent.

## 12 Details and Analysis of the Doubly-Robust Test for the Distributional Policy Effect

**Theorem 7** ((Asymptotic normality of the test statistic).). *Suppose that the conditions of Theorem 6 hold. Suppose that $\mathbb{E}_{P_0}\left[\|\varphi_{\pi,\pi'}(y, a, x)\|^4\right]$ is finite, that $\mathbb{E}_{P_0}[\varphi_{\pi,\pi'}(y, a, x)] = 0$ and $\mathbb{E}_{P_0}[\langle\varphi_{\pi,\pi'}(y, a, x), \varphi_{\pi,\pi'}(y', a', x')\rangle] > 0$. Suppose also that $r_{\pi_0}(n, \delta) \cdot r_C(n, \delta, b, c) = \mathcal{O}(n^{-1/2})$. Set $\lambda = n^{-1/(c+1/b)}$ and $m = \lfloor n/2 \rfloor$. then it follows that*

$$T_{\pi,\pi'}^\dagger \xrightarrow{d} \mathcal{N}(0, 1).$$

The proof uses the steps of Kim and Ramdas [66] and Martinez Taboada et al. [28], but is restated as it leverage the theorems and assumptions relevant to CPME. Specifically we provide a result similar on asymptotic normality to that of Luedtke and Chung [41, Theorem 2], which holds for the non-cross fitted estimator.

**Lemma 12.1.** *(Asymptotic linearity and weak convergence of the one-step estimator). Suppose that the conditions of Theorem 6 hold. Suppose also that $r_{\pi_0}(n, \delta) \cdot r_C(n, \delta, b, c) = \mathcal{O}(n^{-1/2})$. Set $\lambda = n^{-1/(c+1/b)}$ Under these conditions,*

$$n^{1/2}\left[\hat{\chi}_{dr}(\pi) - \chi(\pi)\right] \rightsquigarrow \mathbb{H},$$

*where $\mathbb{H}$ is a tight $\mathcal{H}$-valued Gaussian random variable that is such that, for each $h \in \mathcal{H}$, the marginal distribution $\langle\mathbb{H}, h\rangle_{\mathcal{H}}$ is $N\left(0, E_0\left[\langle\psi^\pi(y, a, x), h\rangle_{\mathcal{H}}^2\right]\right)$.*

This lemma can be obtained following the arguments of Luedtke and Chung [41], where the cross-fitted estimator essentially requires for $j \in \{1, 2\}$, $\mathcal{R}_n^j = o_p\left(n^{-1/2}\right)$ and $\mathcal{T}_n^j = o_P\left(n^{-1/2}\right)$ to apply Slutksy's lemma and a central limit theorem for Hilbert-valued random variables [94].

*Proof.* We split the dataset $\{(x_i, a_i, y_i)\}_{i=1}^n$ into two disjoint parts:

$$\mathcal{D} = \{(x_i, a_i, y_i)\}_{i=1}^m, \quad \tilde{\mathcal{D}} = \{(x_j, a_j, y_j)\}_{j=m+1}^n.$$

Further,

$$f_{\pi,\pi'}(y, a, x) = \frac{1}{n-m}\sum_{j=m+1}^n \langle\varphi_{\pi,\pi'}(y, a, x), \varphi_{\pi,\pi'}(y_j, a_j, x_j)\rangle, \quad T_{\pi,\pi'} = \frac{\sqrt{n}\bar{f}_{\pi,\pi'}}{S_{\pi,\pi'}}$$

where $\bar{f}_{\pi,\pi'}$ and $S_{\pi,\pi'}^2$ are the empirical mean and variance respectively:

$$\bar{f}_{\pi,\pi'} = \frac{1}{n}\sum_{i=1}^n f_{\pi,\pi'}(y_i, a_i, x_i), \quad S_{\pi,\pi'}^2 = \frac{1}{n}\sum_{i=1}^n \left(f_{\pi,\pi'}(y_i, a_i, x_i) - \bar{f}_{\pi,\pi'}\right)^2$$

We define the test statistic using the doubly robust estimators $\hat{\varphi}_{\pi,\pi'}$ and $\tilde{\varphi}_{\pi,\pi'}$, which are computed respectively from $\mathcal{D}$ and $\tilde{\mathcal{D}}$:

$$f_{\pi,\pi'}^\dagger(y_i, a_i, x_i) := \frac{1}{n-m}\sum_{j=m+1}^n \langle\hat{\varphi}_{\pi,\pi'}(y_i, a_i, x_i), \tilde{\varphi}_{\pi,\pi'}(y_j, a_j, x_j)\rangle,$$

$$\bar{f}^\dagger_{\pi,\pi'} := \frac{1}{m}\sum_{i=1}^m f^\dagger_{\pi,\pi'}(Z_i), \quad (S^\dagger_{\pi,\pi'})^2 := \frac{1}{m}\sum_{i=1}^m \left( f^\dagger_{\pi,\pi'}(Z_i) - \bar{f}^\dagger_{\pi,\pi'} \right)^2,$$

$$T^\dagger_{\pi,\pi'} := \frac{\sqrt{m}\,\bar{f}^\dagger_{\pi,\pi'}}{S^\dagger_{\pi,\pi'}}.$$

As [28, 66], the asymptotic normality results in four steps:

1. Consistency of the mean: $\quad m\bar{f}^\dagger_{\pi,\pi'} = m\bar{f}_{\pi,\pi'} + o_{\mathbb{P}}(1)$

2. Consistency of the variance: $\quad m(S^\dagger_{\pi,\pi'})^2 = m(S_{\pi,\pi'})^2 + o_{\mathbb{P}}(1)$

3. Bounded variance under conditional law: $\quad \frac{1}{\mathbb{E}[mf_{\pi,\pi'}(Z)^2|\mathcal{D}_2]} = \mathcal{O}_{\mathbb{P}}(1)$

4. Conclude with asymptotic normality: $\quad T^\dagger_{\pi,\pi'} \xrightarrow{d} \mathcal{N}(0,1)$

**Consistency of the mean** We follow the same outline as Martinez Taboada et al. [28] did, using Lemma 12.1 for the asymptotic normality of $\varphi_{\pi,\pi'}$.

**Consistency of the variance** We follow the same outline as Martinez Taboada et al. [28] did.

**Bounded variance** We now show that the denominator in the normalization of $T^\dagger_{\pi,\pi'}$ is bounded away from zero in probability:

$$\frac{1}{\mathbb{E}\left[mf^2_{\pi,\pi'}(Z) \mid \tilde{\mathcal{D}}\right]} = \mathcal{O}_P(1).$$

For compactness, we define:

$$\tau = \frac{1}{\sqrt{m}}\sum_{i=1}^m \varphi_{\pi,\pi'}\left(y_i, a_i, x_i\right), \quad \gamma = \frac{1}{\sqrt{n-m}}\sum_{j=m+1}^n \varphi_{\pi,\pi'}\left(y_j, a_j, x_j\right),$$

and

$$\hat{\tau} = \frac{1}{\sqrt{m}}\sum_{i=1}^m \hat{\varphi}_{\pi,\pi'}\left(y_i, a_i, x_i\right), \quad \tilde{\gamma} = \frac{1}{\sqrt{n-m}}\sum_{j=m+1}^n \tilde{\varphi}_{\pi,\pi'}\left(y_j, a_j, x_j\right)$$

so that:

$$m\bar{f}_{\pi,\pi'} = \langle \tau, \gamma \rangle, \quad m\bar{f}^\dagger_{\pi,\pi'} = \langle \hat{\tau}, \tilde{\gamma} \rangle,$$

$$(\sqrt{m}S_{\pi,\pi'})^2 = \sum_{i=1}^m \langle \varphi_{\pi,\pi'}(y_i, a_i, x_i), \gamma \rangle^2 - m(\bar{f}_{\pi,\pi'})^2,$$

$$(\sqrt{m}S^\dagger_{\pi,\pi'})^2 = \sum_{i=1}^m \langle \hat{\varphi}_{\pi,\pi'}(y_i, a_i, x_i), \tilde{\gamma} \rangle^2 - m(\bar{f}^\dagger_{\pi,\pi'})^2.$$

Recall that $f_{\pi,\pi'}(Z) = \langle \varphi_{\pi,\pi'}(Z), \gamma \rangle$, and that $\gamma = \frac{1}{\sqrt{n-m}}\sum_{j=m+1}^n \varphi_{\pi,\pi'}(Z_j) \in \mathcal{H}$ is a random element measurable with respect to $\tilde{\mathcal{D}}$. Conditional on $\tilde{\mathcal{D}}$, the variance of the test statistic is:

$$\mathbb{E}\left[mf^2_{\pi,\pi'}(Z) \mid \tilde{\mathcal{D}}\right] = \langle C\gamma, \gamma \rangle,$$

where $C = \mathbb{E}[\varphi(Z) \otimes \varphi(Z)]$ is the covariance operator over $\mathcal{H}$, which is compact, self-adjoint and positive semi-definite.

Using the eigendecomposition of $C$ (see Section 9.1), we write:

$$C = \sum_{j=1}^{\infty} \lambda_j v_j \otimes v_j, \quad \gamma = \sum_{j=1}^{\infty} \beta_j v_j,$$

so that

$$\mathbb{E}\left[mf_{\pi,\pi'}^2(Z) \mid \tilde{\mathcal{D}}\right] = \sum_{j=1}^{\infty} \lambda_j \beta_j^2.$$

From Assumption 16, we know that the eigenvalues satisfy $\lambda_j \leq Cj^{-b}$ for some $b \geq 1$. This decay implies that the kernel is not degenerate and the operator $C$ has at least one strictly positive eigenvalue: $\lambda_1 > 0$.

Moreover, by Lemma 12.1 and the Central Limit Theorem in separable Hilbert spaces [93], the limiting distribution of $\gamma$ is Gaussian:

$$\gamma \xrightarrow{d} \sum_{j=1}^{\infty} \sqrt{\lambda_j} N_j v_j, \quad \text{where } N_j \sim \mathcal{N}(0, 1).$$

Hence,

$$\beta_1 = \langle \gamma, v_1 \rangle \xrightarrow{d} \sqrt{\lambda_1} N_1, \quad \Rightarrow \quad \lambda_1 \beta_1^2 \xrightarrow{d} \lambda_1^2 N_1^2.$$

Therefore, the conditional variance is lower bounded:

$$\mathbb{E}\left[mf_{\pi,\pi'}^2(Z) \mid \tilde{\mathcal{D}}\right] = \sum_{j=1}^{\infty} \lambda_j \beta_j^2 \geq \lambda_1 \beta_1^2 \xrightarrow{d} \lambda_1^2 N_1^2.$$

This shows that the variance remains bounded away from zero in probability. More formally, for any $\epsilon > 0$, we can find $M > 0$ such that:

$$\mathbb{P}\left(\frac{1}{\mathbb{E}\left[mf_{\pi,\pi'}^2(Z) \mid \tilde{\mathcal{D}}\right]} > M\right) < \epsilon.$$

Hence,

$$\frac{1}{\mathbb{E}\left[mf_{\pi,\pi'}^2(Z) \mid \tilde{\mathcal{D}}\right]} = \mathcal{O}_P(1).$$

**Asymptotic normality** We now conclude the asymptotic normality of $T_{\pi,\pi'}^{\dagger}$, following Martinez Taboada et al. [28]. Suppose that $\mathbb{E}_{P_0}\left[\|\varphi_{\pi,\pi'}(y, a, x)\|^4\right]$ is finite, that $\mathbb{E}_{P_0}\left[\varphi_{\pi,\pi'}(y, a, x)\right] = 0$ and $\mathbb{E}_{P_0}\left[\langle \varphi_{\pi,\pi'}(y, a, x), \varphi_{\pi,\pi'}(y', a', x')\rangle\right] > 0$, from Kim and Ramdas [66], we have:

$$\frac{\sqrt{m}\bar{f}_{\pi,\pi'}}{\sqrt{\mathbb{E}[f_{\pi,\pi'}^2(Z) \mid \tilde{\mathcal{D}}]}} \xrightarrow{d} \mathcal{N}(0, 1), \quad \frac{S_{\pi,\pi'}^2}{\mathbb{E}[f_{\pi,\pi'}^2(Z) \mid \tilde{\mathcal{D}}]} \xrightarrow{p} 1.$$

Using the previous steps, we have:

$$m\bar{f}_{\pi,\pi'}^{\dagger} = m\bar{f}_{\pi,\pi'} + o_P(1), \quad mS_{\pi,\pi'}^{\dagger 2} = mS_{\pi,\pi'}^2 + o_P(1),$$

which implies:

$$\frac{m\bar{f}_{\pi,\pi'}^{\dagger}}{\sqrt{\mathbb{E}[mf_{\pi,\pi'}^2(Z) \mid \tilde{\mathcal{D}}]}} = \frac{m\bar{f}_{\pi,\pi'}}{\sqrt{\mathbb{E}[mf_{\pi,\pi'}^2(Z) \mid \tilde{\mathcal{D}}]}} + o_P(1) \xrightarrow{d} \mathcal{N}(0, 1),$$

Moreover,

$$\frac{mS_{\pi,\pi'}^{\dagger 2}}{\mathbb{E}[mf_{\pi,\pi'}^2(Z) \mid \tilde{\mathcal{D}}]} \xrightarrow{p} 1.$$

Taking square roots on both sides (which preserves convergence in probability by the continuous mapping theorem), we obtain:

$$\frac{\sqrt{\mathbb{E}[m f^2_{\pi,\pi'}(Z) \mid \tilde{\mathcal{D}}]}}{\sqrt{m} S^{\dagger}_{\pi,\pi'}} \xrightarrow{p} 1.$$

By Slutsky's theorem by combining the last two:

$$T^{\dagger}_{\pi,\pi'} = \frac{\sqrt{m} \bar{f}^{\dagger}_{\pi,\pi'}}{S^{\dagger}_{\pi,\pi'}} \xrightarrow{d} \mathcal{N}(0,1). \quad \square$$

$\square$

## 13  Details and Analysis of the sampling from the counterfactual distribution

**Proposition 9** ((Convergence of MMD of herded samples, weak convergence to the counter-factual outcome distribution).)**.** *Suppose the conditions of Lemma 4.1 and Assumption 8 hold. Let $(\tilde{y}_{dr,j})$ and $\tilde{P}^m_{Y,dr}$ (resp. $(\tilde{y}_{pi,j})$, $\tilde{P}^m_{Y,pi}$) be generated from $\hat{\chi}_{dr}(\pi)$ (resp. $\hat{\chi}_{pi}(\pi)$) via Algorithm 2. Then, with high probability, $\mathrm{MMD}(\tilde{P}^m_{Y,pi}, \nu(\pi)) = O_p(r_C(n,b,c) + m^{-1/2})$ and $\mathrm{MMD}(\tilde{P}^m_{Y,dr}, \nu(\pi)) = O_p(n^{-1/2} + r_{\pi_0}(n) r_C(n,b,c) + m^{-1/2})$. Moreover, $(\tilde{y}_{dr,j}) \rightsquigarrow \nu(\pi)$ and $(\tilde{y}_{\pi,j}) \rightsquigarrow \nu(\pi)$.*

*Proof.* Fix $\pi \in \Pi$. By Theorem 6, the estimated embedding $\hat{\chi}_{dr}(\pi)$ satisfies:

$$\|\hat{\chi}_{dr}(\pi) - \chi(\pi)\|_{\mathcal{H}_{\mathcal{Y}}} = O_p\left(n^{-1/2} + r_{\pi_0}(n) \cdot r_C(n,b,c)\right).$$

Let $\{\tilde{y}_t\}^m_{t=1}$ be the herded samples generated from $\hat{\chi}_{dr}(\pi)$ using Algorithm 2. According to Bach et al. [97, Section 4.2], the empirical mean embedding of these samples approximates $\hat{\chi}_{dr}(\pi)$ at rate:

$$\left\| \hat{\chi}_{dr}(\pi) - \frac{1}{m} \sum_{t=1}^m \phi_{\mathcal{Y}}(\tilde{y}_t) \right\|_{\mathcal{H}_{\mathcal{Y}}} = \mathcal{O}(m^{-1/2}).$$

By the triangle inequality:

$$\left\| \frac{1}{m} \sum_{t=1}^m \phi_{\mathcal{Y}}(\tilde{y}_t) - \chi(\pi) \right\|_{\mathcal{H}_{\mathcal{Y}}} = O_p\left(n^{-1/2} + r_{\pi_0}(n) r_C(n,b,c) + m^{-1/2}\right).$$

By definition of MMD and the reproducing property, we have:

$$\mathrm{MMD}(\tilde{P}^m_Y, \nu(\pi)) = \left\| \frac{1}{m} \sum_{t=1}^m \phi_{\mathcal{Y}}(\tilde{y}_t) - \chi(\pi) \right\|_{\mathcal{H}_{\mathcal{Y}}},$$

so the same rate applies.

For the plug-in estimator $\hat{\chi}_{pi}(\pi)$, which does not involve nuisance estimation, we obtain:

$$\|\hat{\chi}_{pi}(\pi) - \chi(\pi)\|_{\mathcal{H}_{\mathcal{Y}}} = O_p(r_C(n,b,c)),$$

yielding, with the same arguments

$$\mathrm{MMD}(\tilde{P}^m_{Y,pi}, \nu(\pi)) = O_p(r_C(n,b,c) + m^{-1/2}).$$

Finally, weak convergence of the empirical measures $\tilde{P}^m_Y$ to $\nu(\pi)$ follows from convergence in MMD norm with a characteristic kernel; see Simon-Gabriel et al. [67, Theorem 1.1] and Sriperumbudur [98]. $\square$ $\square$

# 14 Experiment details

In this Appendix we provide additional details on the simulated settings as well as additional experiment results.

## 14.1 Testing experiments

We are given a logged dataset $\mathcal{D}_{\text{init}} = \{(x_i, a_i, y_i)\}_{i=1}^n \sim P_0$, collected under a logging policy $\pi_0$. For two target policies $\pi$ and $\pi'$, the objective is to test the null hypothesis:

$$H_0 : \nu(\pi) = \nu(\pi'), \quad \text{vs.} \quad H_1 : \nu(\pi) \neq \nu(\pi'),$$

where $\nu(\pi)$ and $\nu(\pi')$ denote the counterfactual distributions of outcomes under $\pi$ and $\pi'$, respectively.

### 14.1.1 Baseline

We use baselines to evaluate the ability of our framework to detect differences in counterfactual outcome distributions induced by different target policies, compared to alternative approaches.

**Kernel Policy Test (KPT).** An adaptation of the kernel treatment effect test of Muandet et al. [17], extended to the OPE setting. It tests whether the counterfactual distributions $\nu(\pi)$ and $\nu(\pi')$ differ by comparing reweighted outcome samples using the maximum mean discrepancy (MMD). The key idea is to view both outcome distributions as being implicitly represented by importance-weighted samples from the logging distribution.

Given two importance weight vectors $w_\pi$ and $w_{\pi'}$ corresponding to the target policies $\pi$ and $\pi'$, respectively, the test computes the unbiased squared MMD statistic:

$$\widehat{\text{MMD}}_u^2 = \frac{1}{n(n-1)} \sum_{i \neq j} \left[ w_i^\pi w_j^\pi k(y_i, y_j) + w_i^{\pi'} w_j^{\pi'} k(y_i, y_j) - 2 w_i^\pi w_j^{\pi'} k(y_i, y_j) \right],$$

where $k(y_i, y_j)$ is a positive definite kernel on the outcome space (typically RBF). To obtain a $p$-value, KPT uses a permutation-based null distribution. It repeatedly permutes the correspondence between samples and their importance weights (thus preserving the outcome data while randomizing their "assignment") and recomputes the MMD statistic under each permutation. The $p$-value is estimated as the proportion of permuted statistics that exceed the observed MMD. As Muandet et al. [17], we use 10000 permutations.

**Average Treatment Effect Test (PT-linear).** A simple variant of KPT using linear kernels, testing only for differences in means. It serves as a reference for detecting average treatment differences.

**Doubly Robust Kernel Policy Test (DR-KPT).** We construct a doubly robust test statistic based on the difference of efficient influence functions:

$$T_{\pi,\pi'}^\dagger = \frac{\sqrt{m} \bar{f}_{\pi,\pi'}^\dagger}{S_{\pi,\pi'}^\dagger},$$

where $\bar{f}^\dagger$ is the empirical mean of pairwise inner products of influence function differences across data splits, and $S^\dagger$ the empirical standard deviation. The null is rejected when $T_{\pi,\pi'}^\dagger$ exceeds a standard normal threshold.

### 14.1.2 Model selection and tuning

We repeat each experiment 100 times and report test powers with 95% confidence intervals. For DR-KPT and KPT, the kernel $k_\mathcal{Y}$ is RBF. For DR-KPT the regularization parameter $\lambda$ is selected via 3-fold cross-validation in the range $\{10^{-4}, \ldots, 10^0\}$, as done in [17]. We use the median heuristic for the lengthscales of the kernel $k_\mathcal{A}$, $k_\mathcal{X}$ and $k_\mathcal{Y}$.

### 14.1.3 Simulated Synthetic Setting

The experiments are conducted in a synthetic continuous treatment setting. Covariates $x_i \in \mathbb{R}^d$ are sampled independently from a multivariate standard normal distribution $\mathcal{N}(0, I_d)$. Treatments $a_i \in \mathbb{R}$ are drawn from a Gaussian logging policy $\pi_0(a \mid x) = \mathcal{N}(x^\top w, 1)$, where the weight vector is fixed as $w = \frac{1}{\sqrt{d}} \mathbf{1}_d$. Outcomes are generated according to a linear outcome model with additive noise:

$$y_i = x_i^\top \beta + \gamma a_i + \varepsilon_i, \quad \varepsilon_i \sim \mathcal{N}(0, \sigma^2),$$

where $\beta \in \mathbb{R}^d$ is a linearly increasing vector and $\gamma \in \mathbb{R}$ controls the treatment effect strength.

We evaluate four distinct scenarios, each specifying a different relationship between the target policies $\pi$, $\pi'$, and the logging policy $\pi_0$. These scenarios are designed to induce progressively more complex shifts in the treatment distribution, affecting the downstream outcome distribution. We set the covariate dimension to $d = 5$, $\gamma = 1$ and evaluate $\beta$ in the grid $\beta = [0.1, \ 0.2, \ 0.3, \ 0.4, \ 0.5]$. $\beta$ is taken at different values across samples to reflect heterogeneity in user features and outcome interactions.

**Scenario I (Null).** This is the calibration setting in which $\pi = \pi'$. The two policies generate treatments from the same Gaussian distribution with shared mean and variance, ensuring no counterfactual distributional shift. Under the null hypothesis, we expect all tests to maintain the nominal Type I error rate.

**Scenario II (Mean Shift).** Here, the target policy $\pi$ remains identical to the logging policy, while the alternative policy $\pi'$ is a Gaussian with the same variance but a shifted mean. Specifically, $\pi'$ uses a weight vector $w' = w + \delta$, with $\delta = 2 \cdot \mathbf{1}_d$. This results in a systematic mean shift in treatment assignment, causing a change in the marginal distribution of outcomes through the linear outcome model. This tests whether the methods can detect simple, mean-level differences in counterfactual outcomes.

**Scenario III (Mixture).** In this case, the policy $\pi$ remains a standard Gaussian as in previous scenarios, while the alternative $\pi'$ is a 50/50 mixture of two Gaussian policies with opposing shifts in their means: $w_1 = w + \mathbf{1}_d$, $w_2 = w - \mathbf{1}_d$. Although the resulting treatment distribution is bimodal, its overall mean matches that of $\pi$. This scenario introduces a change in higher-order structure (e.g., variance, modality) without altering the first moment, allowing us to test whether the methods detect distributional differences beyond the mean.

**Scenario IV (Shifted Mixture).** This is the most complex scenario. As in Scenario III, the alternative policy $\pi'$ is a mixture of two Gaussian components, but this time only one component is shifted: $w_1 = w + 2 \cdot \mathbf{1}_d$, $w_2 = w$. The resulting treatment distribution under $\pi'$ differs from $\pi$ in both mean and higher-order moments. This scenario combines characteristics of Scenarios II and III and evaluates whether the tests remain sensitive to subtle and structured counterfactual shifts.

Across all scenarios, we generate $n = 1000$ samples per run and estimate importance weights for $\pi$ and $\pi'$ using fitted models based on the observed data. Specifically, we fit a linear regression model to the logged treatments $T$ as a function of the covariates $X$ to estimate the mean of the logging policy $\pi_0$, and evaluate its Gaussian density to obtain estimated propensities. This experimental design enables evaluation of the calibration and power of distributional tests under a range of realistic divergences.

In all scenarios (Tables 3–6), **DR-KPT consistently demonstrates the best computational efficiency**, with runtimes typically two orders of magnitude lower than both KPT and PT-linear. This efficiency stems from the closed-form structure of its test statistic, which avoids repeated resampling or kernel matrix permutations. In contrast, KPT relies on costly permutation-based MMD calculations, and PT-linear, while simpler, still requires repeated reweighting. For readability and to emphasize this computational advantage, we reorder the tables so that DR-KPT appears in the last row of each scenario.

We provide an additional Table 7 below with larger sample sizes and two kernels (RBF and polynomial).

Table 3: Average runtime (in seconds) for Scenario I. Values are reported as mean $\pm$ std over 100 runs.

| Method | 100 | 150 | 200 | 250 | 300 | 350 | 400 |
|---|---|---|---|---|---|---|---|
| KPT | $0.495 \pm 0.070$ | $0.740 \pm 0.039$ | $1.134 \pm 0.081$ | $1.623 \pm 0.075$ | $2.257 \pm 0.074$ | $3.204 \pm 0.118$ | $4.180 \pm 0.136$ |
| PT-linear | $0.592 \pm 0.061$ | $0.774 \pm 0.038$ | $1.060 \pm 0.051$ | $1.553 \pm 0.076$ | $2.373 \pm 0.202$ | $3.384 \pm 0.160$ | $4.358 \pm 0.251$ |
| DR-KPT | $0.004 \pm 0.005$ | $0.007 \pm 0.004$ | $0.010 \pm 0.009$ | $0.008 \pm 0.002$ | $0.013 \pm 0.007$ | $0.025 \pm 0.023$ | $0.019 \pm 0.007$ |

Table 4: Average runtime (in seconds) for Scenario II. Values are reported as mean $\pm$ std over 100 runs.

| Method | 100 | 150 | 200 | 250 | 300 | 350 | 400 |
|---|---|---|---|---|---|---|---|
| KPT | $0.559 \pm 0.044$ | $0.794 \pm 0.040$ | $1.173 \pm 0.063$ | $1.764 \pm 0.093$ | $2.301 \pm 0.085$ | $3.342 \pm 0.126$ | $4.204 \pm 0.182$ |
| PT-linear | $0.486 \pm 0.035$ | $0.767 \pm 0.037$ | $1.071 \pm 0.030$ | $1.630 \pm 0.062$ | $2.405 \pm 0.182$ | $3.738 \pm 0.251$ | $4.767 \pm 0.228$ |
| DR-KPT | $0.004 \pm 0.003$ | $0.007 \pm 0.005$ | $0.012 \pm 0.006$ | $0.014 \pm 0.008$ | $0.023 \pm 0.012$ | $0.022 \pm 0.009$ | $0.027 \pm 0.031$ |

Next, to empirically illustrate the benefits of sample-splitting in the test statistic provided in Section 5.1, we provide below in Figure 4 the same histograms as given in Figure 1. Concretly, instead of splitting the samples in $m$ and $n-m$, we use all the samples in the definition of $T_{\pi,\pi'}^{\dagger}$, $f_{\pi,\pi'}^{\dagger}(y_i, a_i, x_i)$ and in the test statistics in Eq. (14). As we can see, the resulting distribution is not normal, the QQ plot does not conclude and the test is not at all calibrated.

### 14.1.4 Warfarin Semi-Synthetic Setting

We build a semi-synthetic evaluation based on the publicly available Warfarin dosing data, following the spirit of Kallus and Zhou [4], Zenati et al. [69] and our distributional setup. Starting from the raw table from [68], we first (i) keep only subjects with a recorded stable therapeutic dose and a stable observed INR (columns 38–39 not NA and stability flag at column 37 equal to 1), (ii) construct a covariate matrix $X$ comprising demographics (gender, race, ethnicity, age group), anthropometrics (height, weight), BMI, clinical indications (8 binary indicators), selected comorbidities and concomitant medications (aspirin, acetaminophen including high dose, statins, amiodarone, carbamazepine, phenytoin, rifampin, antibiotics, antifungals, herbals), smoking, and pharmacogenetic markers (CYP2C9 and VKORC1 genotypes), and (iii) remove near-constant columns (empirical standard deviation $< 0.05$) and patients with missing/degenerate BMI (post-filter BMI $> 3 \times 10^{-3}$). Let $n$ denote the resulting sample size.

**Outcome construction (semi-synthetic).** Let TherDose$_i$ be the recorded stable therapeutic dose and let $a$ denote a candidate weekly dose. We define an expert-motivated absolute–tolerance cost

$$y(a, x) = \max\Big( |a - \text{TherDose}(x)| - 0.1 \cdot \text{TherDose}(x), \ 0 \Big),$$

and add a small observation noise $\mathcal{N}(0, 0.1^2)$. For each patient $i$, the observed outcome is

$$Y_i = y(T_i, X_i) + \varepsilon_i, \qquad \varepsilon_i \sim \mathcal{N}(0, 0.1^2).$$

**Logging policy (data-generating mechanism).** Write $\mu_T^* = \frac{1}{n} \sum_i \text{TherDose}_i$ and $\sigma_T^*$ its empirical standard deviation. Let $Z_{\text{BMI}} = \frac{\text{BMI} - \mu_{\text{BMI}}}{\sigma_{\text{BMI}}}$ be the standardized BMI. The logged treatment is generated by a contextual Gaussian policy with BMI-driven mean and homoskedastic variance:

$$T = \mu_T^* + \sigma_T^*\big(\sqrt{\theta}\, Z_{\text{BMI}} + \sqrt{1-\theta}\, \epsilon\big), \qquad \epsilon \sim \mathcal{N}(0, 1), \quad \theta = 0.5.$$

Equivalently, $T \mid X \sim \mathcal{N}\big(\mu_T^* + \sigma_T^* \sqrt{\theta}\, Z_{\text{BMI}}, \ (\sigma_T^*)^2 (1 - \theta)\big)$, i.e., a continuous normal density over

$$\frac{T - \mu_T^* - \sigma_T^* \sqrt{\theta}\, Z_{\text{BMI}}}{\sigma_T^* \sqrt{1-\theta}}.$$

**Propensity estimation.** To form importance weights for target policies, we fit a linear regression of $T$ on $X$ (no intercept in the BMI-only fit, standard scikit-learn linear model in the full fit) to obtain

Table 5: Average runtime (in seconds) for Scenario III. Values are reported as mean $\pm$ std over 100 runs.

| Method | 100 | 150 | 200 | 250 | 300 | 350 | 400 |
|---|---|---|---|---|---|---|---|
| KPT | $0.523 \pm 0.063$ | $0.836 \pm 0.025$ | $1.161 \pm 0.018$ | $1.596 \pm 0.008$ | $2.157 \pm 0.042$ | $3.174 \pm 0.014$ | $4.044 \pm 0.021$ |
| PT-linear | $0.505 \pm 0.052$ | $0.802 \pm 0.015$ | $1.134 \pm 0.013$ | $1.577 \pm 0.014$ | $2.142 \pm 0.043$ | $3.181 \pm 0.041$ | $4.051 \pm 0.024$ |
| DR-KPT | $0.004 \pm 0.003$ | $0.008 \pm 0.009$ | $0.011 \pm 0.005$ | $0.015 \pm 0.009$ | $0.020 \pm 0.010$ | $0.025 \pm 0.013$ | $0.025 \pm 0.014$ |

Table 6: Average runtime (in seconds) for Scenario IV. Values are reported as mean $\pm$ std over 100 runs.

| Method | 100 | 150 | 200 | 250 | 300 | 350 | 400 |
|---|---|---|---|---|---|---|---|
| KPT | $0.548 \pm 0.065$ | $0.839 \pm 0.014$ | $1.171 \pm 0.012$ | $1.611 \pm 0.013$ | $2.176 \pm 0.042$ | $3.239 \pm 0.032$ | $4.142 \pm 0.032$ |
| PT-linear | $0.523 \pm 0.062$ | $0.831 \pm 0.008$ | $1.160 \pm 0.014$ | $1.626 \pm 0.058$ | $2.385 \pm 0.127$ | $3.282 \pm 0.115$ | $4.153 \pm 0.043$ |
| DR-KPT | $0.004 \pm 0.005$ | $0.009 \pm 0.007$ | $0.015 \pm 0.008$ | $0.015 \pm 0.010$ | $0.018 \pm 0.010$ | $0.023 \pm 0.011$ | $0.025 \pm 0.015$ |

$\widehat{\mu}_0(X)$ and assume Gaussian residuals with variance fixed to $(\sigma_T^*)^2$. The estimated logging density is modeled as a Gaussian with mean $\widehat{\mu}_0(X)$ and variance $(\sigma_T^*)^2$:

$$\widehat{\pi}_0(T \mid X) \;\propto\; \exp\!\Big( - \tfrac{1}{2(\sigma_T^*)^2} \big\| T - \widehat{\mu}_0(X) \big\|^2 \Big).$$

and importance weights for a candidate policy $\pi$ are $w_\pi = \pi(T \mid X)/\widehat{\pi}_0(T \mid X)$, clipped at $10^5$ as in the code.

**Policies under comparison and scenarios.** We obtain a baseline linear score $w_{\text{base}}$ by regressing $T$ on $X$ and (optionally) adding small Gaussian jitter to the coefficients. Let scale $= \sigma_T^*$ and $\Delta = \sigma_T^*$ (the intercept shift unit). We evaluate four scenarios by specifying a target policy $\pi$ and an alternative policy $\pi'$ that generate normal treatments with means linear in $X$ and common scale $\sigma_T^*$. Mixtures are implemented as equal-weight mixtures of two Gaussians via intercept shifts.

- **Scenario I (Null).** $\pi = \mathcal{N}(X^\top w_{\text{base}}, \sigma_T^{*2})$ and $\pi' = \pi$. No counterfactual shift; tests should control Type I error.
- **Scenario II (Mean Shift).** $\pi = \mathcal{N}(X^\top w_{\text{base}}, \sigma_T^{*2})$ and $\pi'$ is the same Gaussian with an intercept increased by $\Delta$ (mean shift with unchanged variance). This probes sensitivity to first-moment shifts.
- **Scenario III (Mixture, mean preserved).** $\pi = \mathcal{N}(X^\top w_{\text{base}}, \sigma_T^{*2})$ and $\pi'$ is a 50/50 mixture of two Gaussians with intercepts shifted by $\pm\Delta$. The overall mean matches $\pi$ while the treatment distribution becomes bimodal, altering higher moments only.
- **Scenario IV (Shifted Mixture).** $\pi = \mathcal{N}(X^\top w_{\text{base}}, \sigma_T^{*2})$ and $\pi'$ is a 50/50 mixture where one component is intercept-shifted by $+\Delta$ and the other unshifted. Both mean and higher-order structure differ from $\pi$.

**Experimental protocol.** For each scenario, we use all $n$ patients after preprocessing and repeat over 100 independent seeds. For kernel choices on outcomes, we consider linear, polynomial, and RBF kernels; the RBF bandwidth uses the median heuristic on $\{Y_i\}$ when stable. We compare KPT (reweighted two-sample tests) and DR-KPT (doubly robust sample-split statistic) using the same weights $w_\pi, w_{\pi'}$, with the DR regularization set to $\lambda = 10^2$ in the kernel ridge step. We report empirical rejection rates at $\alpha = 0.05$ across seeds for Scenarios I–IV, thereby assessing calibration (I) and power to detect mean-only (II), higher-moment-only (III), and combined (IV) counterfactual shifts in clinically meaningful cost outcomes.

We provide in Table 8 runtime of our tests on the Warfarin data.

### 14.1.5 dSprites structured-outcome semi-synthetic data

We evaluate distributional tests on structured outcomes using the dSprites dataset [70, 71]. Each outcome is a $64 \times 64$ grayscale image obtained from a fixed renderer that decodes spatial latents while holding shape, scale, orientation, and color constant. Let latent contexts be $X \sim \mathcal{U}([0, 1]^2)$ and treatments $A \in \mathbb{R}^2$. For each context–action pair, the renderer deterministically outputs an image

$$Y := g(X, A) \in \mathbb{R}^{64 \times 64},$$

Table 7: Runtime (in seconds) of DR-KPT and KPT variants on the synthetic dataset.

| Method | 100 | 200 | 500 | 1000 | 2000 |
|---|---|---|---|---|---|
| KPT-RBF | $1.712 \pm 0.170$ | $4.786 \pm 0.157$ | $104.256 \pm 16.576$ | $366.104 \pm 99.063$ | $306.406 \pm 44.161$ |
| KPT-Poly | $1.824 \pm 0.260$ | $4.587 \pm 0.246$ | $106.186 \pm 55.371$ | $354.062 \pm 13.737$ | $334.608 \pm 81.867$ |
| KPT-Linear | $1.801 \pm 0.191$ | $4.573 \pm 0.465$ | $84.999 \pm 14.167$ | $387.368 \pm 262.004$ | $285.999 \pm 71.099$ |
| **DR-KPT-RBF** | $\mathbf{0.135 \pm 0.022}$ | $\mathbf{0.147 \pm 0.016}$ | $\mathbf{0.325 \pm 0.019}$ | $\mathbf{1.196 \pm 0.158}$ | $\mathbf{1.126 \pm 0.135}$ |
| **DR-KPT-Poly** | $\mathbf{0.118 \pm 0.011}$ | $\mathbf{0.140 \pm 0.021}$ | $\mathbf{0.314 \pm 0.020}$ | $\mathbf{1.155 \pm 0.172}$ | $\mathbf{1.119 \pm 0.126}$ |

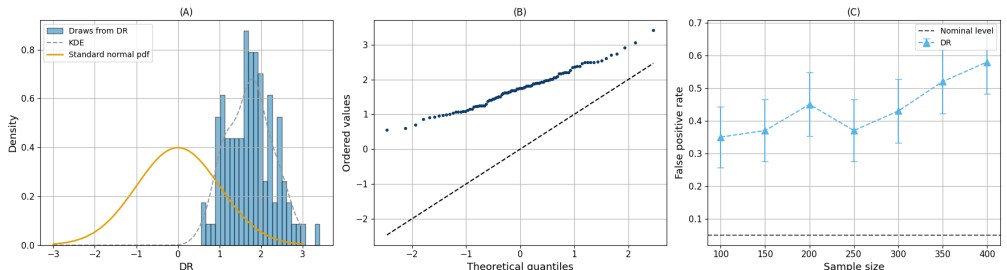

Figure 4: Illustration of 100 simulations of the non-sample-splitted DR-KPT under the null: (A) Histogram of DR-KPT alongside the pdf of a standard normal for $n = 400$, (B) Normal Q-Q plot of DR-KPT for $n = 400$, (C) False positive rate of DR-KPT against different sample sizes.

where $g$ maps the spatial latents to pixel intensities through the dSprites generative process.

**Policies and logging data.** We define contextual Gaussian policies in $\mathbb{R}^2$ with diagonal covariance $\sigma^2 I_2$. For parameters $(\theta, \beta, \sigma)$,

$$\mu_\theta(X) = \begin{bmatrix} X_1 \cos\theta + \beta \\ X_2 \sin\theta + \beta \end{bmatrix}, \qquad A \mid X \sim \mathcal{N}\big(\mu_\theta(X),\, \sigma^2 I_2\big).$$

Logged data are generated from a Gaussian logging policy with $\sigma = 0.5$. We compute analytical propensities for the target $\pi$ and alternative $\pi'$ and form importance weights

$$w_\pi = \frac{\pi(A \mid X)}{\pi_0(A \mid X)}, \qquad w_{\pi'} = \frac{\pi'(A \mid X)}{\pi_0(A \mid X)},$$

clipped at $10^5$.

**Scenarios.** We consider two scenarios that parallel our continuous-treatment experiments, now in a structured image setting:

- **Scenario I (Null).** $\pi$ and $\pi'$ share the same $(\theta, \beta, \sigma)$, hence produce identical treatment and outcome distributions.

- **Scenario IV (Policy Shift).** $\pi$ and $\pi'$ share $\theta$ and $\sigma$ but differ by an intercept shift $\beta \mapsto \beta \pm 0.3$, inducing a mean shift in $A \mid X$ and corresponding differences in the rendered image outcomes.

All other latent generative factors are fixed, ensuring that observed shifts arise purely from policy changes.

**Experimental protocol.** We generate $n = 3000$ samples per seed. Images are flattened into vectors in $\mathbb{R}^{4096}$ for kernel computations. We compare KPT (reweighted kernel two-sample tests with linear, RBF, and polynomial kernels) and DR-KPT (doubly robust, cross-fitted) using the same weights $w_\pi, w_{\pi'}$. For RBF kernels, the bandwidth is set by the median heuristic; the DR regularization parameter is fixed to $\lambda = 10^2$. Each scenario is repeated over 100 random seeds, and we report empirical rejection rates at $\alpha = 0.05$, assessing calibration (I) and power under policy shifts (IV), consistent with our synthetic and semi-synthetic Warfarin setups.

Table 8: Runtime (in seconds) of DR-KPT and KPT variants on the Warfarin dataset.

| Method | 1000 | 2000 | 3000 | 4000 |
|--------|------|------|------|------|
| KPT-RBF | $375.355 \pm 112.770$ | $1338.653 \pm 262.921$ | $2672.691 \pm 20.895$ | $5657.573 \pm 413.196$ |
| KPT-Poly | $331.831 \pm 48.219$ | $1315.537 \pm 212.152$ | $3308.014 \pm 1219.752$ | $5165.057 \pm 667.347$ |
| KPT-Linear | $364.378 \pm 35.433$ | $1302.173 \pm 300.546$ | $2651.653 \pm 103.833$ | $3623.222 \pm 373.766$ |
| **DR-KPT-RBF** | $\mathbf{0.426 \pm 0.024}$ | $\mathbf{2.530 \pm 1.447}$ | $\mathbf{5.775 \pm 0.125}$ | $\mathbf{11.701 \pm 1.869}$ |
| **DR-KPT-Poly** | $\mathbf{0.485 \pm 0.076}$ | $\mathbf{1.862 \pm 0.030}$ | $\mathbf{6.660 \pm 3.180}$ | $\mathbf{11.184 \pm 0.170}$ |

## 14.2 Sampling experiments

We study whether our estimated counterfactual policy mean embeddings (CPMEs) can be used to generate samples that approximate the true counterfactual outcome distribution. Formally, given a logged dataset $\mathcal{D}_{\text{init}} = \{(x_i, a_i, y_i)\}_{i=1}^{n} \sim P_0$ and a target policy $\pi$, we aim to generate samples $\{\tilde{y}_j\}_{j=1}^{m}$ such that their empirical distribution $\tilde{P}_Y^m$ approximates the counterfactual outcome distribution $\nu(\pi)$ under $\pi$.

### 14.2.1 Procedure

We employ kernel herding to deterministically sample from the estimated embedding $\hat{\chi}(\pi)$ in RKHS. The algorithm sequentially selects samples $\tilde{y}_1, \ldots, \tilde{y}_m$ that approximate the target embedding via greedy maximization:

$$\tilde{y}_t = \arg\max_{y \in \mathcal{Y}} \left\{ \hat{\chi}(\pi)(y) - \frac{1}{t-1} \sum_{\ell=1}^{t-1} k_{\mathcal{Y}}(\tilde{y}_\ell, y) \right\},$$

where $k_{\mathcal{Y}}$ is a universal kernel on the outcome space.

Since no comparable baselines for counterfactual sampling are available in the literature, we focus on comparing the quality of samples generated from two estimators of $\chi(\pi)$: the plug-in estimator and the doubly robust estimator. Both versions yield distinct herded samples, which we evaluate against ground truth samples generated under the target policy $\pi$.

### 14.2.2 Model selection and tuning

To report the distance metrics, we repeat each experiment 100 times and report the associated metric with 95% confidence intervals. For both plug-in and DR estimators, the kernel $k_{\mathcal{Y}}$ is RBF and the regularization parameter $\lambda$ is selected via 3-fold cross-validation in the range $\{10^{-4}, \ldots, 10^0\}$, as done in the sampling experiments of Muandet et al. [17]. We use the median heuristic for the lengthscales of the kernel $k_{\mathcal{A}}$, $k_{\mathcal{X}}$ and $k_{\mathcal{Y}}$.

### 14.2.3 Simulated Setting

We simulate logged data under different outcome models and logging policies. Covariates $x_i \in \mathbb{R}^d$ are sampled from a standard Gaussian distribution. Treatments $a_i \in \mathbb{R}$ are drawn either from a uniform distribution or from a logistic policy whose parameters depend on $x_i$. Outcomes $y_i$ are then generated via one of the following nonlinear functions:

$$\text{Nonlinear:} \quad y = \sin(x^\top \beta) + a^2 + \varepsilon, \qquad \text{Quadratic:} \quad y = (x^\top \beta)^2 + a^2 + \varepsilon,$$

where $\beta$ is a fixed coefficient vector and $\varepsilon \sim \mathcal{N}(0,1)$. For each synthetic setup, we generate logged data under the logging policy $\pi_0$ and obtain oracle samples under the target policy $\pi$ for evaluation. We set the covariate dimension to $d = 5$ and evaluate $\beta$ in the grid $\beta = [0.1, \ 0.2, \ 0.3, \ 0.4, \ 0.5]$. $\beta$ is taken at different values across samples to reflect heterogeneity in user features and outcome interactions.

Figure 5 illustrates the counterfactual outcome distributions recovered via kernel herding using both PI-CPME and DR-CPME estimators under different logging policies and outcome functions.

To assess the fidelity of the sampled distributions, we compare the empirical distribution $\tilde{P}_Y^m$ of herded samples to the true counterfactual distribution using two metrics:

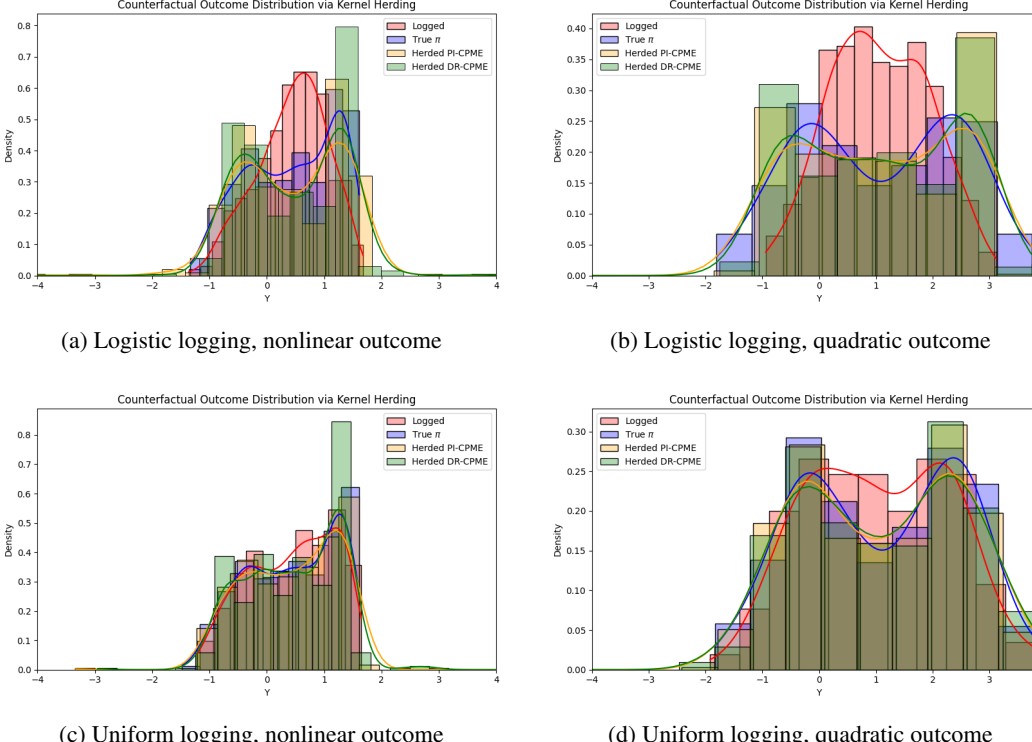

(a) Logistic logging, nonlinear outcome       (b) Logistic logging, quadratic outcome

(c) Uniform logging, nonlinear outcome       (d) Uniform logging, quadratic outcome

Figure 5: Counterfactual outcome distributions estimated via kernel herding from PI-CPME and DR-CPME samples, compared to the logged and true outcome distributions.

- **Wasserstein distance** between the sampled and ground truth outcomes,
- **Maximum Mean Discrepancy (MMD)** with a Gaussian kernel.

Table 9: Wasserstein distance between herded samples and samples from the oracle counterfactual distribution

| Method | logistic-nonlinear | logistic-quadratic | uniform-nonlinear | uniform-quadratic |
|---|---|---|---|---|
| Plug-in | 1.29e-01 ± 2.6e-01 | 1.41e-01 ± 4.9e-02 | 9.08e-02 ± 3.7e-01 | 6.78e-02 ± 1.9e-02 |
| DR | 8.60e-02 ± 2.2e-02 | 1.36e-01 ± 3.9e-02 | 5.00e-02 ± 1.5e-02 | 6.63e-02 ± 1.6e-02 |

Table 10: MMD distance between herded samples and samples from the oracle counterfactual distribution

| Method | logistic-nonlinear | logistic-quadratic | uniform-nonlinear | uniform-quadratic |
|---|---|---|---|---|
| Plug-in | 1.11e-03 ± 5.9e-03 | 9.85e-04 ± 6.0e-04 | 1.92e-04 ± 1.2e-03 | 3.31e-04 ± 2.5e-04 |
| DR | 4.38e-04 ± 3.6e-04 | 9.80e-04 ± 6.0e-04 | 6.49e-05 ± 4.4e-05 | 3.51e-04 ± 2.5e-04 |

Results in Table 9, 10 show that samples obtained from the doubly robust estimator exhibit lower discrepancy to the oracle distribution.

## 14.3 Off-policy evaluation

We are given a dataset of $n$ i.i.d. *logged* observations $\{(x_i, a_i, y_i)\}_{i=1}^n \sim P_0$. Given only this logged data from $P_0$, the goal of *off-policy evaluation* is to estimate $R(\pi)$, the expected outcomes induced by a target policy $\pi$ belonging to the policy set $\Pi$:

$$R(\pi) = \mathbb{E}_{P_\pi}\left[(Y(a))\right]. \tag{44}$$

After identification, the risk of the policy simply boils down to $R(\pi) = \mathbb{E}_{P_\pi}\left[(Y(a))\right]$, and the CPME $\chi(\pi) = \mathbb{E}_{P_\pi}\left[\phi_{\mathcal{Y}}(Y(a))\right]$ describes the risk when the feature map $\phi_{\mathcal{Y}} = y$ is linear.

### 14.3.1 Baselines

We compare our method against the following baseline estimators on synthetic datasets.

**Direct Method (DM).** The direct method [33] fits a regression model $\hat{\eta} : \mathcal{U} \times \mathcal{A} \to \mathbb{R}$ on the logged dataset $\mathcal{D}_{\text{init}} = \{(y_i, a_i, x_i)\}_{i=1}^n$, and estimates the expected reward under a target policy $\pi$ as

$$\widehat{R}_{\text{DM}}(\pi) = \frac{1}{n}\sum_{i=1}^n \int \hat{\eta}(x_i, a)\pi(a|x_i)da].$$

Since the evaluated policy differs from the logging policy $\pi_0 \neq \pi$, a covariate shift is induced over the joint space $\mathcal{A} \times \mathcal{X}$. It is well known that under the covariate shift, a parametric regression model may produce a significant bias [99]. To demonstrate this, we use a 3-layer feedforward neural network as the regressor and call it DM-NN.

**Weighted Inverse Propensity Score (wIPS).** This estimator reweights logged rewards using inverse propensity scores [90]:

$$\widehat{R}_{\text{wIPS}}(\pi) = \frac{\sum_{i=1}^n w_i y_i}{\sum_{i=1}^n w_i}, \quad w_i = \frac{\pi(a_i \mid u_i)}{\pi_0(a_i \mid u_i)}.$$

This estimator is unbiased when the true propensities are known.

**Doubly Robust (DR).** The DR estimator [33] combines the two previous methods, that is $\hat{\eta}$ and $w_i$ using:

$$\widehat{R}_{\text{DR}} = \frac{1}{n}\sum_{i=1}^n \left( \int \hat{\eta}(x_i, a)\pi(a|x_i)da + w_i(y_i - \hat{\eta}(x_i, a_i)) \right),$$

and remains consistent if either $\hat{\eta}$ or $\pi_0$ is correctly specified. We use the same parametrization for $\hat{\eta}$ as we do for the DM method and therefore call this doubly robust approach DR-NN.

**Counterfactual Policy Mean Embeddings (CPME).** We define a product kernel $k_{\mathcal{AX}}((a, x), (a', x')) = k_{\mathcal{A}}(a, a')k_{\mathcal{X}}(x, x')$, with Gaussian kernels on $a$ and $x$. The outcome kernel $k_{\mathcal{Y}}$ is linear.

**Relation to DM.** When $\hat{\eta}$ is fit via kernel ridge regression (see Exemple 9.1), the DM estimate becomes:

$$\widehat{R}_{\text{DM}}(\pi) \approx Y^\top(K + n\lambda I)^{-1} \cdot \frac{1}{n}\sum_{i=1}^n k_{\mathcal{AX}}(\tilde{a}_i, x_i)$$

where $K_{ij} = k_{\mathcal{AX}}((a_i, x_i), (a_j, x_j))$, and $\tilde{a}_i \sim \pi(\cdot \mid x_i)$. This matches the CME form proposed in [17], showing that CME/CPME is as a nonparametric version of the DM. Because kernel methods mitigate covariate shift, CMPE is consistent and asymptotically unbiased. We will therefore refer to the plug-in $\hat{\chi}_{pi}(\pi)$ and the doubly robust $\hat{\chi}_{dr}(\pi)$ estimators as DM-CPME and DR-CPME.

### 14.3.2 Model selection and tuning

Each estimator is tuned by 5-fold cross-validation procedure for OPE setting introduced in [17, Appendix B]: For the DM and DR-NN models, we vary the number of hidden units $n_h \in 50, 100, 150, 200$. For CPME and DR-CPME, the regularization parameter $\lambda$ is selected from the range $\{10^{-8}, \ldots, 10^{-3}\}$. We repeat each experiment 30 times and report mean squared error (MSE) with 95% confidence intervals. For CPME, the kernel $k_{\mathcal{Y}}$ is linear, and the regularization parameter $\lambda$ is selected via cross-validation. We use the median heuristic for the lengthscales of the kernel $k_{\mathcal{A}}$ and $k_{\mathcal{X}}$.

### 14.3.3 Simulated setting

We simulate the recommendation scenario of Muandet et al. [17] where users receive ordered lists of $K$ items drawn from a catalog of $M$ items. Each item $m \in \{1, \ldots, M\}$ is represented by a feature vector $v_m \in \mathbb{R}^d$, and each user $j \in \{1, \ldots, N\}$ is assigned a feature vector $x_j \in \mathbb{R}^d$, both sampled i.i.d. from $\mathcal{N}(0, I_d)$. A recommendation $a = (v_{m_1}, \ldots, v_{m_K}) \in \mathbb{R}^{d \times K}$ is formed by sampling items without replacement.

The user receives a binary outcome based on whether they click on any item in the recommended list. Formally, given a recommendation $a_i$ and a user feature vector $x_j$, the probability of a click is defined as

$$\theta_{ij} = \frac{1}{1 + \exp\left(-\bar{a}_i^\top x_j + \epsilon_{ij}\right)},$$

where $\bar{a}_i$ is the average of the $K$ item vectors in the list $a_i$, and $\epsilon_{ij} \sim \mathcal{N}(0, 1)$ is independent noise. The binary reward is then sampled as $y_{ij} \sim \text{Bernoulli}(\theta_{ij})$.

In our experiment, a target policy $\pi(a \mid x)$ generates a recommendation list $a = (v_{m_1}, \ldots, v_{m_K})$ by sampling $K$ items without replacement from the $M$-item catalog, where sampling is governed by a multinomial distribution. For a given user $j$, each item's selection probability is proportional to $\exp(b_j^\top v_l)$, where $b_j$ is the user-specific parameter vector. If we set $b_j = x_j$, the policy is optimal in the sense that it aligns with user preferences.

To construct the policies for the experiment, we first generate user features $x_1, \ldots, x_N$. The target policy $\pi$ uses $b_j^* = p_j \odot x_j$, where $p_j \in \{0, 1\}^d$ is a binary mask with i.i.d. $\text{Bernoulli}(0.5)$ entries, zeroing out about half the dimensions of $x_j$. The logging policy $\pi_0$ is then defined by scaling: $b_j = \alpha b_j^*$ with $\alpha \in [-1, 1]$. The parameter $\alpha$ controls policy similarity: $\alpha = 1$ recovers $\pi_0 = \pi$, while $\alpha = -1$ results in maximal divergence.

We generate two datasets $\mathcal{D}_{\text{init}} = \{(y_i, a_i, x_i)\}_{i=1}^n$ and $\mathcal{D}_{\text{target}} = \{(\tilde{y}_i, \tilde{a}_i, x_i)\}_{i=1}^n$, using $\pi_0$ and $\pi$ respectively, with shared user features $x_i$. The target outcomes $\tilde{y}_i$ are reserved for evaluation.

We evaluate performance across five setting where we vary the the values of: (i) number of observations ($n$), (ii) number of recommendations ($K$), (iii) number of users ($N$), (iv) dimension of context ($d$), (v) policy similarity ($\alpha$). Results (log scale) are shown in Figure 6.

We observe:

- All estimators generally show improved performance as the number of observations increases, except for IPS, which exhibits a slight decline between $n = 2000$ and $n = 5000$.
- The performance of all estimators deteriorates as either the number of recommendations ($K$) or the context dimension ($d$) increases.
- All estimators degrade as $\alpha \to -1$, with IPS and CPME/DR-CPME demonstrating the better robustness.
- CPME and DR-CPME consistently outperform the other estimators across most settings.
- Our proposed doubly robust method, DR-CPME, offers a performance improvement over the CPME algorithm.

### 14.4 Computation infrastructure

We ran our experiments on local CPUs of desktops and on a GPU-enabled node (in a remote server) with the following specifications:

- **Operating System:** Linux (kernel version 6.8.0-55-generic)
- **GPU:** NVIDIA RTX A4500
  - Driver Version: 560.35.05
  - CUDA Version: 12.6
  - Memory: 20 GB GDDR6

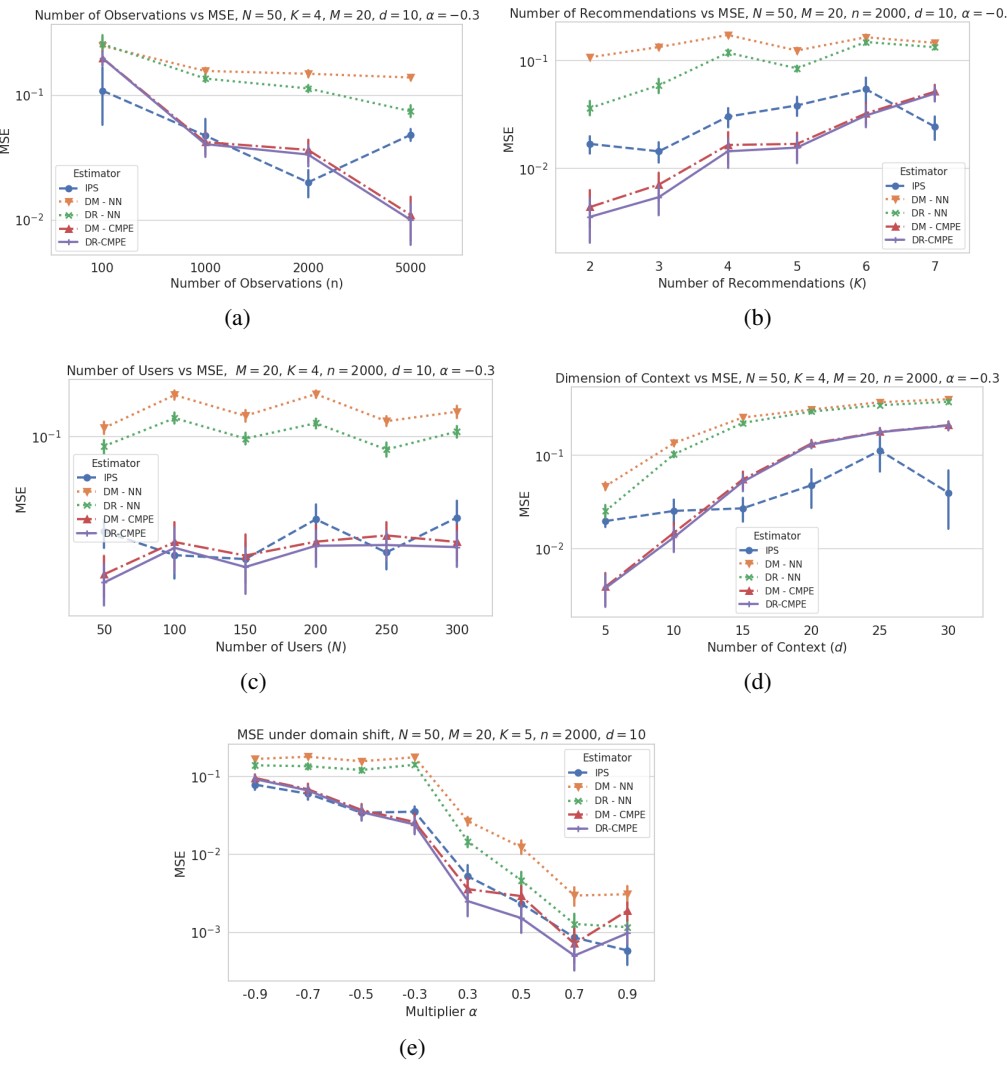

Figure 6: Mean squared error results for the off-policy evaluation experiment described in Appendix 14.3.3, reported across variations in: (a) the number of observations $n$, (b) the number of recommendations $K$, (c) the number of users $N$, (d) the context dimension $d$, and (e) the policy shift multiplier $\alpha$.

