# OpenReview forum: "Doubly-Robust Estimation of Counterfactual Policy Mean Embeddings"
_NeurIPS.cc/2025/Conference — NeurIPS 2025 poster_

### Official Review · Reviewer_Q3YK · 2025-07-01

**Clarity:** 2
**Significance:** 3
**Originality:** 3
**Rating:** 4
**Confidence:** 3

**Summary:**

The paper describes a representation of the distribution of counterfactual outcomes in reproducing kernel Hilbert spaces (RKHS). It is computed via a vector being the Hilbert space embedding of the joint distribution of actions and covariates, and an operator acting on that vector.
Further, it elaborates on the optimal estimation of these objects in detail.

**Questions:**

-	Please provide more intuition about (10).

-	“cumulative density functions”? Do you mean cumulative distribution functions?

-	Regarding “counterfactual mean embedding”. Actually, we are still talking about interventional distributions, which is rung 2 in Pearl and Mackenzies’s ladder of causation, while the term “counterfactual” suggests rung 3. Please clarify which notion of counterfactual you are referring to.

-	Line 84: “we will also abbreviate the joint distributional”  check grammar

-	Line 169: Please explain resampling strategy in a few words

-	As stated above, Section 4 is getting technically heavy and doesn’t feel self-consistent: can you provide some sentences in your rebuttal that explain eq. (9). “Riesz representer of the pathwise derivative”, all this would require some reading of the references, unless one is an expert on that specific topic.
-

-	Explain statement “which are often doubly robust for many causal parameters [31]” which causal parameters tdo you refer to and specify definition of “doubly robust”.

-	x_i instead of x in eq (11)?

**Ethical Concerns:**

["NO or VERY MINOR ethics concerns only"]

**Final Justification:**

This is a technically challenging paper which appears to be very solid work. Some concerns remain that it is not accessible to a broad audience.

**Limitations:**

the paper does not explicitly discuss limitations

**Quality:**

3

**Strengths And Weaknesses:**

Strength:

This work defines, to the best of my knowledge, an interesting step in the direction of non-parametric confounder correction. It allows analyzing outcome distributions beyond means. Although it is not the first work to leverage the RKHS framework for this purpose, it clearly states which parts are novel and which are not.

Weakness:

-	Section 4 is hard to read. While the sections before mainly require some familarity with RKHS, Section 4  requires knowledge from double robust ML, one-step estimators, effective influence functions, pathwise derivates. All these concepts occur simulateously within a short section before Lemma 4.1. This way, the paper stays a bit in the tradition of a lot of works in the field of “doubly robust ML”, where one gets easily lost in formulas without intuition. The RKHS framework is very elegant. Therefore, the authors should be more ambitious to leverage its elegance also for a presentation of dr ML that is more didactic than most of the dr ML literature out there.  I’m not saying that this is easy, but it would be worth an additional effort.

Please use the rebuttal to suggest some formulations that render Section 4 more readable for readers who have a good math background, but do not know all the details of dr ML and do not want to read all the appendix.

-	The paper uses the term “embedding” in  slightly inconsistent way. Eq. (5) is called a policy mean embedding, although it is actually an embedding of the joint distribution of actions and covariates, while the policy is defined as a conditional distribution.  Unfortunately, similar confusions already occur in the literature when people talk about “Hilbert space embedding of conditional distributions” in which conditional distributions are actually Hilbert space operators, and not vectors. In case these operators are Hilbert Schmidt class, they can be considered as vectors in the tensor product, but conditionals are often not Hilbert Schmidt, e.g., the identity.

---

> ### Author Rebuttal · Authors · 2025-07-31
>
> We thank reviewer Q3YK for their helpful and constructive feedback. In what follows, we address the main concerns raised to the best of our ability. If any points remain unclear, we would be happy to clarify further. Should our responses resolve the reviewer’s concerns, we kindly hope they will consider revising their evaluation favorably.
>
> **Clarification of Section 4**:
> > "Please use the rebuttal to suggest some formulations that render Section 4 more readable".
>
> We thank the reviewer for highlighting that Section 4 could benefit from a more didactic presentation, especially for readers less familiar with the doubly robust (DR) literature. Below are reformulations focused on specific terms/concepts of the DR literature.
>
> 1. Revision l. 191, first sentence of Section 4.
> > "To design our estimator, we rely on the recent semiparametric efficiency theory of Hilbert-valued parameters..."
>
> "To design our estimator, we build on semiparametric efficiency theory extended to Hilbert space-valued parameters, which allows us to characterize the sensitivity of the functional object of interest -the counterfactual policy mean embedding- to small changes in the data distribution $P_\pi$."
>
> 2. Remove l. 194
> > ", which are often doubly robust for many causal parameters [31]."
>
> Doubly robustness is explained in l. 212, there is no need to mention this property here, especially because it is hard to explain it without priorly giving the form in Eq. (10).
>
> 3. Revision l. 194,
>
> > Assuming the existence of an EIF $\psi^\pi$ for the CPME
> $\chi(\pi)$, the one-step correction of the plug-in estimator requires an estimate $\hat \psi^\pi$ and takes the form:
>
> "Assuming the existence of an EIF $\psi^\pi$ for the CPME $\chi(\pi)$, we apply a one-step correction to the plug-in estimator. Recall the plug-in estimator computes the target quantity by substituting estimates of a nuisance component—the conditional mean embedding $\mu_{Y|X,A}$— into its defining formula. However, when these components are estimated from data, the resulting estimator inherits their bias. The one-step estimator eliminates this leading bias by using the EIF, which captures the local sensitivity of the parameter to perturbations in the data distribution. Such an EIF provides the optimal linear adjustment to restore the statistical efficiency even when one of the nuisance estimators (outcome regression or propensity score) is imperfect.
> "
>
> 4. Revision l.196
> > "One-step estimators rely on pathwise differentiability, which captures the smoothness of a parameter along regular univariate submodels [58, 59]. When this condition holds, the efficient influence function (EIF) exists and coincides with the Riesz representer of the pathwise derivative [40]."
>
> "One-step estimators rely on *pathwise differentiability*, a smoothness condition that characterizes how a parameter changes under infinitesimal perturbations of the data-generating distribution along regular univariate submodels \citep{pfanzagl1990estimation, van1991differentiable}. When this condition holds, the *efficient influence function* (EIF) exists and serves as the best linear approximation to the parameter’s variation under such perturbations. In the Hilbert space setting, the EIF is formally the *Riesz representer* of the pathwise derivative \citep{luedtke2024}, meaning it is the unique element in the RKHS whose inner product with any score function recovers the directional derivative of the parameter."
>
>
> 5. Revision l. 205
>
> > "Note that the EIF defined in Eq. (10), similar to the EIF of the expected policy risk in OPE..."
>
> "The structure of Eq. (10) mirrors classical doubly robust EIFs: the first term adjusts for distribution shift via importance weighting, while the second term re-centers the estimate. Together, they form a Hilbert-valued function that acts as the optimal correction to the naïve plug-in embedding."
>
> 6. Revision l. 212
>
>
> > "Like all one-step estimators in OPE, our estimator enjoys a doubly robust property: it remains consistent if either $\hat \pi_0$ or $\hat \mu_{Y|X,A}$ is correctly specified."
> >
> "Like other one-step estimators in OPE, our estimator enjoys doubly robustness: i) it remains consistent even when one of the propensity score $\hat{\pi_0}$ or the conditional mean embedding $\hat{\mu_{Y \mid X,A}}$ is not consistently estimated. ii) As shown in Theorem 6 below and as most DR estimators, the estimation error of $\hat{\chi}_{dr}$ involves a product of the errors from these two nuisance components, which will improve the convergence rate."
>
> **Embedding terminology**
>
> > "The paper uses the term “embedding” in slightly inconsistent way. Eq. (5) is called a policy mean embedding, although it is actually an embedding of the joint distribution of actions and covariates, while the policy is defined as a conditional distribution."
>
> We thank the reviewer for pointing out the ambiguity in the use of the term “embedding.” Throughout the paper, we use *policy mean embedding* to refer to the RKHS embedding of the counterfactual marginal outcome distribution induced by the policy $\pi$, namely $\mu_\pi = \mathbb{E}_{\pi \times P_X}\left[\phi(a, x)\right]$. Indeed this object is formally an embedding of the joint distribution $P_X \times \pi( \cdot \mid X)$, not only the conditional distribution $\pi$. We will revise the terminology to reflect this more accurately (e.g., by referring to it as the *mean embedding under the policy-induced joint distribution*) to avoid confusion.
>
>
> **Questions:**
> 1. Intuition on EIF in (10): the EIF consists of two terms. The first is an importance-weighted residual $\frac{\pi(a \mid x)}{\pi_0(a \mid x)}\left( \phi_Y(y) - \mu_{Y \mid A,X}(a,x) \right)$, which adjusts for discrepancy between the logging and target policies. The second term, $\int \mu_{Y \mid A,X}(a', x) \pi(da' \mid x) - \chi(\pi)$, recenters the EIF to ensure it has zero mean. This decomposition mirrors classical EIFs for doubly robust estimators and makes clear how the RKHS structure naturally accommodates corrections for distribution shift between $\pi$ and $\pi_0$.
>
> 2. CDF: yes, the correct term is “cumulative distribution functions (CDFs)”, we will revise the manuscript accordingly.
> 3. "Counterfactual": our estimand corresponds indeed to interventional distributions (rung 2 in Pearl & Mackenzie's ladder), not counterfactuals in the strict sense (rung 3). We use the term "counterfactual" in line with prior work (e.g., Muandet et al., 2018; Park et al., 2021), where it refers to the distribution of potential outcomes under a hypothetical target policy $\pi$. We will clarify this usage explicitly in the revision to avoid confusion.
> 4. "joint distributional": we will change into "joint distribution".
> 5. "resampling": For continuous actions, we approximate the policy mean embedding by resampling actions from the target policy $\pi(\cdot \mid x_i)$ at each logged covariate $x_i$, and then compute the empirical average of the corresponding kernel features. This avoids costly integration and enables a practical plug-in estimator even in continuous action spaces. We will clarify this in the main text accordingly.
>
> 6. Riesz representer of the pathwise derivative. The target parameter $\chi(\pi)(P)$ lies in an RKHS and varies with the data distribution $P$. Its pathwise derivative describes this variation and defines a bounded linear functional over valid score functions. By the Riesz representation theorem, this derivative has a unique representer called the efficient influence function (EIF), which provides the most variance-efficient correction term for asymptotically linear estimators. This is because the EIF is the element of $L^2(P; H_{Y})$ that both (i) satisfies the required linearity condition imposed by the pathwise derivative, and (ii) has the smallest possible norm among all such elements. Since asymptotic variance is given by the squared norm of the influence function, the EIF minimizes the variance by construction. Consequently, among all regular estimators, using the EIF $\psi^*_\pi$ ensures minimal asymptotic variance in the RKHS norm.
>
> 7. "Doubly robust" we refer to the property in l. 212 that our one-step estimator remains consistent if either the propensity score $\pi_0(a \mid x)$ or the conditional mean embedding $\mu_{Y \mid A,X}(a,x)$ is correctly specified. We will clarify this.
>
> 8. x_i, yes there should only be $x_i$ in Eq. 11, we will correct this.
>
> **Limitations**:
>
> We will include a discussion of the method’s limitations in the revised manuscript, incorporating the points raised above as well as feedback from all reviewers.

---

> > ### Comment · Reviewer_Q3YK · 2025-08-04
> > **thank you**
> >
> > the rebuttal provides some clarification.

---

> > > ### Author Response · Authors · 2025-08-05
> > > **thank you**
> > >
> > > Thank you again for your thoughtful engagement and for acknowledging our clarifications. We're glad the rebuttal provided some clarifications. If the remaining concerns have been addressed to your satisfaction, we kindly hope you might consider updating your score accordingly. We sincerely appreciate your time and constructive feedback in rewriting Section 4.

---

### Official Review · Reviewer_Gtoa · 2025-07-01

**Clarity:** 3
**Significance:** 2
**Originality:** 2
**Rating:** 4
**Confidence:** 4

**Summary:**

This paper proposes counterfactual policy mean embedding (CPME), using kernel mean embedding of counterfactual distributions. Both a plug in estimator and a doubly robust estimator are proposed. Some theoretical analysis of these estimators is performed, and the authors show that hypothesis testing and sampling can be performed using this embedding.

**Questions:**

On line 142, what decomposition are you referring to? Also, $\mu_\pi$ can be defined without the need for the operators to exist.

In (8), does this mean that you are using the uniform distribution over the action space $\mathcal{A}$? And shouldn't you divide by the cardinality of $\mathcal{A}$ here?

**Ethical Concerns:**

["NO or VERY MINOR ethics concerns only"]

**Final Justification:**

I've read the authors' responses, and see no reason to change my positive (but not overly positive) evaluation of the paper.

**Limitations:**

There is no discussion of limitations. Societal impact is not relevant for this paper.

**Paper Formatting Concerns:**

No formatting concerns.

**Quality:**

3

**Strengths And Weaknesses:**

**Strengths**

This paper, while it does not provide anything groundbreaking, is a very solid paper which uses several existing methods and proof techniques in kernel mean embeddings, doubly robust estimation, cross U-statistics and kernel herding in the context of counterfactual policy. I think that showing that one can use existing methods and theory in different contexts is valuable.

It is in general a very well-written paper, it was a pleasure to read. Thank you very much.

**Weaknesses**

However, there are some drawbacks that make me hesitate to give a solid accept recommendation for this paper. Some specific points are listed below and in the questions section, but in general, among many things one can do for counterfactual policy analysis, hypothesis testing of whether there is *any* difference and sampling seem like small subsets. I think it is a bit of a shame that only these were studied, and I have a feeling that, in particular, the hypothesis testing was chosen because of the choice of kernel embeddings, and what they allow one to do. For sampling, I do not kernel herding is perceived as the best way to sample from distributions, unless I'm mistaken?

In Assumption 1 and below, the authors say that the SUTVA assumption and the no-interference assumption are the same, but this is not true. Actually, what is written in maths in Assumption 1(i) is the *consistency* assumption, whereas its description below on lines 117-119 is the *no interference* assumption. SUTVA is usually considered as the combination of the *consistency* and *no interference* assumptions, but there is no maths given for the no-interference assumption. In fact, on line 1124 in the appendix, the authors write that the consistency assumption is the same as SUTVA, which is also not true.

Lines 140-141: The authors brush aside the assumption for the existence of the conditional mean operator as a *regularity condition* but this is actually a severe restriction that is violated with most common kernels and domains. Moreover, the authors miss out another severe assumption, that the covariance operator $\mathcal{C}_{A,X}$ is invertible. This is another assumption that is very often violated. Proposition 4 relies on these assumptions, and therefore will not hold in most common situations.

All of this and the subsequent development can go through without operators, and the severe entailed assumptions. The conditional mean embeddings can be viewed as simply conditional expectations (which they are), and then estimation can be done by regression.

The theory in Section 3 is solid of course, but nothing new, as they are pretty much lifted from existing works.

**Minor Points**

Line 38: over -> or

Equations (8), (11) and (14) need full stops. So does the displayed equation after line 230.

Algorithm 3 is in the Appendix but referred to without any reference in Theorem 5. I think it would be better to put it in the main body.

In (13) and the line below, the hats should be removed right? This discussion is all about population statistics.

Algorithm 1: For consistency of notation, the first line should have lowercase data $x_i,a_i,y_i$.

---

> ### Author Rebuttal · Authors · 2025-07-31
>
> We thank reviewer Gtoa for their encouraging and attentive review. We have carefully considered the points raised and provide our responses below. If any issues remain ambiguous, we would be pleased to elaborate further. We hope these clarifications will support a more favorable evaluation and lead the reviewer to consider raising their score.
>
> **Weaknesses**
>
> > "among many things one can do for counterfactual policy analysis, hypothesis testing of whether there is any difference and sampling seem like small subsets"
>
> We agree that hypothesis testing and sampling cover only a subset of what can be done in counterfactual policy analysis. We highlight hypothesis testing here because it plays a key role in off-policy model selection, complementing evaluation and optimization: for instance, a learned policy may optimize expected reward, but selection may hinge on distributional properties like variance or CVaR. Moreover, sampling from the counterfactual distribution is similarly useful for policy simulation, safety checks, or data augmentation, especially in structured outcome settings. Regarding kernel herding, we considered it because it is a well-established and principled method for generating low-discrepancy samples from RKHS embeddings. That said, recent advances in MMD-based gradient flows provide powerful alternatives. In particular, "Maximum Mean Discrepancy Gradient Flow" (Arbel et al., NeurIPS 2019) and "Deep MMD Gradient Flow without Adversarial Training" (Galashov et al., ICLR 2025) propose to sample from distributions by transporting particles along a Wasserstein gradient of the MMD functional, enabling scalable generation in complex domains. Incorporating such approaches in our counterfactual framework is a promising direction for future work. We will discuss this in our paper.
>
>
> > "In Assumption 1 and below, the authors say that the SUTVA assumption and the no-interference assumption are the same, but this is not true"
>
> Thank you for this precise and helpful comment. The reviewer is right that our current presentation conflates the *consistency* assumption with the full Stable Unit Treatment Value Assumption (SUTVA). The expression $Y = Y(a)$ when $A = a$ captures *consistency*, meaning the observed outcome equals the potential outcome under the treatment actually received. However, *SUTVA* consists of both consistency and *no interference*, the latter meaning that one unit’s potential outcome does not depend on the treatment assignments of other units—formally, $Y_i(\mathbf{a}) = Y_i(a_i)$, where $\mathbf{a}$ is the treatment vector for all units. We thank the reviewer for highlighting this imprecision and will revise the main text and appendix accordingly.
>
> > " The authors brush aside the assumption for the existence of the conditional mean operator as a regularity condition but this is actually a severe restriction that is violated with most common kernels and domains."
>
> We thank the reviewer for pointing out this important issue. We agree that the existence of the Hilbert–Schmidt conditional mean operator $C_{Y \mid A,X} \in S_2(H_{A,X}, H_Y)$
> is a nontrivial assumption and, as the reviewer correctly notes, can be violated with certain combinations of unbounded domains and common kernels (e.g., Gaussian RBF on $\mathbb{R}^d$). However, we kindly disagree with the statement that this assumption is violated with *most* common kernels and domains. In fact, as shown in the recent JMLR paper by Li et al. (2024), *Towards Optimal Sobolev Norm Rates for the Vector-Valued Regularized Least-Squares Algorithm*, when the widely used Matérn kernel is employed on the $X$-space, the conditional mean function $F_\star(x) := \mathbb{E}[\phi(Y) \mid X = x]$ can be written as $C_{Y \mid X} \phi(x)$ (where $C_{Y \mid X} \in \mathcal{L}(H_X, H_Y)$ is the conditional mean operator) as long as $F_*$ is smooth enough in a traditional sense.
> Specifically, Proposition 1 in that work establishes that such a representation exists as long as $F_\star \in H^m(X; H_Y)$, the Sobolev-type vector-valued RKHS associated with a Matérn kernel on $\mathbb{R}^d$ with smoothness $m$. Thus, the existence of the conditional mean operator $C_{Y \mid X}$ is not assumed universally—it is *induced* as a representation whenever $F_\star \in H^m(X; H_Y)$, as is typical in the kernel ridge regression literature for vector-valued targets.
> In addition, the membership $F_\star \in H^m(X; H_Y)$ can be established under a mild regularity assumption on the conditional distribution $p(y \mid x)$. Concretely, by differentiating under the integral sign, one obtains
> $$
> D^\alpha F_\star(x) = \int \phi(y) D^\alpha p(y \mid x) dy,
> $$
> and thus
> $$
> \Vert F_\star\Vert_{H^m}^2 = \sum_{|\alpha| \le m} \int_X \left\Vert \int \phi(y) D^\alpha p(y \mid x) dy \right\Vert_{H_Y}^2 d\pi(x).
> $$
> Assuming the feature map is bounded $\Vert \phi(y)\Vert_{H_Y} \le \kappa$ (which is the case when the kernel is bounded), this norm is finite as soon as
> $$
> \max_{|\alpha| \leq m} \sup_{x \in \mathcal{X}} |D^\alpha p(y \mid x)| < \infty \quad \text{for all } |\alpha| \le m.
> $$
> This condition corresponds to a smoothness assumption on the conditional density—an assumption that is mild and often satisfied in practice. Therefore, under such conditions, $F_\star \in H^m(X; H_Y)$, which in turn implies that the conditional mean embedding operator $C_{Y \mid X}$ exists and acts boundedly from $H_X$ into $H_Y$.
> We will clarify these points in the revised manuscript and would be happy to provide additional technical details if needed.
>
> > "Moreover, the authors miss out another severe assumption, that the covariance operator $C_{A,X}$ is invertible."
>
> We respectfully disagree with the claim that our framework relies on the invertibility of the joint covariance operator $C_{A,X}$. Throughout our work, we do not make such an assumption, nor is it required for the validity of our results or the results we invoked from the literature. The conditional mean embedding operator $C_{Y \mid A,X}$ is defined as the solution to the vector-valued regression problem $\mu_{Y|A,X}(a, x) := \mathbb{E}[\phi(Y) \mid A = a, X = x]$, but can be estimated via regularized kernel regression without requiring the operator $C_{A,X} \in \mathcal{L}(H_{A,X})$ to be invertible.
>
> For further discussion on the invertibility of kernel covariance operators in conditional mean embeddings, we refer the reviewer to Remark 2 of the NeurIPS 2022 paper by Li et al. "Optimal Learning Rates for Regularized Conditional Mean Embedding", which clarifies that operators like $C_{XX}^{-1}$ are not globally defined in infinite-dimensional RKHSs and highlights how well-specified formulations (with CME operators) circumvent this limitation.
>
> > "All of this and the subsequent development can go through without operators, and the severe entailed assumptions. The conditional mean embeddings can be viewed as simply conditional expectations (which they are), and then estimation can be done by regression."
>
> We thank the reviewer for the suggestion. Indeed, it is possible to develop an alternative formulation that bypasses the explicit use of operators and relies purely on the regression view of conditional expectations, as done in "A Measure-Theoretic Approach to Kernel Conditional Mean Embeddings" (Park & Muandet, 2020).
>
> Nevertheless, in the well-specified case, as explained above, there exists an isomorphism between the space of conditional mean embeddings and that of conditional mean operators. This correspondence enables the use of the operator-theoretic viewpoint, which is particularly valuable for deriving convergence guarantees and facilitating theoretical analysis.
>
>
> > "The theory in Section 3 is solid of course, but nothing new, as they are pretty much lifted from existing works.
> "
>
> While the theory in Section 3 builds on established foundations, it supports a plug-in estimator that, to our knowledge, has not been previously studied in the context of distributional kernel treatment effects. Prior works—such as Muandet et al. (2018), Park et al. (2021), Fawkes et al. (2023), and Martinez-Taboada et al. (2023)—focused on IPW or DR-type estimators, and their proposed DR statistics were not analyzed through the lens of semiparametric efficiency theory.
>
>
> **Minor points**: we thank the reviewer for pointing out these corrections l.38, in Eq (8), (11), (14) and Algorithm 1. In Eq. (13) indeed, there should be hats. We will implement them and do our best given space constraints to put Algorithm 3 (and even a one-step estimator Algorithm!) in the main text and/or add references in Theorem 5, 6.
>
> **Questions**:
> 1. Decomposition
> > On line 142, what decomposition are you referring to? Also,
>  can be defined without the need for the operators to exist.
>
> The "decomposition" mentioned on line 142 prematurely refers to the decoupled representation of the counterfactual policy mean embedding as a composition of the conditional mean operator and the kernel policy mean embeddin. We aknowledge a typo, this formulation should appear after the proper definition of $\mu_\pi$ to avoid confusion. Importantly indeed, the definition of $\mu_\pi$ itself does not rely on the existence of the conditional mean operator or this decomposition.
>
> 2. Equation (8)
> > "does this mean that you are using the uniform distribution over the action space $\mathcal{A}$ ? And shouldn't you divide by the cardinality of $\mathcal{A}$ here?"
>
> We are not assuming a uniform distribution over the action space, and the expression does not require normalization by the cardinality of $\mathcal{A}$. It simply provides the analytical form of the expectation implied by definition of the kernel policy mean embedding for discrete actions under a general target policy $\pi(a \mid x)$. We will revise the text to make this clearer.
>
> **Limitations**:
>
> We will include a discussion on the limitations of the methods, including these remarks above and feedbacks from all reviewers.

---

> > ### Comment · Reviewer_Gtoa · 2025-08-04
> > **Thank you for your response**
> >
> > Dear authors,
> >
> > Thank you for your response. Most of my queries are answered, and I'm happy to keep my score.
> >
> > On the point of the regularity assumptions, I just want to make this comment. There are two different questions here, one is the *existence* of the operator itself, and the other is *estimating* it from data. Of course, just like any problem in statistics, for estimation from data, you need regularity assumptions, and the smoothness assumptions that you stated very much have the flavour of regularity assumptions that you need for *statistical learning*. On the other hand, the *existence* of the regressor itself shouldn't be reliant on any assumptions. For example, in usually regression problems, the *existence* of the conditional mean $E[Y|X]$ is not reliant on any assumptions, but assumptions on $E[Y|X]$ (e.g. smoothness) are required when proving results about how particular models *learn* this. The point I want to make is that the operator-based definition of CMEs require assumptions to even exist.
> >
> > I am aware of the works you have kindly sent me, but they also do not answer this question positively.

---

> > > ### Author Response · Authors · 2025-08-05
> > > **Clarification on Conditional Mean and Operator-Based Assumptions**
> > >
> > > We thank the reviewer again for their thoughtful engagement. Following your clarification regarding the conditional mean, we believe we are now in alignment. As you rightly point out, the conditional expectation $\mathbb{E}[\phi(Y) \mid X]$ can indeed be defined without requiring the existence of a conditional mean operator in the operator-theoretic sense.
> > >
> > > Our initial phrasing referred to the operator-based formulation, which does require additional regularity conditions—such as those related to the RKHS inclusion and smoothness—for estimation and learning rate analysis. The existence of the conditional mean in an RKHS indeed does not require such assumptions on smoothness, as supported by Park and Muandet's (2020) characterization. The operator-based CME definition applies in those circumstances where additional regularity conditions are assumed, for the purpose of estimating the conditional mean from data and for obtaining learning rates. We will clarify this point in the presentation.
> > >
> > > We appreciate the opportunity to clarify this distinction and will revise the manuscript accordingly to make it clear that our assumptions pertain to the estimation setting, not to the existence of conditional expectations per se.
> > >
> > > Thank you again for your helpful and constructive feedback.

---

### Official Review · Reviewer_k5BY · 2025-07-02

**Clarity:** 3
**Significance:** 3
**Originality:** 3
**Rating:** 4
**Confidence:** 3

**Summary:**

This paper proposes a novel framework for distributional off-policy evaluation based on Counterfactual Policy Mean Embeddings (CPME), extending kernel mean embedding methods to estimate the full distribution of counterfactual outcomes under stochastic policies. The authors introduce both a plug-in estimator and a doubly robust (DR) estimator leveraging efficient influence functions, achieving improved convergence rates. Furthermore, the work provides a doubly robust kernel test for hypothesis testing and a sampling procedure from the counterfactual distribution. The paper presents theoretical guarantees, including asymptotic normality of the test statistic and convergence rates for the estimators, and supports claims with numerical experiments.

**Questions:**

- Have you considered applying the method to more datasets, e.g., semi-synthetic benchmarks?
- How sensitive is the performance to kernel choice, particularly for structured outcomes?
- Can you provide empirical insights into computational cost and scalability on larger datasets?

**Ethical Concerns:**

["NO or VERY MINOR ethics concerns only"]

**Final Justification:**

Maintain my score. See discussion above. Thank you.

**Limitations:**

yes

**Quality:**

3

**Strengths And Weaknesses:**

**Strengths**

I think the theoretical contribution of this paper is fairly strong, including

- A novel generalization of CME, from binary interventions to general stochastic policies, significantly broadening applicability;
- Introduction of a DR estimator for the CPME is technically sound, leveraging efficient influence functions to improve convergence that's approaching parametric rates;
- Development of the DR-KPT test for comparing counterfactual distributions avoids expensive resampling techniques, improving computational efficiency.

The paper is also very well-motivated and clearly written.

**Weaknesses**
- Experiments are rather limited, currently only restricted to synthetic settings.

- Evaluation of sampling quality is limited to standard metrics (MMD, Wasserstein distance) but lacks application-driven insights (e.g., downstream policy performance or decision-making).

- Scalability analysis - While the DR-KPT test is computationally efficient relative to permutation tests, kernel methods can still struggle with large datasets. Empirical results on scalability (e.g., runtime vs. dataset size) are missing.

---

> ### Author Rebuttal · Authors · 2025-07-31
>
> We are grateful to Reviewer k5BY for their valuable comments. In what follows, we have done our best to respond to the main concerns raised. If any points remain unclear, we would be happy to clarify further. Should our responses address the reviewer’s concerns, we kindly hope they will consider revising their evaluation favorably.
>
> **Semi-synthetic experiments**
>
> > "Experiments are rather limited, currently only restricted to synthetic settings." "Have you considered applying the method to more datasets, e.g., semi-synthetic benchmarks?"
>
> We thank reviewer VKqz for this valuable suggestion. In response to similar feedback from reviewers k5BY and 9xPR, we have extended our experiments to include two semi-synthetic benchmarks. The first is based on the Warfarin dataset, using real-world patient covariates and simulated treatment policies to evaluate OPE under realistic conditions. The second leverages the d-Sprites image dataset, where outcomes are high-dimensional images rendered from latent action–context pairs. These setups allow us to assess our method on both real covariates and structured outcomes, and we observe that DR-KPT consistently outperforms baselines in both settings. We kindly refer the reviewer to answers to reviewers k5BY and 9xPR.
>
> Full details and results will be included in the revised manuscript.
>
>
> **Sampling and application driven insights**
> > "Evaluation of sampling quality is limited to standard metrics (MMD, Wasserstein distance) but lacks application-driven insights (e.g., downstream policy performance or decision-making)."
>
>
> We emphasize that sampling from the counterfactual outcome distribution has broader practical relevance—particularly when outcomes are structured (e.g., images or graphs). In such settings, visualizing samples from $\nu(\pi)$ can offer intuitive and interpretable insights into the qualitative effects of deploying different policies, which are not captured by scalar metrics like MMD or Wasserstein distance. Beyond visualization, access to samples enables application-driven evaluations, such as measuring diversity, safety, or coverage of outcomes under a given policy, and supports expert-in-the-loop assessments that are critical in high-stakes domains.
>
> Moreover, working with the entire outcome distribution embeddings allows practitioners to reason about distributional properties that directly affect decision-making—such as the presence of rare but critical outcomes (e.g., adverse drug effects), the tail behavior of policies (e.g., in finance), or the modality of outcomes. For instance, when outcome distributions are multimodal, a decision-maker may prefer policies that induce more predictable, unimodal behaviors. These considerations go beyond the expectation of the outcome distribution and are central to robust and informed policy selection.
>
> We thank the reviewer for this feedback, we will clarify and emphasize these broader perspectives in the revised manuscript.
>
> **Structured outcomes, choice of kernel**
>
> > "How sensitive is the performance to kernel choice, particularly for structured outcomes?"
>
> We thank the reviewer for raising this point. To assess sensitivity to kernel choice, we compare RBF and polynomial kernels in both the Warfarin and dSprites experiments (Tables in answers to k5BY and 9xPR). In both settings, DR-KPT achieves strong performance with both kernels, consistently outperforming baselines. While some differences appear across scenarios, the method remains robust, indicating limited sensitivity to the specific kernel choice in practice.
>
>
> **Scalability analysis**
>
> > "Scalability analysis - While the DR-KPT test is computationally efficient relative to permutation tests, kernel methods can still struggle with large datasets. Empirical results on scalability (e.g., runtime vs. dataset size) are missing." "Can you provide empirical insights into computational cost and scalability on larger datasets?"
>
> We thank the reviewer for raising this important point. We would like to clarify that Tables 1–4 in Appendix 14.1.3 report empirical results on scalability, showing how performance varies with dataset size (sample sizes ranging from 100 to 400). We acknowlege that this could have been made clearer, and we will revise the table captions and update the main text to highlight these results more explicitly.
>
> Additionally, we provide further runtime tables (in seconds) below to strengthen our empirical support in this direction.
>
> **Synthetic data**
> | Method        | 100                | 200                | 500                 | 1000                | 2000                |
> |---------------|--------------------|--------------------|---------------------|---------------------|---------------------|
> | DR-KPT-RBF    | **0.135 ± 0.022**      | **0.147 ± 0.016**      | **0.325 ± 0.019**       | **1.196 ± 0.158**       | **1.126 ± 0.135**       |
> | DR-KPT-Poly   | **0.118 ± 0.011**      | **0.140 ± 0.021**      | **0.314 ± 0.020**       | **1.155 ± 0.172**       | **1.119 ± 0.126**       |
> | KPT-RBF       | 1.712 ± 0.170      | 4.786 ± 0.157      | 104.256 ± 16.576    | 366.104 ± 99.063    | 306.406 ± 44.161    |
> | KPT-Poly      | 1.824 ± 0.260      | 4.587 ± 0.246      | 106.186 ± 55.371    | 354.062 ± 13.737    | 334.608 ± 81.867    |
> | KPT-Linear    | 1.801 ± 0.191      | 4.573 ± 0.465      | 84.999 ± 14.167     | 387.368 ± 262.004   | 285.999 ± 71.099    |
>
> **Warfarin data**
> | Method        | 1000                 | 2000                  | 3000                  | 4000                  |
> |---------------|----------------------|------------------------|------------------------|------------------------|
> | DR-KPT-RBF    | **0.426 ± 0.024**        | **2.530 ± 1.447**          | **5.775 ± 0.125**          | **11.701 ± 1.869**         |
> | DR-KPT-Poly   | **0.485 ± 0.076**        | **1.862 ± 0.030**          | **6.660 ± 3.180**          | **11.184 ± 0.170**         |
> | KPT-RBF       | 375.355 ± 112.770    | 1338.653 ± 262.921     | 2672.691 ± 20.895      | 5657.573 ± 413.196     |
> | KPT-Poly      | 331.831 ± 48.219     | 1315.537 ± 212.152     | 3308.014 ± 1219.752    | 5165.057 ± 667.347     |
> | KPT-Linear    | 364.378 ± 35.433     | 1302.173 ± 300.546     | 2651.653 ± 103.833     | 3623.222 ± 373.766     |
>
> As the tables show, DR-KPT achieves significantly lower runtimes than all KPT variants—often by several orders of magnitude—while maintaining efficiency at scale. This underscores its clear computational advantage. We will include these additional runtime experiments in the updated manuscript.

---

> > ### Comment · Reviewer_k5BY · 2025-08-04
> > **Thank you for the reponse!**
> >
> > Thank you for your thoughtful response and for conducting the additional experiment. It has addressed my questions. I will keep my score.

---

> > > ### Author Response · Authors · 2025-08-05
> > > **thank you**
> > >
> > > Thank you for your feedback. We are pleased to hear that our responses have addressed your concerns. We truly appreciate your time and support.

---

### Official Review · Reviewer_9xPR · 2025-07-03

**Clarity:** 2
**Significance:** 2
**Originality:** 2
**Rating:** 5
**Confidence:** 3

**Summary:**

The paper introduces a new method for distributional off-policy evaluation in causal inference. The method is based on so-called counterfactual mean embeddings, which allows to employ kernel-based methods. The authors combine the method with established concepts from causal inference such as doubly robust estimators and develop kernel-based statistical inference. The method is evaluated using synthetic data.

**Questions:**

- Could the authors elaborate on the novelty of their method, particularly compared to existing kernel-based methods for distributional causal inference. A list of technical novelties (method or theoretical results), ideally understandable without an extensive background in kernel-based methods would be highly appreciated.
- What is the specific advantage of using kernel-based methods in the setting studied (as compared to using other ML-based approaches combined with influence-function based estimators)?

**Ethical Concerns:**

["NO or VERY MINOR ethics concerns only"]

**Final Justification:**

Technically solid and decently novel methodology for a relevant problem. In their rebuttal, the authors addressed my concerns regarding novelty and limited experimental evaluation. I recommend acceptance.

**Limitations:**

The discussion section (Sec. 7) would benefit from a more comprehensive discussion on limitations.

**Paper Formatting Concerns:**

No concerns.

**Quality:**

3

**Strengths And Weaknesses:**

Strengths:
- Distributional off-policy evaluation is an important problem in various disciplines
- The proposed method is theoretically sound
- The experimental results seem promising


Weaknesses:
- From a causal inference point of view the novelty of the proposed method is limited: while an influence function-based estimator for the distribution of policy induced outcomes is proposed, its structure is similar to existing AIPTW-type estimators (see e.g., Kennedy et al., Semiparametric counterfactual density estimation). Thus the contribution seems to stem mostly for the kernel-based approach via counterfactual policy mean embeddings. Unfortunately, I am not very familiar with the kernel-based ML literature and thus unable to judge the novelty as compared to related literature (e.g., Muandet et al., Couterfactual mean embeddings). However, techniques for off-policy evaluation tend to be structurally very similar as compared to e.g., ATE estimation (in the world of expectations), which is why I suspect that the proposed method shares similarities with existing ones.
- One main motivation as compared to existing methods (e.g., Chernozhukov et al.) is that the proposed methods allows for complex outcome types such as images. However, the synthetic experiments are based on simple DGPs and one-dimensional continuous outcomes. Hence, (i) existing baselines seem very well applicable and should be compared against, and (ii) I think the paper would benefit from an experiment using more complex outcome distirbutions. Additionally, a case study using real-world data could strengthen the motivation.
- The method is restricted to a kernel-based approach. In contrast, many state-off-the-art causal inference methods are model-agnostic and allow for arbitrary estimation of nuisance functions. Would it be possible to instantiate the method using a different ML methodology (e.g., neural nets)?

---

> ### Author Rebuttal · Authors · 2025-07-31
>
> We thank reviewer 9xPR for their thoughtful and constructive comments. In what follows, we have done our best to address the key concerns raised in the review. If any aspect remains unclear or could benefit from further discussion, we would be happy to elaborate. If our responses satisfactorily resolve the concerns, we would greatly appreciate it if the reviewer considered updating their evaluation.
>
> **Novelty**:
>
> > "From a causal inference point of view the novelty of the proposed method is limited: while an influence function-based estimator for the distribution of policy induced outcomes is proposed, its structure is similar to existing AIPTW-type estimators (see e.g., Kennedy et al., Semiparametric counterfactual density estimation). Thus the contribution seems to stem mostly for the kernel-based approach via counterfactual policy mean embeddings."
>
> Our contribution does not stem only from the kernel-based approach in counterfactual policy mean embedding (CPME). While our estimator shares a double robustness structure reminiscent of AIPTW, extending it to **Hilbert space-valued parameters** like the CPME is far from trivial. Unlike pointwise density estimation (e.g., Kennedy et al. 2021) which is a scalar-valued parameter, our method targets the entire counterfactual distribution as a functional object in an RKHS, requiring the following non-trivial and different from scalar-valued semiparametric effiency theory:
> i) formal pathwise differentiability analysis in Hilbert norm,
> ii) derivation of a Hilbert-valued efficient influence function, and
> iii) guarantees for asymptotic linearity in infinite-dimensional spaces.
>
> To our knowledge, this is the first work to provide a one-step estimator and inference procedure for RKHS-valued causal estimands in OPE, making our contribution novel from both a statistical inference and kernel-based representation standpoint. We thank the reviewer for this feedback and we will include clarifications in our manuscript.
>
> > "Could the authors elaborate on the novelty of their method, particularly compared to existing kernel-based methods for distributional causal inference. A list of technical novelties (method or theoretical results), ideally understandable without an extensive background in kernel-based methods would be highly appreciated."
>
> Below we provide a list of technical points (methods, results) with a clear positionning with regard to existing kernel-based methods for distributional causal inference.
>
> - Novel CPME framework: The kernel-based treatment effect literature did not tackle OPE and has largely focused on distributional effect variants of binary treatment ATE/ATT (e.g., Muandet et al., 2018; Martinez-Taboada et al., 2023; Fawkes et al., 2023), with the exception of [1] (Park et al., 2021), which considers a CATE variant. In that sense, our distributional CPME embedding introduces and motivates novel objects.
> - Novel estimators and results. The plug-in and one-step estimators we propose—together with their theoretical analysis—are novel contributions of this work. The results in Theorems 5 and 6 establish consistency guarantees in RKHS that, to the best of our knowledge, are not available in prior literature. Previous works such as (Muandet et al., 2018) focused exclusively on IPW-type estimators, while (Fawkes et al., 2023) and (Martinez-Taboada et al., 2023) proposed DR-based test statistics for hypothesis testing but did not provide consistency results in the RKHS.
> - Existing t-statistic for hypothesis testing. Among these, Muandet et al., Park et al., and Fawkes et al. rely on computationally intensive permutation tests, whereas we use a doubly robust test statistic as in Martinez-Taboada et al. (2023), yielding improved efficiency and tractability. Our definition of t-statistic in OPE is novel itself but it shares similar elementary ideas and properties with that of Martinez-Taboada et al. (2023) and Kim and Randas (2024).
> - Novel EIF in RKHS, further applications. However, unlike Martinez-Taboada et al., who focus exclusively on hypothesis testing our CPME framework for OPE is grounded semiparametric efficiency theory of RKHS-valued parameters. As stated above, our analysis of the (EIF) efficient influence function in the RKHS is exclusive and novel. A consequence of these is that unlike Martinez-Taboada et al., our work improves the applications of Muandet et al, 2018 beyond hypothesis testing, such as policy sampling (Section 5.2) and recommendation (Section 14.3.3).
>
> We will update our paper to better list its technical novelties.
>
> **Structured outcomes**
>
> > "One main motivation as compared to existing methods (e.g., Chernozhukov et al.) is that the proposed methods allows for complex outcome types such as images. However, the synthetic experiments are based on simple DGPs and one-dimensional continuous outcomes. Hence, (i) existing baselines seem very well applicable and should be compared against, and (ii) I think the paper would benefit from an experiment using more complex outcome distirbutions. Additionally, a case study using real-world data could strengthen the motivation."
>
> We agree entirely with the reviewer’s suggestion regarding the need to evaluate more complex outcome distributions. To this end, we include additional experiments on the dSprites dataset (Matthey et al., 2017; Xu & Gretton, 2023), which enables structured image outcomes. Unlike scalar outcomes in our other simulations, here the counterfactual effect of a policy is assessed through 2D images rendered from latent state variables. The structural causal model is defined by latent contexts $U \sim \mathcal{U}([0,1]^2)$, actions $A \sim \pi(\cdot \mid U)$, and outcomes $Y := g(U, A) \in \mathbb{R}^{64 \times 64}$, where $g$ maps each context–action pair to an image by decoding spatial latents (X, Y) in the dSprites generative process, with all other latent factors (shape, scale, orientation) held fixed. As in previous experiments, the logging and target policies $\pi, \pi'$ are contextual Gaussians $\mathcal{N}(\mu(U), \sigma^2 I)$, where $\mu(U)$ encodes a rotated and shifted transformation of the context. In this semi-synthetic setup, the covariates $U$ and treatments $A$ are synthetic, and the outcome $Y$ is deterministically retrieved from the  image database via the fixed renderer $g$. Since we are considering distributions over images, scenarios I and IV are relevant, where outcome distributions are either identical (null case) or differ due to policy shifts, enabling controlled evaluation under limited outcome flexibility. We follow the same evaluation procedure as for the synthetic and semi-synthetic Warfarin (we kindly refer to the answer to Reviewer VKqz) experiments.
>
> | Scenario | KPT-linear | KPT-rbf | KPT-poly | DR-KPT-rbf | DR-KPT-poly |
> |----------|------------|---------|----------|------------|-------------|
> | I        | 0.394   | 0.401| 0.375 | **0.024**   | **0.000**    |
> | IV       | 0.081   | 0.054| 0.073 | **0.656**   | **0.502**    |
>
> The results above highlight the poor calibration of baseline methods under the null (Scenario I), with inflated rejection rates approaching 40%, while DR-KPT maintains near-nominal levels. In the alternative scenario (IV), DR-KPT achieves substantially higher power than all baselines, confirming its robustness and sensitivity in detecting structured distributional shifts with complex outcomes.
>
> We thank the reviewer for this suggestions; in the revised manuscript, we will include those and additionally add an illustration of Section 5.2 by sampling images from the estimated counterfactual outcome distributions, offering visual insight into the detected differences.
>
>
> **Alternative to kernel-based methods**
>
> > "The method is restricted to a kernel-based approach. In contrast, many state-off-the-art causal inference methods are model-agnostic and allow for arbitrary estimation of nuisance functions. Would it be possible to instantiate the method using a different ML methodology (e.g., neural nets)?"
>
> We use kernel methods because our purpose is to estimate a kernel embedding of the outcome distribution. Tackling distributional causal inference by embedding such distributions in a RKHS is at the core of the literature on kernel treatment effects. Indeed, kernel mean embeddings (KMEs) provide the main practical and theoretically grounded approach for distribution embeddings in causal inference and statistical learning. That being said, the reviewer raises an important point: distribution representations could also be learned via neural mean embeddings, as explored by Xu & Gretton (2022) in "A Neural Mean Embedding Approach for Back-door and Front-door Adjustment." We agree this is a compelling direction and view it as a promising avenue for future work.
>
> > "What is the specific advantage of using kernel-based methods in the setting studied (as compared to using other ML-based approaches combined with influence-function based estimators)?"
>
> The influence function in our setting is inherently tied to a reproducing kernel Hilbert space (RKHS) due to the structure of the counterfactual policy mean embedding (CPME), which is an RKHS-valued quantity. This naturally leads to the use of kernel methods, particularly for estimating the conditional mean embedding $\mu_{Y|X,A}$, which admits a closed-form solution via kernel ridge regression. Other nuisance components, such as the propensity score, are not RKHS-valued and can be estimated using any suitable machine learning method.
>
> **Limitations**
>
> We thank the reviewer for their feedback on Section 7, we will include further limitations discussed above and from other reviews.

---

### Official Review · Reviewer_VKqz · 2025-07-03

**Clarity:** 3
**Significance:** 3
**Originality:** 2
**Rating:** 4
**Confidence:** 3

**Summary:**

In this work, the authors propose Counterfactual Policy Mean Embedding (CPME), an approach that uses RKHS to represent the full counterfactual outcome distribution in OPE tasks. The authors propose plug-in and doubly-robust estimators leveraging this approach and characterize their consistency and convergence properties. The authors further validate the framework via semisynthetic experiments and illustrate that the approach demonstrates strong statistical properties.

**Questions:**

The authors argue that existing approaches leverage cumulative density functions, which are not suite for inference on more complex structured outcomes. Can the authors please clarify the limitations of CDF-based approaches in this setting?

Under the this work's formal setup, which specific technical challenges are specific to OPE beyond existing frameworks on treatment effect estimation? For example, fitting one-sided CATE can be viewed as an OPE problem.

Can the authors provide further detail on the stochastic equicontinuity condition and when it is likely to hold? This detail is important for understanding whether the proposed approach is a viable alternative to cross-fitting.

**Ethical Concerns:**

["NO or VERY MINOR ethics concerns only"]

**Final Justification:**

I thank the authors for their detailed response to my questions. The additional experiments also provide further evidence for the practical applicability of the approach. My overall (favorable) assessment of the work remains unchanged. In particular, reviewer Gtoa gives an excellent summery of my assessment :

> This paper, while it does not provide anything groundbreaking, is a very solid paper which uses several existing methods and proof techniques in kernel mean embeddings, doubly robust estimation, cross U-statistics and kernel herding in the context of counterfactual policy. I think that showing that one can use existing methods and theory in different contexts is valuable.

**Limitations:**

The authors do not provide a discussion of framework limitations. While the identifying assumptions listed by the authors (Assumption 1) are standard, the work would benefit from clearly exploring the limitations of imposing these assumptions on the reliability of the estimation approach. Ideally, this discussion should also be supported by empirical or theoretical analysis.

**Paper Formatting Concerns:**

None.

**Quality:**

3

**Strengths And Weaknesses:**

## Significance:
Off-policy evaluation is a core problem relevant to a number of domains. The authors' approach targeting counterfactual distributions of outcomes enables recovering a richer space of target parameters than traditional approaches targeting means. The authors show clear applications of the framework in the form of hypothesis testing. The embedding-based approach is also consistent with a range of modern applications, including inference over graph, text, and image modalities.

## Originality:
The authors clearly situate the proposed framework in related work. The application of counterfactual mean embeddings to OPE appears to be novel. However, the overlap between this OPE setting and prior applications to distributional treatment effects (e.g., 1,2) limits originality to an extent and affects my score. The author's note on how this work differs from Mundanet et al. (i.e., propensity is unknown in advance) is helpful, and the introduction of an alternative to crossfitting (line 215) to improve statistical power is also a benefit of the proposed approach.

[1]  Conditional distributional treatment effect with kernel conditional mean embeddings and u-statistic regression. https://arxiv.org/abs/2102.08208
[2]  An efficient doubly-robust test for the kernel treatment effect. https://arxiv.org/abs/2304.13237

## Quality:

The work is generally well-supported by theoretical and empirical analysis. Theoretical details in the appendix supports claims made in the main text. Further, applications with testing (Section 5) illustrate why a distributional approach is useful in supporting downstream inference applications.

However, while the fully synthetic experimental setup is sufficient for demonstrating that the proposed approach satisfies basic coverage properties, it could be strengthened. In particular, the work would benefit from a semi-synthetic or real-world data application illustrating how the framework can be applied to real-world OPE tasks. The fully synthetic setup is limiting. However, the additional synthetic results reported in the appendix do strengthen the empirical validation of the approach.


## Clarity.

The writing and mathematical exposition are generally clear. However, the work has several typos and would benefit from a careful editing pass.

---

> ### Author Rebuttal · Authors · 2025-07-31
>
> We thank reviewer VKqz for their detailed and thoughtful review. Below, we aimed to address the reviewer’s concerns point by point. If any aspects remain unclear, we would be glad to provide further clarification. Should our responses satisfactorily resolve the reviewer’s concerns, we hope they will consider raising their score accordingly.
>
> **Related works**:
>
> > "The application of counterfactual mean embeddings to OPE appears to be novel. However, the overlap between this OPE setting and prior applications to distributional treatment effects (e.g., 1,2) limits originality to an extent and affects my score."
>
> While our approach tackles indeed adjacent problems to [1] (Park et al, 2021) and [2] (Martinez-Tabaoda et al, 2023) we would like to emphasize the nontrivial technical novelty of CPME and its associated methods. Unlike [2] who focused their analysis exclusively on the asymptotic normality of the test statistic in kernel two sample tests, we provided a general framework but also a one-step estimator which convergence rate was rigorously proven using semi-parametric theory in RKHS. The latter estimator was compared to a plug-in estimator which is also our novel contribution. Our focus on semiparametric effiency of the estimation of that CPME allows us to derive guarantees and expand to more applications than [2], for example in Section 5.2 (sampling) and Appendix 14.3.3 (recommendation).
>
> **Semi-synthetic/real-world application**:
>
> > " In particular, the work would benefit from a semi-synthetic or real-world data application illustrating how the framework can be applied to real-world OPE tasks."
>
> We thank the reviewer for suggesting a semi-synthetic and real-world validation. In response, we conduct additional experiments using 1-the Warfarin dataset (PharmGKB, 2009), commonly used in continuous-treatment causal inference (Kallus & Zhou, 2018) and 2- on the d-sprite dataset (Matthey et al., 2017; Xu & Gretton, 2023) with structured outcomes, as requested by Reviewer 9xPR.
>
> 1- **Warfarin:** The setup simulates treatment assignments based on standardized BMI and Gaussian noise, while outcomes are defined via a loss function penalizing predictions outside a ±10% margin of expert therapeutic doses $A_i^*$. Logging and target policies are modeled as stochastic perturbations around this process, enabling computation of importance weights in an off-policy evaluation setting.
>
> We mirror the synthetic testing protocol of Section 6 by evaluating four scenarios: I (Null), II (Mean Shift), III (Mixture), and IV (Shifted Mixture), each introducing distinct shifts in the treatment and outcome distribution. As in the synthetic setup, both outcome models and propensity scores are unknown and learned from data. We include the same baselines using our adapted Kernel Policy Test (KPT) with estimated propensities and a linear-kernel variant (KPT-linear). Following Reviewer k5BY’s feedback, we also report results with RBF and polynomial kernels.
>
> The results below demonstrate that DR-KPT is well-calibrated under the null (Scenario I), with near-nominal rejection rates. In all alternative scenarios (II–IV), DR-KPT outperforms the baselines or matches the top performance, showing strong power across complex distributional shifts.
>
> | Scenario | KPT-linear | KPT-rbf | KPT-poly | DR-KPT-rbf | DR-KPT-poly |
> |----------|------------|---------|----------|------------|-------------|
> | I        | 0.00       | 0.00    | 0.00     | **0.02**       | **0.06**        |
> | II       | 0.77       | 0.01    | 0.29     | **0.80**       | 0.66        |
> | III      | **1.00**       | 0.00    | 0.66     | **0.99**       | 0.95        |
> | IV       | 0.24       | 0.00    | 0.11     | **0.76**       | 0.55        |
>
> We will include a detailed description of the Warfarin dataset, the experiment setting and the associated results in the manuscript.
>
> 2- **d-Sprite:** We kindly refer reviewer VKqz to the answer given to reviewer 9xPR for a more detailed presentation of our experiments on the structured outcome d-sprite dataset.
>
> **Questions**:
>
> 1) **Structured outcomes**
>
> >"Can the authors please clarify the limitations of CDF-based approaches in this setting?"
>
> CDF-based approaches are not applicable in the setting where structured outcomes (e.g., images, graphs) lack a natural total order, making notions like $\mathbb{P}(Y \leq y)$ ill-defined. In contrast, our CPME framework leverages kernel embeddings, which remain well-defined and effective on arbitrary outcome spaces.
>
> 2) **Existing frameworks, CATE**:
>
> > "Under the this work's formal setup, which specific technical challenges are specific to OPE beyond existing frameworks on treatment effect estimation?"
>
> To the best of our knowledge, kernel-based treatment effect literature has largely focused on RKHS variant of binary treatment ATE/ATT (Muandet et al., 2018; Martinez-Taboada et al., 2023; Fawkes et al., 2023), except [1] (Park et al, 2021). Our OPE setup introduces distinct challenges: (i) the estimand integrates the conditional outcome embedding over the policy $\pi(a \mid x)$, rather than evaluating it at a fixed $a$; this requires defining and estimating novel functionals of the conditional mean embedding over $\pi \times P_X$, rather than existing methods evaluating it at a single atomic $a$ (see Eq (8) and Appendix 10.3). (ii) we address stochastic interventions and continuous actions beyond binary treatment settings, whereas fitting a CATE estimate would require extra efforts with continous treatments, instead we use an embedding on $\pi$ which directly integrates over the actions sampled from $\pi$. (iii) the variance issue due to support mismatch between the two $\pi$ and $\pi_0$ in importance weighting can be particularly severe, this led us to introduce a DR correction anchored in RKHS semiparametric efficient theory unlike all previous works.  Eventually, (iv) should the reviewer refers to a scalar-valued CATE, we state that our target is an RKHS-valued mean embedding, making the analysis of the estimation and inference more technically demanding.
>
> 3) **Stochastic equicontinuity**:
> > "Can the authors provide further detail on the stochastic equicontinuity condition and when it is likely to hold?"
>
> To build more intuition, stochastic equicontinuity essentially means that the empirical process is "smooth" in a probabilistic sense—it does not fluctuate wildly when the functions it is applied to vary slightly. Formally, stochastic equicontinuity of the empirical process $\mathcal{T}_n$ (as in Definition 11.4) is equivalent to showing
> $\mathcal{T}_n = (P_n - P)(\widehat{\psi}_n^\pi - \psi^\pi) = o_P(n^{-1/2})$.
>
> Under Assumptions 1, 3, 15, 17, and 21, the functions $\widehat{\psi_n}^\pi - \psi^\pi $ lie in a shrinking ball of the RKHS $\mathcal{H}_\mathcal{Y}$ (see Lemma 11.3).
>
> If the output kernel $k_\mathcal{Y}$ is a $C^\infty$ Mercer kernel, then this ball has exponentially decaying covering numbers (see Cucker & Smale, 2002 \[75, Thm D]).
> Thus, we can apply Theorem 6 of Park & Muandet, 2023 \[60] on the class
> $\mathcal{G} = \{ \widehat{\psi_n}^\pi - \psi^\pi \} \subset L^2(P; \mathcal{H}_\mathcal{Y})$,
> which ensures that the empirical process $\mathcal{T}_n$ is asymptotically equicontinuous.
>
> Note that more generally, stochastic equicontinuity holds when the output kernel $k_{\mathcal{Y}}$ is smooth and the RKHS ball satisfies one of:
> - it is totally bounded and has finite Assouad dimension (Theorem 4, Park & Muandet, 2023), meaning local covering numbers scale uniformly across the set;
> - it has finite box-counting dimension (Theorem 5, Park & Muandet, 2023), capturing how the number of covering balls grows as their radius shrinks;
> - or it has exponential entropy (Theorem 6, Park & Muandet, 2023).
>
>
> **Limitations**:
>
> We agree that the limitations of Assumption 1 merit further discussion. We will include a brief discussion on potential failures of these assumptions—particularly violations of exchangeability or positivity—and their implications for the reliability of our estimator. A more extensive empirical or theoretical analysis of these violations is an important direction we will leave for future work.

---

> ### Comment · Reviewer_VKqz · 2025-08-07
>
> I thank the authors for their detailed response to my questions. The additional experiments also provide further evidence for the practical applicability of the approach. My overall (favorable) assessment of the work remains unchanged. In particular, reviewer Gtoa gives an excellent summary of my assessment :
>
> > This paper, while it does not provide anything groundbreaking, is a very solid paper which uses several existing methods and proof techniques in kernel mean embeddings, doubly robust estimation, cross U-statistics and kernel herding in the context of counterfactual policy. I think that showing that one can use existing methods and theory in different contexts is valuable.

---

### Decision · Program_Chairs · 2025-09-17

**Decision:**

Accept (poster)

**Comment:**

I concur with the reviewers’ positive view of the submission and recommend acceptance. The rebuttal and discussion added semi-synthetic and structured-outcome experiments that strengthen the empirical support. I ask the authors to incorporate these results, clarify the role of assumptions and limitations, and polish some technical exposition (e.g., SUTVA, operator assumptions, embedding terminology) in the final version.